



# An LES-based airborne Doppler lidar simulator for investigation of wind profiling in inhomogeneous flow conditions

Philipp Gasch[1], Andreas Wieser[1], Julie K. Lundquist[2], and Norbert Kalthoff[1]

[1]Institute of Meteorology and Climate Research, Karlsruhe Institute of Technology, Karlsruhe, Germany
[2]Department of Atmospheric and Oceanic Sciences, University of Colorado Boulder, Boulder, Colorado, USA

**Correspondence:** Philipp Gasch (philipp.gasch@kit.edu)

**Abstract.** Wind profiling by Doppler lidar is common practice and highly useful in a wide range of applications. Airborne observations can provide additional insights to ground-based systems by allowing for spatially resolved and targeted measurements. This study prepares the ground for an upcoming airborne Doppler lidar system by investigating the measurement process theoretically. To evaluate the future system characteristics and measurement accuracy, a first LES-based airborne Doppler lidar simulator (ADLS) has been developed.

The accuracy of retrieved wind profiles under inhomogeneous flow conditions in the boundary layer is investigated. In general, when using reasonable system setups, wind profiling is possible with acceptable error margins. Results allow for determination of preferential system setups and wind profiling strategies. Under the conditions considered, flow inhomogeneities exert the dominant influence on wind profiling error. In comparison, both the errors caused by random radial velocity fluctuations due to laser system noise and beam pointing inaccuracy due to system vibrations are of smaller magnitude.

Airborne Doppler lidar wind profiling at low wind speeds ($< 5 \, \mathrm{m \, s^{-1}}$) requires adequate system setups, retrieval strategies and quality filtering as the retrieved wind speeds can be biased otherwise. The utility of quality filtering criteria for wind profile reliability (coefficient of determination and condition number) is examined. While the filtering by the condition number is useful for all circumstances, an inadequate coefficient of determination threshold can bias the retrieved wind speeds. Even with strict quality filtering criteria applied, considerable wind profile retrieval error can exist, especially for steep scan elevation angles of more than 70° from the horizontal or short horizontal averaging distances.

## 1 Introduction

Doppler lidar has experienced rapidly growing importance and usage in remote sensing of atmospheric winds over the past decades (Weitkamp et al., 2005). Sectors with widespread usage include boundary layer meteorology, wind energy and airport management. Compared to ground-based Doppler lidar systems, airborne systems can provide advantages and are seen as a promising tool for future research (Baker et al., 1995, 2014; Davis et al., 2018). Airborne Doppler lidar extends the spatial coverage of flow phenomena, enabling stream-wise and span-wise investigation of flow phenomena (Kiemle et al., 2011; De Wekker et al., 2012; Chouza et al., 2016b) as well as targeted observations with rapid deployment (Weissmann et al., 2005;



Zhang et al., 2018). Further, airborne Doppler lidar can serve as a testbed and validation tool for upcoming and existing space-based Doppler lidar systems (Paffrath et al., 2009; Lux et al., 2018; ESA, 2018; Baidar et al., 2018; Tucker et al., 2018).

Due to their benefits various airborne Doppler lidar systems have been developed in the past. Most of the systems are based on long-range aircraft flying in the upper troposphere at high speeds (O(250m s$^{-1}$)), thereby they are destined for sensing the

free troposphere due to the coarser spatial resolution (O(5 km) for wind profiling). Some deployments of airborne Doppler lidar have been reported based on medium-range aircraft flying in the lower troposphere and at slower speeds (O(50m s$^{-1}$)). These systems yield higher spatial resolution (O(1 km) for wind profiling) and enable boundary-layer studies. Thus far, only results for wind profiling of the mean horizontal wind have been reported (De Wekker et al., 2012; Godwin et al., 2012; Koch et al., 2014). As of yet, a study analyzing the vertical wind through nadir or other measurement geometries such as Dual-Doppler

have not been attempted for the medium-range aircraft. This study provides a critical component for an airborne Doppler lidar system for boundary-layer research by investigating flexible scanning geometries theoretically, considering both wind profiling and nadir measurements of the wind field.

Airborne Doppler systems are often used to assess wind profiles of the mean horizontal flow (Weissmann et al., 2005; De Wekker et al., 2012; Kavaya et al., 2014; Guimond et al., 2014; Tian et al., 2015; Baidar et al., 2018). The theory and problems

associated with airborne Doppler wind profiling are very similar between lidar and radar systems. Therefore, no distinction is made between the studies using either of the two instruments in the following, unless necessary due to explicit differences.

Similar to ground-based wind profiling, airborne wind profiling is usually conducted by scanning in conical scans along the flight path. Retrieval of the mean wind vector can be achieved through inversion of the beam matrix, yielding a least-squares solution to the problem (Leon and Vali, 1998). De Wekker et al. (2012) and Tian et al. (2015) apply the original Velocity-

Azimuth-Display (VAD) analysis directly, while neglecting the non-standard beam geometry due to aircraft movement. Other methods at higher computational cost exist as well. Guimond et al. (2014) discuss a variational approach which can improve the traditional solution by adding anelastic mass continuity constraints on the estimated solution. Accumulation of the Doppler spectra can be conducted for Doppler lidar and has been shown to extend the retrieval limits in clear air conditions with little return signal (Weissmann et al., 2005; Witschas et al., 2017).

Most currently used airborne wind profiling approaches assume homogeneous conditions throughout the sample volume. Especially in turbulent environments such as the atmospheric boundary layer, this condition is rarely fulfilled and wind profiling at high spatial resolution remains challenging (Leon and Vali, 1998; Guimond et al., 2014; Tian et al., 2015). The problem is further intensified by the fact that airborne profiling systems use high elevation angles (closer to nadir) in order to constrain the footprint of the measurement. As a result, the retrieved radial velocity is strongly influenced by the vertical wind along

the scan circle. Thereby, deviations from the mean homogeneous conditions can lead to non-negligible errors in the retrieved mean wind vector (Tian et al., 2015; Bucci et al., 2018). This problem also applies to ground-based wind profiling in complex terrain, at short sampling durations, e.g. when analyzing single scans, or for short sector scans (Cheong et al., 2008; Bingöl et al., 2009; Wang et al., 2015).

The coefficient of determination is often used to detect a violation of the homogeneity assumption (Päschke et al., 2015;

Wang et al., 2015). When using this approach, it is assumed that deviations from the homogeneous state show up as deviations



of the measurements from the LSQ-fit. If the matrix inversion is performed based on a singular value decomposition, additional quality criteria such as the condition number, describing the degree of collinearity among the model geometry, become available and are frequently utilized (Boccippio, 1995; Holleman, 2005; Cheong et al., 2008; Shenghui et al., 2014; Päschke et al., 2015; Wang et al., 2015). However, as shown already by Koscielny (1984), a linear change in the vertical wind biases the retrieved horizontal components and a linear change in the horizontal components biases the retrieved vertical component. Both deviations are not detectable as deviations from the LSQ-fit.

A common method to assess the reliability of retrieved Doppler lidar wind profiles is by comparison to wind profiles from other measurement systems. For airborne systems, additional problems exist for in-situ comparisons: instrumented towers are of limited use due to their small vertical extent and fixed location (as well as lidar measurement problems due to chirp close to the ground detailed by Godwin et al. (2012)), simultaneous aircraft measurement are challenging and expensive to execute (and still suffer from co-location problems) and systems with similar remote sensing characteristics also suffer from co-location problems and results can show large differences (De Wekker et al., 2012; Tian et al., 2015; Bucci et al., 2018). Therefore, the most prominent approach is the comparison of retrieved wind profiles to radiosondes and/or dropsondes, which can provide verification if conducted for a sufficiently large dataset (Weissmann et al., 2005; Chouza et al., 2016a; Bucci et al., 2018). Nevertheless, both systems still exhibit very different sampling characteristics and volumes, often making a direct comparison of the results challenging as it is difficult to determine if the observed deviations occur due to the differing sampling volumes, spatial inhomogeneity or actual instrument error.

Due to these challenges, idealized simulations of Doppler measurement systems can provide detailed insight into the limitations and capabilities of these systems. Early studies determined the representativeness of Doppler lidar measurements based on a statistical description of turbulence and for idealized system set-ups. Many of the first studies emphasized the effects of measurement geometry and turbulence on system characteristics and performance (Waldteufel and Corbin, 1978; Boccippio, 1995; Banakh et al., 1995; Baker et al., 1995; Frehlich, 2001; Banakh and Werner, 2005). As a result, the reliability of measured radial velocities under different turbulent intensity conditions and its impact on retrieval quality are well described.

With increasing computational capabilities a numerical approach based on simulated atmospheric wind fields became feasible. For ground-based systems a number of investigations detailing the error characteristics associated with wind profiling exist based on LES-simulated wind fields. These include Muschinski et al. (1999); Scipión et al. (2009); Scipion (2011); Stawiarski et al. (2013); Wainwright et al. (2014); Lundquist et al. (2015) and Klaas et al. (2015).

For airborne (or satellite-based) systems, the moving platform alters the measurement process and viewing geometry, thereby introducing new problems. These challenged have been investigated with statistical models (Baker et al., 1995; Gamache et al., 1995; Frehlich, 2001) or real measurement data (Leon and Vali, 1998; Weissmann et al., 2005; Tian et al., 2015; Chouza et al., 2016a). For airborne systems, Lorsolo et al. (2013) and Guimond et al. (2014) show the importance of model-based simulator studies, while relying on coarser resolution model output and focused on errors introduced due to the measurement system inaccuracies. So far, to our knowledge, a simulation of an airborne wind profiling system with complex scanning geometries and high resolution atmospheric wind fields (O(100 m)) is missing.





Extending previous studies, this work aims to investigate the impact of wind field inhomogeneities on airborne wind profiling at high resolution (evaluation of single scans, O(1 km) for wind profiling). The focus of this study is on the error introduced by the wind retrieval algorithm due to the mismatch between assumed homogeneous wind field models and the wind field inhomogeneities present during the measurement process. Towards this goal, an LES-based airborne Doppler lidar simulator

(ADLS) allowing for simulation of various systems setups and retrieval settings is presented. Using the ADLS the measurement and retrieval process can be replicated, however, with the advantage of knowing the atmospheric input data in the sampling volume exactly. The error observed in the ADLS between input and retrieved wind profile is then directly traceable to wind field inhomogeneities and its magnitude can be evaluated against other sources of error (e.g. random radial velocity fluctuations and beam pointing direction inaccuracy).

Consequently, the research questions addressed are as follows:

1. Is the violation of the homogeneity assumption relevant for airborne Doppler wind profiling?

2. Is it possible to determine optimal system setups and retrieval settings under inhomogeneous conditions?

3. Can the violation of the homogeneity assumption and collinearity in model geometry be reliably detected from the measurements? In other words, are the coefficient of determination and condition number valid quality filtering measures

for wind profiling quality?

The outline of this paper is as follows: In the second section the ADLS framework is introduced, in the third section the airborne Doppler wind profiling theory is provided and in the fourth section airborne wind profiling error is evaluated for various system setups and retrieval settings.

## 2   Airborne Doppler lidar simulator

Extending previous studies, this work investigates the impact of wind field inhomogeneities on airborne wind profiling at highest resolution (O(1 km)) using an LES-based airborne Doppler lidar simulator. The description of the ADLS consists of four sections outlining and mimicking the ADLS structure and operation. First the underlying wind field options are specified, then the simulation of the airborne Doppler lidar system components is discussed and the measurement procedure is explained. Last, the wind profile retrieval can be performed on the simulated measurement data. The wind profiles are then evaluated

against the original wind field data supplied as input.

### 2.1   Atmosphere - Wind field

The simulator is tested with a set of underlying wind fields. In order to be as close as possible to a realistic measurement environment this study utilizes LES generated wind fields (WND). The LES fields are simulated using the parallelized LES model (PALM) and provided by the University of Hanover (Raasch and Schröter, 2001). The LES employs a grid spacing of 10 m (cor-

responding to a resolution finer than O(100 m)) and a domain size of 5 x 5 x 1.8 km, data output started with fully developed





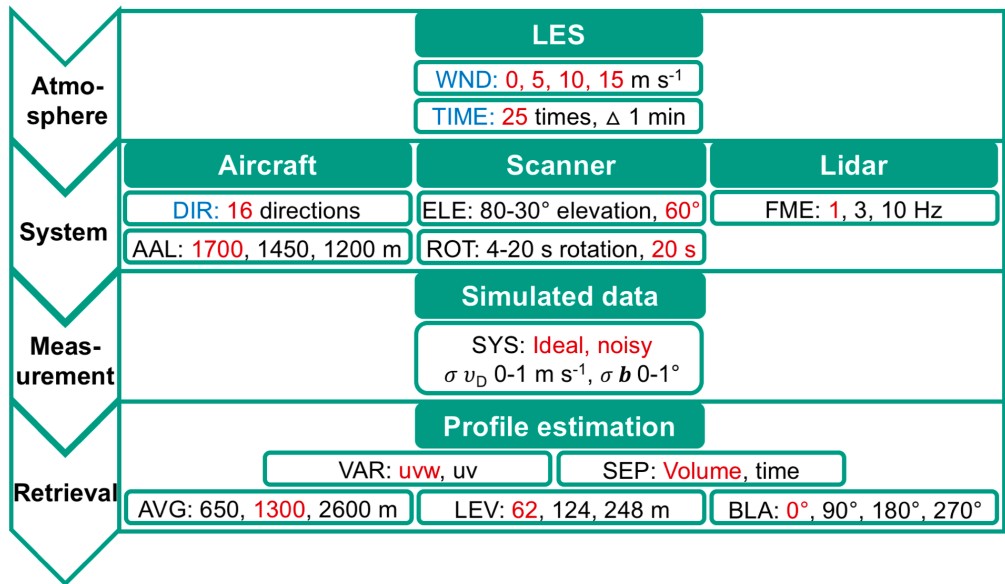

**Figure 1.** Simulator operation scheme and settings for the wind profiling quality analysis. System characteristics and retrieval options are varied according to the specifications given in Sec. 2.2 and Sec. 3. The corresponding abbreviations can be found in Tab. A1. Standard values used for system setup and wind profile retrieval are marked in red. Parameters which are always varied to generate statistical dataset are marked in blue.

turbulence after a spin-up time of 1 h. The LES is driven with a geostrophic background wind of $u_G = 0, 5, 10$ and $15$ m s$^{-1}$ and a constant kinematic surface heat flux $\overline{w'\Theta'} = 0.03$ K m s$^{-1}$ ($\overline{w'\Theta'} = 0.23$ K m s$^{-1}$ for the $0$ m s$^{-1}$ background wind case). The LES wind fields are the same as used by Stawiarski et al. (2013), and so a detailed description is available in Stawiarski (2014)[chap. 5.1] and Stawiarski et al. (2015). The convective situation is classified as unstable stratification with

the stability criteria pointing to the development of organized convective structures for $u_G > 0$ m s$^{-1}$. The boundary layer height is approximately 600 m (1200 m for the $0$ m s$^{-1}$ background wind case), with the entrainment zone extending from $600 - 1000$ m and gravity waves being present above. The stream-wise integral length scales for u and v are in the range of $200 - 500$ m, whereas the span-wise integral length scales are much smaller in the range $50 - 200$ m (Stawiarski et al., 2015). Due to the in-depth description in Stawiarski et al. (2015) the used LES data is not detailed further here.

The virtual aircraft transects the LES 25 times at 1-minute temporal spacing, as a singular flight through the LES does not yield sufficient statistics. At the given turbulence characteristics, flight directions and advection speed, the temporal and horizontal spacing is sufficient for the sampled air volume and turbulent structures to be different from each other during each transect.

The ADLS processes time-varying LES wind fields and the input can be adapted to other LES models easily. As an alternative
to the time-varying LES wind fields, a fixed LES time step could be utilized because the virtual aircraft and scanner move





through the LES data. This corresponds to the assumption of frozen turbulence which is reasonable for an airborne system. Both approaches were tested as a part of this study, yielding very similar results. In order to be as close as possible to reality, albeit requiring higher computational cost, only results obtained from the time-varying wind field are presented in this study.

## 2.2 System - Airborne Doppler lidar components

In the second section all main system components consisting of aircraft, scanner and lidar are simulated geometrically. An illustrative overview of the results obtained after system simulation is shown in fig. B1.

### 2.2.1 Aircraft

The aircraft trajectory is specified by the coordinates of the desired start and end points together with a prescribed aircraft speed relative to air. Curvilinear trajectories are also possible with a specified turn time.

To evaluate the effect of the aircraft motion relative to the wind, the aircraft transects the LES volume in 16 directions, with a heading difference of $22.5°$ between each (DIR). As the different flight tracks lead to different air mass volumes being sampled, the different flight directions are independent and can be used to increase the sample size of the statistical analysis. Consequently, the flight directions are varied for all investigations to generate a greater number of wind profiles for statistical analysis.

Another parameter varied is the aircraft flight altitude (AAL). Here, a standard value of 1700 m is chosen, representative for an unpressurized turboprop aircraft flying above the boundary layer under visual flight rules. The effect of lowering the AAL to 1450 m and 1200 m is analyzed in this study. Lower flight levels yield a smaller footprint of the measurement, however, the number of retrieved wind profile points is decreased as less atmosphere is covered vertically.

   The aircraft speed relative to air (IAS) is set to $65 \text{ m s}^{-1}$, representative for a medium-range turboprop aircraft at measure-
ment speed.

   Depending on whether a frozen-in-time or time varying wind field is chosen, the aircraft trajectory and sampling positions inside the LES must be calculated differently because the aircraft moves relative to the air mass and not the ground, which results in a difference between aircraft heading and ground track (Lee et al., 1994). Therefore, depending on whether the LES coordinate system coincides with the ground or the air mass, different sampling distances have to be applied in the LES. To
our knowledge, this manuscript is the first presentation of a correct airborne sampling simulation for LES studies, therefore it is explained in-depth in the following.

   For the frozen-in-time wind field, the LES coordinate system coincides with that of the air mass. The air mass and turbulent elements contained within are not advected through the domain during the measurement process. Thereby, the sampling is done at equidistant intervals in LES space along the flight trajectory. The spacing of the sampling points is calculated through
the simple formula $s = IAS \cdot t$. Consequently, for a given sampling time and aircraft speed an equal volume of air mass (and turbulence) is sampled as is done by a real aircraft. However, in this case the aircraft motion due to the wind speed needs to be factored into the retrieval at a later point using a triangle of velocities calculation (Appendix A1), because the aircraft track with respect to the ground is influenced by the wind speed. To illustrate the concept, albeit being unrealistic, consider





an aircraft flying at $65\,\mathrm{m\,s^{-1}}$, either up- or downwind, with a LES wind speed of $65\,\mathrm{m\,s^{-1}}$. The frozen-in-time wind field sampling distance in LES space will be the same, but compared to the ground, the aircraft will have moved a large distance in the first case and not at all in the second case.

For the time-varying wind field, the aircraft still moves relative to the air mass but now the LES coordinate system coincides with the ground coordinate system. In this case, the motion of the aircraft due to the air mass motion has to be considered during each time step to yield the correct measurement positions in the LES. Therefore, the aircraft heading and ground speed are determined iteratively for subsequent time steps, also based on the triangle of velocities (Appendix A1). The effect can again be imagined with an aircraft flying at $65\,\mathrm{m\,s^{-1}}$ into $65\,\mathrm{m\,s^{-1}}$ headwind. Relative to the LES coordinate system this aircraft will stay in the same place. It will thereby sample at the same geographic coordinate in the LES at all times. Summarizing, the correct simulation of the aircraft motion is important as it changes the sampling of the wind field data by the lidar system. In the ADLS, the aircraft track development is dependent on the wind field. Additionally, pitch, yaw and roll moments can be added to the aircraft position to simulate the effect of aircraft accelerations on the measurement process. These are termed aircraft movement (see Appendix Tab. A2 for an overview of general terminology) in this study and superimposed artificially and thereby independent of the track development. This independence is not realistic but deemed sufficient, as aircraft position correction maneuvers should generally not alter the track development (and thereby sampling) significantly, as they will cancel out over short periods of time. Thereby, the effect of aircraft movement due to flight maneuvers can be emulated as well.

### 2.2.2 Scanner

The scanner movement is simulated with freely selectable scanning geometries and includes an option to correct for the aircraft movement. Subsequent scan positions are calculated in an aircraft relative coordinate system by specifying the position which should be reached by the scanner, a time needed for the scan movement and a scan mode. Five scan modes are available. The scanner can exhibit stare-mode, scan with constant azimuth, constant elevation, along the shortest possible distance on a sphere between two positions or focus on specific positions in the ground reference system (thereby also correcting for aircraft motion). When the aircraft movement correction is enabled, scanner positions are calculated in an earth relative coordinate system and then transferred back to the respective aircraft relative coordinate system positions corrected for aircraft movement. The standard scan pattern (ELE, ROT) used for comparison is based on the scan geometry of existing systems (Weissmann et al., 2005; De Wekker et al., 2012). It consist of a 20-second full circle scan time (ROT, scan speed $18°\,\mathrm{s^{-1}}$) at $60°$ elevation (ELE, $30°$ off-nadir). For optimization, both elevation and speed of the scanner are varied (Sec. 4.2).

Two scanner system simulations are performed as a part of this study, an 'ideal' system without noise and a 'noisy' system with emulated scanner pointing direction noise. For the first setup, in order to focus on the effect of atmospheric inhomogeneities, an ideal scanner system is assumed without any beam pointing inaccuracy (so-called 'ideal' system in the following). For the second setup, in order to simulate an imperfect measurement system, a random Gaussian inaccuracy can be added to the scanner azimuth and elevation angles before being used for wind profile retrieval (details are explained in Sec. 4.1).





### 2.2.3 Lidar

In this study, the simulated lidar is based on the specifications of a Lockheed Martin Coherent Technologies WTX WindTracer system. Lidar systems with similar characteristics are often used in airborne Doppler lidar studies (Weissmann et al., 2005; De Wekker et al., 2012; Chouza et al., 2015; Witschas et al., 2017; Zhang et al., 2018). The lidar beam is emulated in accordance

with Stawiarski et al. (2013) and their range gate weighting function based on a pulse width of $\sigma_\tau = 3 \cdot 10^{-7}$ s is applied. The same cut-off value, 20 % of the maximum value of the weighting function, is chosen for calculating the effective length of the range gates for the averaging process. Variable range gate lengths and spacings can be specified, in accordance with the standard WTX settings a range gate length of $l_{RG} = 72$ m with a spacing of $\Delta x_{RGC} = 72$ m, starting at a distance of 400 m from the lidar is used for this study. For wind profiling, the standard lidar measurement frequency (FME) used in this study is

1 Hz. The details of how the volume scanned by the laser beam is constructed for averaging during the measurement process are explained in the next section.

Again, an 'ideal' and a noisy system are defined but this time for the laser. Two laser system simulations are performed as a part of this study, an 'ideal' system without noise and a 'noisy' system with emulated measurement noise. For the first setup, in order to focus on the effect of atmospheric inhomogeneities, an ideal laser system is assumed without any random radial

velocity fluctuations. However, for coherent systems, random radial velocity fluctuations (uncorrelated noise) due to detector noise, speckle effects and turbulence within the measurement volume can occur as part of the measurement process (Frehlich, 1997, 2001). Random radial velocity fluctuations in the high-signal return regime can be approximated by a Gaussian distribution with zero mean and standard deviation $\sigma v_D$. Therefore, in the ADLS, a Gaussian noise with standard deviation $\sigma v_D$ can be added to the simulated measured radial velocities. This inclusion allows for investigation of the effect of random radial

velocity fluctuations on the quality of wind profiles (see Sec. 4.1). For the lidar system simulated here, based on previous studies, the random noise is expected to be below 0.25 m s$^{-1}$ for 1 Hz measurement frequency, 0.5 m s$^{-1}$ for 5 Hz measurement frequency and 1 m s$^{-1}$ for 10 Hz measurement frequency (all at 750 Hz pulse repetition frequency) (Frehlich, 2001; Stawiarski et al., 2013).

### 2.3 Measurement procedure - motion combination and correction

For the measurement, the aircraft and atmospheric motion vectors need to be combined to give the measured radial Doppler velocity. Additionally, in order to obtain the atmospheric motion vector from the LES, the range gate positions need to be calculated from the state of the system components. Conveniently, this calculation is achieved by defining two separate coordinate systems (fig. 2), following the theory outlined in Lee et al. (1994) and Leon and Vali (1998). The LES wind fields are based in earthbound coordinate system (superscript E) oriented east-north-up (ENU). The aircraft coordinate system (super-

script A) is oriented along aircraft-right wing-down (ARD). The scanner is emulated and the measurement is performed in the aircraft coordinate system. To transfer between the two systems, coordinate transformations are required, the details are given in Appendix A2.





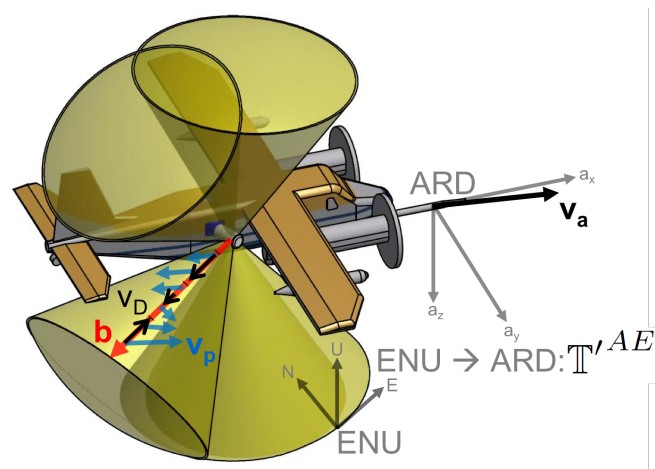

**Figure 2.** Schematic depiction of a possible airborne Doppler lidar system. The sketch is based on an upcoming system for the Dornier 128-6 aircraft of the TU Brunswick (D-IBUF, Corsmeier et al. (2001)). The lidar is inside the aircraft pointing outward, with a scanner mounted on the side of the fuselage directing the beam in the atmosphere. Displayed are the aircraft, scan cone surfaces as well as the coordinate systems and vectors used in eq. 3, for additional information see Sec. 2.3.

The measured Doppler velocity can be calculated through projecting the velocity vectors onto the lidar beam vector. The radial Doppler velocity measured by the lidar has a contribution from the lidar motion vector $\boldsymbol{v}_L^A$ and the atmospheric motion vector $\boldsymbol{v}_p^A$, both projected onto the beam direction vector $\boldsymbol{b}$ which is defined as a unit vector:

$$v_D = \boldsymbol{b}^A \cdot \left( -\boldsymbol{v}_p^A + \boldsymbol{v}_L^A \right). \tag{1}$$

The lidar motion can be split into two contributions, the aircraft motion and an aircraft movement moment arm due to aircraft rotation $\omega$ with the moment arm $\boldsymbol{r}$ (specifying the distance between the aircraft center of gravity and the lidar position),

$$v_D = \boldsymbol{b}^A \cdot \left( -\boldsymbol{v}_p^A + \boldsymbol{v}_a^A + \boldsymbol{\omega}^A \times \boldsymbol{r}^A \right). \tag{2}$$

Depending on the lidar beam direction, an airborne Doppler lidar system measures foremost the aircraft speed as it presents a vector with very large magnitude compared to the atmospheric motion vector and the aircraft movement moment arm. The

10 atmospheric motion vector needs to be transformed into the aircraft coordinate system where the measurement is performed using the rotation matrix $\mathbb{T}'^{AE}$ (see App. A2). The same is true for the aircraft motion vector which is originally calculated with respect to the LES coordinate system. Consequently, the measurement is achieved according to the following formula:

$$v_D = \boldsymbol{b}^A \cdot \left( -\mathbb{T}'^{AE} \boldsymbol{v}_p^E + \mathbb{T}'^{AE} \boldsymbol{v}_a^E + \boldsymbol{\omega}^A \times \boldsymbol{r}^A \right). \tag{3}$$

    Appendix A3 details how the weighted and averaged particle velocity $\boldsymbol{v}_p^E$ is obtained from the LES considering the range gate

position and motion during the measurement process.





Before the application of the retrieval, the contribution of the aircraft motion and aircraft movement moment arm to the measured radial velocity is then removed again using Eq. 3 to restore the original atmospheric contribution.

The ADLS makes it possible to add inaccuracies in any of the components relevant in the measurement or motion correction process. However, please note that throughout this study no system inaccuracies are introduced in the measurement or motion correction process for the ideal system simulation in order to focus on problems in wind profiling due to atmospheric inhomogeneity. In addition, it is assumed that for the real system the optical alignment and beam direction can be reliably quality controlled using a beam calibration procedure based on ground returns. The theory and procedure are outlined in Haimov and Rodi (2013). Consequently, for the ideal ADLS system, the motion-corrected Doppler velocity due to the particle velocity is equal to the particle velocity projection itself again. Therefore, compared to a ground-based system, only the measurement geometry is altered due to the moving system. The above discussed rotations and transformations do not influence the accuracy of the motion correction. As a result, for the ideal system simulation, all wind profiling errors discussed in the next sections stem purely from the inhomogeneous atmospheric conditions. In a homogeneous wind field, the retrieved wind profile is exact.

In real measurements, system noise is expected to occur. Therefore a worst-case scenario simulation with emulated system noise is conducted besides the ideal system simulation to estimate the impact of system noise (see Sec. 2.2.2 and 2.2.3). For the noisy system simulation, two sources of system noise are added, both on the measured radial velocity $v_D$ as well as the beam pointing direction $b^A$. Emulating Doppler lidar noise, a Gaussian noise contribution with measurement frequency dependent $\sigma v_D$ is added to the measured radial velocity. This addition is done before removal of the lidar motion contribution. Emulating an imperfectly known measurement system state, a Gaussian noise contribution is also added to the beam pointing direction, both on azimuth and elevation angles, thereby introducing wind profiling errors due to imperfect motion correction.

## 2.4 Retrieval - nadir as an example application

An illustrative example of the ADLS capability is given by simulated nadir retrievals. In fig. 3 the simulated vertical wind measurement is compared to the LES input along two transects for an ideal system with a measurement frequency of FME = 10 Hz.

The first transect, a crosswind flight (fig. 3 a,b), reveals the spanwise boundary turbulence structure for the 10 m s$^{-1}$ background wind case. Along stream organization of turbulence into rolls occurs and gravity waves are present above the boundary layer. The ADLS results show that the simulated lidar is able to capture turbulent structures in the boundary layer, although smoothing occurs for the finest scales. Vertically, the resolution of the measurement is defined by the range gate length, whereas horizontally it is defined through the combination of aircraft speed and measurement frequency. Another noticeable effect in the measurement is the movement correction capability of the scanner, which is disabled for the first half of the transect and enabled for the second half. During the first half, the lidar beam is not pointing exactly nadir due to the superimposed, artificial aircraft movement. This deviation causes a portion of the horizontal wind to be projected into the measurement. For the crosswind case, this effect is caused by the (rather strong) roll oscillations of the aircraft. Consequently, the measured vertical wind shows some additional superimposed structures compared with the LES for the first half of the transect. This contribution from the horizontal wind could be removed, as a second step, if the horizontal wind profile is known (Chouza et al., 2016b).

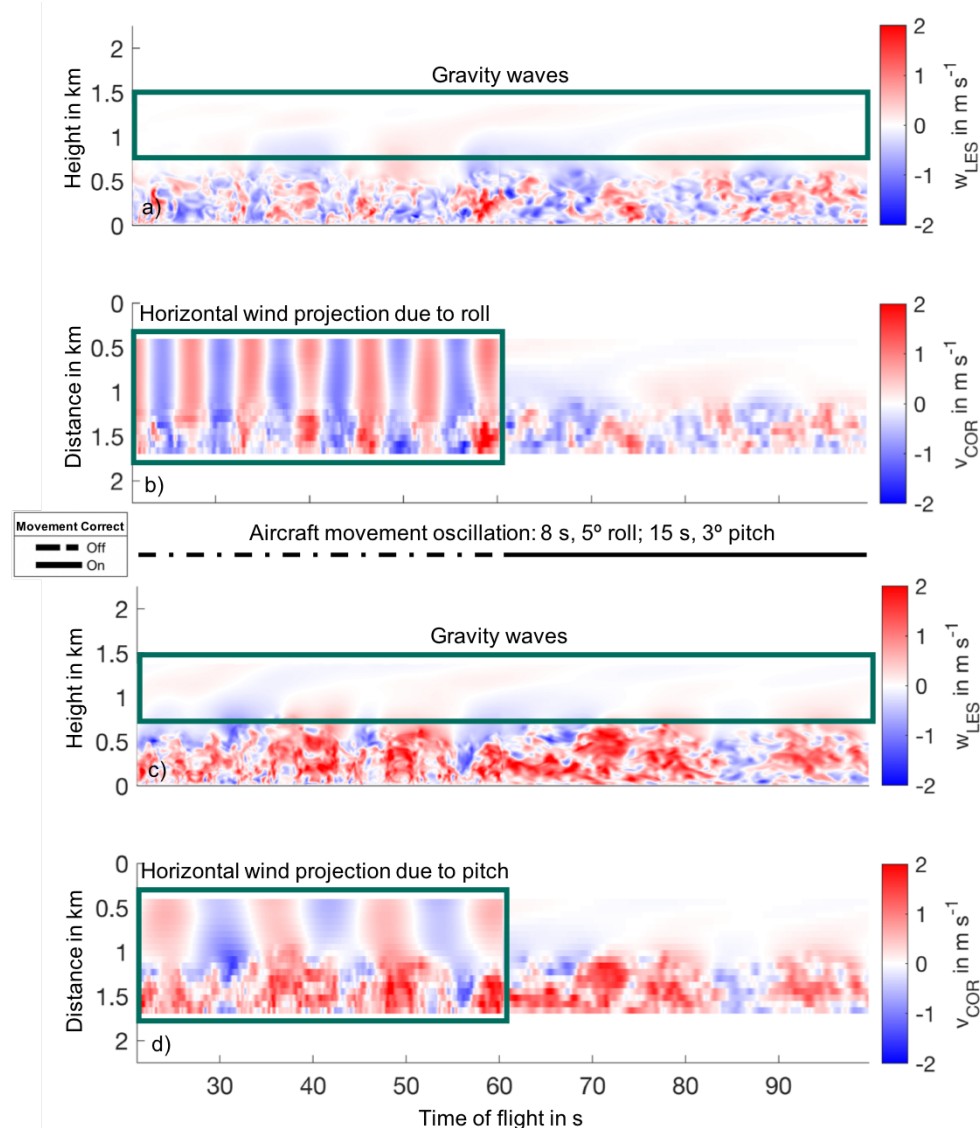

**Figure 3.** Nadir transects through LES for $10\,\mathrm{m\,s^{-1}}$ background wind case with superimposed aircraft oscillation movement at 10 Hz measurement frequency. a) LES vertical wind along a crosswind transect. b) ADLS motion corrected velocity measurement along a crosswind transect. c) LES vertical wind along an upwind transect. d) ADLS motion corrected velocity measurement along an upwind transect. Scanner movement correction is disabled for the first half of the transects, whereas it is enabled for the second half.

For the second half the scanner movement compensation is enabled. Consequently, the beam is pointing nadir at all times and no horizontal wind contribution is visible in the measurements.



The second transect, an upwind flight (fig. 3 c,d), reveals the streamwise boundary turbulence structure for the $10\,\mathrm{m\,s^{-1}}$ background wind case. The flight is conducted in an updraft region of the along-stream organized boundary layer convection, therefore positive vertical wind speeds dominate. In this case, the superimposed pitch movement contaminates the retrieved vertical wind measurement with a contribution from the horizontal wind if the scanner movement correction is disabled. In real measurements, the degree of this beam offset due to pitch oscillations is detectable in the ground return velocity which is non-zero in this case. Thereby, nadir-pointing measurements provide a good opportunity to check the accuracy of the scanner movement compensation and beam direction accuracy using the vertical wind measurement and ground return velocity. Should the movement compensation or pointing accuracy not be satisfactory, it is still possible to calibrate them using the ground return velocity (Haimov and Rodi, 2013). Using this information, the amount of horizontal wind mapped into the vertical wind can again be removed in a post-processing step if the vertical profile of the horizontal wind is known. In order for this method to yield reliable results, an accurate horizontal wind speed estimation is necessary. For this reason, the reliability of wind profiling measurements is the focus of this study.

## 3 Airborne Doppler wind profiling

Doppler wind profiling presents an inverse problem. The system state which produced the observations is inferred from a set of observations and an underlying model. The observations are the measured Doppler velocities. The model is given by the beam geometry, however, limitations arise because only a limited number of measurements, amount of beam directions and viewing geometries are possible. Choosing the beam geometry is an important step that can be altered depending on the desired measurement. On the one hand, it is desirable to maximize the spread among the beam pointing directions to reduce collinearity between the observations. On the other hand, the measurement volume should be minimized in order to have a small and localized footprint of the retrieved wind profile.

The system state is the original wind field on which the observations are performed, which is unknown in real measurements but known for the ADLS. To enable a robust estimation the wind field is usually assumed to be homogeneous, while knowing that this assumption is rarely fulfilled, especially in the boundary layer. Consequently, errors are introduced in the retrieved wind profile compared to the original wind field. This section describes the theory of the inversion process as well as errors associated with it (for a description of the terminology used for different errors see Tab. A3).

### 3.1 Wind profiling theory

Throughout this study, the approach outlined by Leon and Vali (1998) for wind profile retrieval (termed 'AVAD') is followed. In this method, multiple radial velocity measurements $v_{D_n}$ under different beam pointing directions $\boldsymbol{b}_n$ are sampled from an





atmospheric wind field with mean wind vector $\boldsymbol{v}_p$:

$$
\begin{bmatrix} v_{D_1} \\ v_{D_2} \\ v_{D_3} \\ \vdots \\ v_{D_n} \end{bmatrix} = \begin{bmatrix} b_{x_1} & b_{y_1} & b_{z_1} \\ b_{x_2} & b_{y_1} & b_{z_1} \\ b_{x_3} & b_{y_1} & b_{z_1} \\ \vdots & \vdots & \vdots \\ b_{x_n} & b_{y_n} & b_{z_n} \end{bmatrix} \begin{bmatrix} v_{p_x} \\ v_{p_y} \\ v_{p_z} \end{bmatrix}. \tag{4}
$$

The beam directions from multiple measurements make up the model matrix $\mathbb{G}$. Consequently, the relation can be expressed in the following way:

$$
v_D = \mathbb{G}\boldsymbol{v}_p. \tag{5}
$$

Knowing $v_D$ and $\mathbb{G}$, the inverse problem is then solved by calculating the inverse $\mathbb{G}^{-g}$ of the beam pointing model $\mathbb{G}$, in order to obtain an estimate of the wind vector $\boldsymbol{v}_{ret}$ responsible for the observations:

$$
\boldsymbol{v}_{ret} = \mathbb{G}^{-g} v_D. \tag{6}
$$

The general inverse $\mathbb{G}^{-g}$ of the overdetermined least-squares problem can be calculated as (Menke, 2012):

$$
\mathbb{G}^{-g} = [\mathbb{G}^T \mathbb{G}]^{-1} \mathbb{G}^T. \tag{7}
$$

Equation (7) presents a least-squares solution to the problem. Instead of calculating the general inverse in the above way, in this study a singular value decomposition (SVD) is performed, which yields benefits compared to the above direct solution (Boccippio, 1995):

$$
\mathbb{G} = \mathbb{U}\mathbb{S}\mathbb{W}^T, \tag{8}
$$

$$
\mathbb{G}^{-g} = \mathbb{W}_p \mathbb{S}_p^{-1} \mathbb{U}_p^T. \tag{9}
$$

Assessing the reliability of the parameters retrieved through the inversion process is a common problem in inverse theory.

The first often used metric deals with the stability of the inversion process. For the SVD the stability is expressed through the condition number (CN). The CN provides a measure of the spread of the model space, thereby diagnosing collinearity (Boccippio, 1995; Leon and Vali, 1998; Shenghui et al., 2014; Päschke et al., 2015; Wang et al., 2015). The condition number is defined as the ratio of the largest to the smallest diagonal entry of $\mathbb{S}$, which are the singular values of $\mathbb{G}$:

$$
CN = \frac{max(\lambda_S)}{min(\lambda_S)}. \tag{10}
$$

A high CN indicates high collinearity in the model geometry. Consequently, the real system state is not well explored in at least one direction and as a result the inferred system state is prone to a large error amplification (Boccippio, 1995). The CN is often used as a quality filtering criteria to exclude measurements where the spread of the beam pointing directions is not sufficient to explore the wind field adequately (Boccippio, 1995; Päschke et al., 2015; Wang et al., 2015).



Another measure commonly used to detect violations of the homogeneity assumption is the coefficient of determination ($R^2$) which can be obtained from the LSQ-solution. Using the estimated wind vector, an average radial velocity (LSQ-fit) is constructed by projecting it into radial velocities using the model geometry:

$$v_D^{ret} = \mathbb{G} \boldsymbol{v}^{ret}. \tag{11}$$

5   This average radial velocity differs from the measured radial velocities due to wind field inhomogeneities smaller than the measurement volume as well as model misspecification and measurement system noise (only in the noisy system case).

Using the average radial velocity, the $R^2$ is defined as

$$R^2 = 1 - \frac{\sum_i (v_{Di} - v_{Di}^{ret})^2}{\sum_i (v_{Di} - \sum_i v_{Di})^2}. \tag{12}$$

Wind field inhomogeneities smaller than the scan volume size decrease the $R^2$. Therefore, it is commonly assumed that the 10  $R^2$ captures the degree of violation of the homogeneity assumption. Often used filtering criteria are $R^2 > 0.8$ or $R^2 > 0.95$ for ground-based wind profiling (Wang et al., 2015; Päschke et al., 2015).

The reliability of both measures for wind profile quality assessment in airborne Doppler lidar scanning is investigated as a part of this study (Sec. 4.4).

## 4   Evaluation of error in wind profiling

15   Using the ADLS, the accuracy of the retrieved wind profiles can be analyzed under inhomogeneous wind field conditions. For the quantitative analysis, four metrics are employed, consisting of the root-mean-squared error (RMSE), the relative root-mean-squared error (REL), the number of available wind profile points (N) and the retrieval bias. The RMSE is given as:

$$\text{RMSE} = \sqrt{\left[ \frac{\Sigma_i^N (V_{m\,i}^T - V_{m\,i}^R)^2}{N} \right]}. \tag{13}$$

Here, $V_m^T$ is the true wind speed based on the input LES wind speeds, whereas $V_m^R$ is the ADLS retrieved wind speed from wind 20  profiling. The retrieval error is $\Delta V_m = V_m^R - V_m^T$. In this study the wind speed $V_m$ is calculated from all three components,

$$V_m = \sqrt{u^2 + v^2 + w^2}. \tag{14}$$

N is the number of wind profile points fulfilling the quality filtering criteria. The relative root-mean-squared error (REL) is used in accordance with Guimond et al. (2014):

$$\text{REL} = \sqrt{\left[ \frac{\Sigma_i^N (V_{m\,i}^T - V_{m\,i}^R)^2}{\Sigma_i^N (V_{m\,i}^T)^2} \right]}. \tag{15}$$

25   The REL can provide additional information to the RMSE as its magnitude is independent of the mean wind speed, thereby enabling comparisons between the different background wind cases, especially for higher wind speeds. The RMSE is driven



by two factors, the variance and bias of the retrieval errors. Therefore we also report the bias of the retrieval as its average mean deviation,

$$\text{bias} = \frac{1}{N} \sum_{i=1}^{N} (V_{m\ i}^{R} - V_{m\ i}^{T}). \tag{16}$$

To compare the simulated measurement, a model truth has to be defined. Here, the simulator offers more analysis insight

than in-situ measurements, where a representation error exists due to the different sampling volumes between lidar and in-situ measurements, e.g. instrumented towers. From real measurements, it is difficult to determine whether the observed wind profile differences are due to the difference in location and/or sampling volume (lidar scanned volume vs. instrumented tower) or due to violated model assumptions (homogeneous wind field). The ADLS can overcome the co-location problem, as the wind field contributions used to create the measurements are known exactly. Therefore, this study follows the approach described

in Wainwright et al. (2014) to define the model truth. In this, the model truth is the average over the points in the sampling volume touched by the lidar beam after weighting by the lidar weighting function (Sec. 2.2.3). In this study, the vector average is used as the wind speed averaging method. Vector averaging is more representative of the lidar measured wind speed than scalar averaging, as the lidar averages the wind field over a large area rather than summing up individual contributions without respect for directional change. The method used makes a difference especially for lower wind speed cases (see Sec. 4.4). The

advantage of the direct, equal-volume-based LES-lidar comparison, as noted by Wainwright et al. (2014), is that differences which occur between the simulated measurement and model truth are exclusively traceable to the wind field inhomogeneities. Therefore, optimization of the measurement system setup with respect to the impact of wind field inhomogeneities is possible. When an ideal system is assumed with no measurement system inaccuracy and no co-location problem, the results are also useful as they present a lower bound to the observable Doppler lidar error. Under the conditions investigated in this study, a

lower error cannot be observed from real in-situ comparisons (assuming equal statistical basis and comparable conditions), as the error given in this study specifies the error inherent due to the Doppler lidar viewing geometry and retrieval settings. Please note that despite investigating an overall random error, the error between vertically adjacent wind profile points is not necessarily random. This lack of statistical independence is because turbulent structures are correlated between adjacent wind profile points for an individual wind profile, making the profile appear smooth despite overall random error being present.

The simulation results are not directly transferable to the validation of Doppler lidar wind profile measurements through other measurement systems, as to-date all available measurement systems are unable to trace the exact volume touched by the lidar beam. An in-situ comparison should result in larger deviations than what is presented here due to additional co-locations problems, and due to the different wind vector measurement principles (Bradley et al., 2012). It seems worthwhile to extend the analysis in this direction at a later point, as the underlying statistical foundation exists already (Frehlich, 2001). The topic

is not addressed here as it is beyond the scope of this work and the inherent problems caused by the measurement geometry and viewing geometry are voluminous by themselves.

To avoid correlation among the analyzed data only one standard setting for system setup and retrieval setting is used for general analysis (see also Sec. 2 and fig. 1). Further, only wind profiles within the boundary layer and entrainment zone are compared as this study focuses on retrieval error under inhomogeneous conditions. Therefore, the maximum wind profile





altitude is set to 800 m (see also fig. B6). Only the two wind profiles closest to the center, contained fully within the LES, are analyzed to ensure full measurement coverage and avoid an artificially increased CN due to the lidar beam exiting the LES volume. The wind profiling start point is chosen randomly for each transect within one profiling length. The random start point prevents a systematic influence of the transect begin location on measurement quality for the different flight directions. It also

contributes to a further de-correlation of analyzed wind profiles.

The standard system setup (STP) used for analysis is based on the four different wind speed cases and 16 flight directions through the LES. For this setting, only data from the 20 s scan pattern at $60°$ elevation (ROT 20 s, ELE $60°$), 1 Hz measurement frequency (FME 1 Hz) and the 1700 m flight altitude (AAL 1700 m) is analyzed. The standard wind profile retrieval setting (RET) consists of u,v,w component retrieval (VAR u,v,w) using a volume-based profile separation (SEP Volume). Retrievals are

obtained from along-track averaging over $X_a = 1300$ m (AVG 1300 m) at an across-track averaging distance of $X_c = 1963$ m (corresponding to the maximum across-track distance covered by the lidar beam at $60°$ elevation and a flight altitude of 1700 m). The vertical wind profile resolution is chosen as 62 m (LEV 62 m), yielding one range gate within every vertical layer. In Sec. 4.2 each of the parameters are then varied individually to evaluate the effect of different system setups and profile retrieval settings.

The described system setup and retrieval setting allow for retrieval of 38400 wind profile points. These consist of 9600 wind profile points for each background wind case, yielding a sufficient statistical basis for evaluation. For quality filtering purposes a minimum coefficient of determination of $R^2_{min} = 0.90$ and a maximum condition number $\mathrm{CN}_{max} < 9$ are applied. Similar values have been recommended and used in the past by Boccippio (1995); Päschke et al. (2015); Wang et al. (2015). The effect of quality filtering on retrieval quality is investigated in detail in Sec. 4.4.

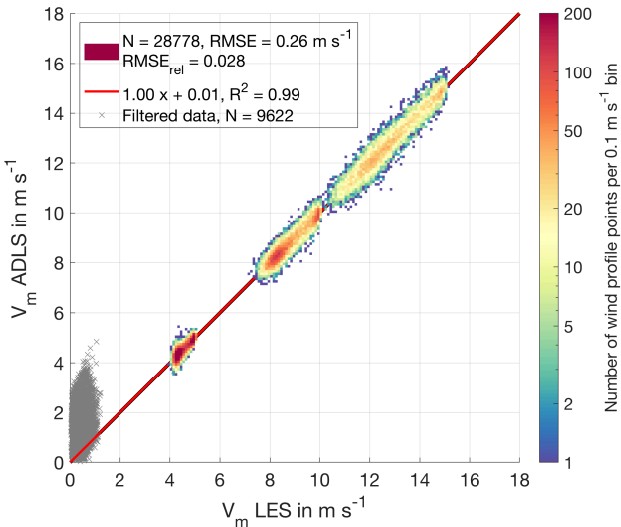

**Figure 4.** Comparison of LES truth and ADLS retrieved wind speed for an ideal measurement system using STP, RET with quality filtering criteria applied. Color-coded are all measurements which pass quality filtering, gray crosses are the ones which do not.



The quality of the standard system setup and retrieval settings can be evaluated from fig. 4. In this case the wind profile retrieval is made assuming an ideal measurement system, thereby all deviations are directly traceable to wind field inhomogeneities.

A total number of 28778 wind profile points pass quality filtering and the results provide a number of interesting observations. Firstly, the wind speed retrieval is unbiased provided that an appropriate system setup is used and appropriate quality filtering criteria are applied (discussed in Sec. 4.4). Overall retrieval quality is high with $R^2 = 0.99$ and $RMSE = 0.26$ m s$^{-1}$. Yet, even with strict quality filtering applied, deviations up to 1 m s$^{-1}$ can occur. The LES cases with higher wind speeds show higher wind variability and increased absolute wind speed retrieval error. With decreasing wind speed (separately for each of the cases), associated with measurements in the boundary layer under turbulent conditions, deviations increase but remain unbiased. Interestingly, the 9600 retrieved wind profile points for the 0 m s$^{-1}$ background wind case are completely excluded by the quality filtering criteria (as well as 22 from the 5 m s$^{-1}$ background wind case). They also show much higher wind speed retrieval scatter and can introduce a positive bias if inappropriate quality filtering criteria are applied, although the individual retrieved components themselves are unbiased (fig. B3). The reasons behind this are discussed in Sec. 4.4, where the effect of quality filtering on retrieval quality is thoroughly investigated.

The vertical distribution of wind profile error mirrors that of the responsible boundary layer turbulence (fig. B6). Errors are largest in the middle of the boundary layer where up- and downdrafts have maximum intensity. Towards the ground, a slight reduction in wind profiling error is observable for all wind speed cases. There, the size of the turbulent elements becomes smaller and consequently the lidar scan averages over more eddies, thereby better fulfilling the homogeneity assumption. This is a theoretical result, in real measurements other effects such as ground chirp will degrade near-ground retrieval of wind profiles (Godwin et al., 2012). Towards the top of the boundary layer, wind profile error decreases as the homogeneity assumption is better fulfilled. Nevertheless, entrainment and detrainment processes can still cause noticeable retrieval error, especially for higher wind speed cases. In the free, homogeneous atmosphere wind profile error vanishes.

The wind direction retrieval (fig. B2) is of similarly good quality compared to the wind speed retrieval. The wind profile points which pass quality filtering cluster in a small area around 270°, representing the westerly wind direction. The wind direction retrievals for the 0 m s$^{-1}$ background wind case scatter throughout the full range as the wind direction is not meaningful without a background wind speed. As before, these values are completely eliminated by quality filtering. Overall, the retrieval exhibits slightly degraded quality criteria (lower $R^2$, small y-axis intercept), but this behavior is due to the data not being distributed over a wide range as is the case for wind speed.

## 4.1 Influence of system noise on wind profiling quality

Using the ADLS the importance of different error sources for wind profiling accuracy can be evaluated. In this study, three error sources are considered. As a standard, the violation of the inhomogeneity assumption due to boundary layer turbulence is always present in the retrievals. Additionally, two error sources due to measurement system noise can be emulated:





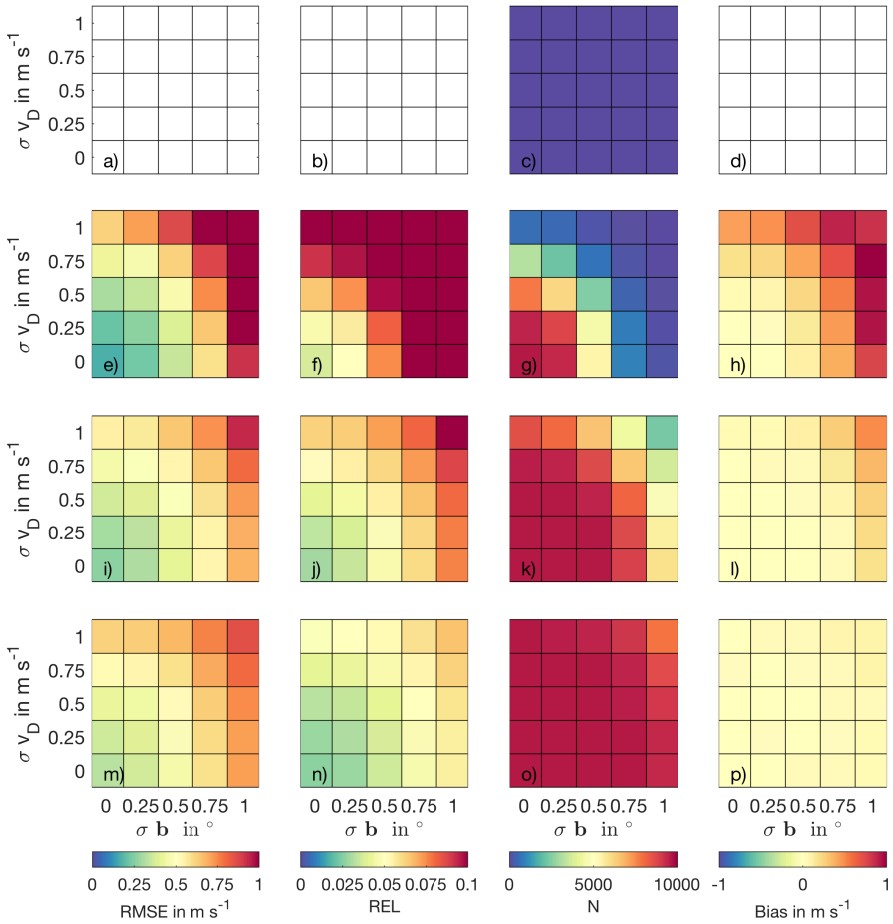

**Figure 5.** RMSE, REL, N and bias for four background wind cases at varying random radial velocity fluctuations and beam pointing inaccuracies. a), b), c), d) $0\,\mathrm{m\,s^{-1}}$ background wind case. e), f), g), h) $5\,\mathrm{m\,s^{-1}}$ background wind case. i), j), k), l) $10\,\mathrm{m\,s^{-1}}$ background wind case. m), n), o), p) $15\,\mathrm{m\,s^{-1}}$ background wind case.

1. The Doppler lidar radial velocity noise is emulated through addition of a Gaussian distributed random fluctuation to the measured radial velocity. The standard deviation of the Gaussian distribution is varied in a range of $\sigma v_D = 0 - 1\,\mathrm{m\,s^{-1}}$. This range of variation presents a worst case scenario, as stated before the WTX-system is expected to show a random radial velocity fluctuation below $0.25\,\mathrm{m\,s^{-1}}$ at 1 Hz measurement frequency.

2. A beam pointing inaccuracy is emulated through addition of Gaussian noise to the beam pointing direction. The emulation is achieved through adding Gaussian distributed random numbers to the beam azimuth and elevation before retrieval. The standard deviation of the Gaussian distribution is varied in a range of $\sigma \boldsymbol{b} = 0 - 1°$. This magnitude also presents a





worst-case scenario assumption, the achievable beam pointing direction accuracy is assumed to be much better and can be improved by using beam pointing calibration (Haimov and Rodi, 2013).

The results obtained for the three error sources are presented separately for each background wind case in fig. 5, using the standard system setup and retrieval strategy as above. As before, a clear influence of the background wind speed on retrieval error is detectable, therefore the error characteristics are discussed individually for the $0, 5, 10$ and $15$ m s$^{-1}$ case. For each background wind case, a maximum number of 9600 wind profile points is retrievable.

For the $0$ m s$^{-1}$ background wind case all wind profile points are eliminated by the quality filtering criteria (fig. 5 c). The responsible criterion is the $R^2$, as all volumes are sufficiently covered by the lidar beam, thereby making CN filtering not applicable. The raw, not quality filtered values exhibit an RMSE and bias in excess of $1$ m s$^{-1}$, highlighting the need for application of quality filtering criteria (fig. B5 a,d).

For the $5$ m s$^{-1}$ background wind case, retrieval error strongly increases if system noise levels exceed $\sigma v_D > 0.25$ m s$^{-1}$ or $\sigma \boldsymbol{b} > 0.25°$ (fig. 5 e,f). Alongside this result, the number of retrieved wind profile points decreases above these noise levels, as quality filtering eliminates more retrievals (fig. 5 g). At higher noise levels a bias develops due to quality filtering (fig. 5 h). This bias is an undesirable feature of the applied quality filtering, although only being present for a very small number of retrieved wind profile points. The unfiltered, raw wind profile retrieval is unbiased and actually shows slightly lower RMSE levels for the highest noise cases due to the non-existent bias (fig. B5 e,h).

For the $10$ m s$^{-1}$ background wind case, the increase of retrieval error with increasing system noise is less pronounced. Turbulence is the main driver of wind profiling error for noise levels up to $0.75$ m s$^{-1}$ for radial velocity noise or $0.75°$ beam pointing inaccuracy, if considered individually, or up to $0.5$ m s$^{-1}$ and $0.5°$, if both system noise sources are combined (fig. 5 i). Again, quality filtering eliminates wind profile points with high error levels, but also introduces a positive bias.

For the $15$ m s$^{-1}$ background wind case, turbulence remains the main driver of wind profiling error for all system noise levels considered, if varied individually (fig. 5 m). If both noise sources are combined, the highest noise level of $1$ m s$^{-1}$ and $1°$ shows a slightly degraded profiling quality and number of retrieved wind profile points.

Overall, the system noise analysis shows that for the expected system accuracy the largest error in wind profiling is caused by flow inhomogeneities. An exception are low wind speeds, when system accuracy can become important if high system noise levels are present. Further, quality filtering is necessary to eliminate unreliable wind speed retrievals. However, quality filtering has to be conducted with adequate thresholds as it can introduce a retrieval bias and thereby even increase retrieval error for low wind speeds (see also Sec. 4.4).

## 4.2 Influence of system setup on wind profiling quality

The results from the previous section emphasize the importance of turbulence as a driver of wind profiling error. Therefore, in order to show the influence of the system setup on wind profiling quality, two cases are considered in the following: The first case focuses purely on the wind profiling error due to the violation of the inhomogeneity assumption. This case is termed 'ideal system' as no system noise is included. The second case presents results for the combined system noise case due to both



radial velocity noise and beam pointing direction inaccuracy, termed 'noisy system'. Lowest level noise ($0.25$ m s$^{-1}$ for radial velocity noise and $0.25°$ for beam pointing inaccuracy) is introduced in order to show the tendencies of wind speed retrieval quality with respect to the different system noise sources. Under normal operating conditions a real system is expected to perform better than even the lowest system noise levels (De Wekker et al., 2012).

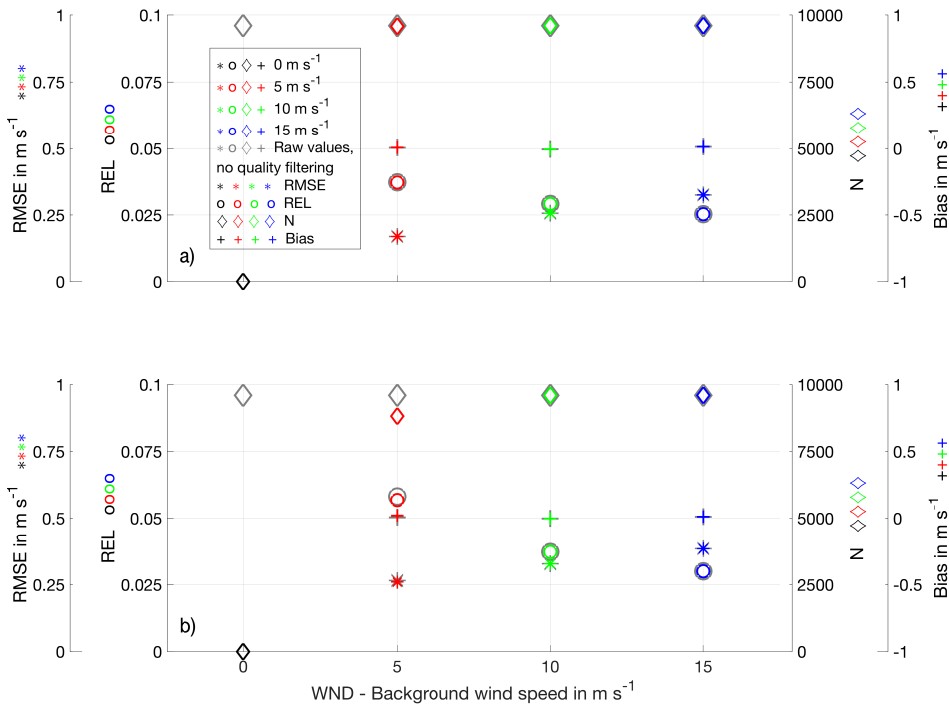

**Figure 6.** Wind profiling error for four different background wind cases WND $0, 5, 10, 15$ m s$^{-1}$ using STP, RET. a) RMSE, REL, N and bias for an ideal measurement system. b) RMSE, REL, N and bias for a noisy measurement system with radial velocity noise $\sigma v_D = 0.25$ m s$^{-1}$ and beam pointing inaccuracy $\sigma \boldsymbol{b} = 0.25°$.

The variation in retrieval quality for the different **background wind cases (WND)** is shown in fig. 6. As discussed before, the absolute RMSE increases with increasing background wind speed as the deviations become larger. The opposite behavior is observed for the REL. The decreasing REL is due to the decreasing ratio of the magnitude of the vertical wind to the horizontal wind. For lower wind speeds, relatively strong up- and downdrafts exist which cause error in the horizontal wind speed retrieval. As the horizontal wind is of smaller magnitude, any deviations due to the vertical wind have a stronger impact if measured in a relative way. For higher wind speeds, the vertical wind magnitude does not increase linearly with the horizontal wind magnitude. Therefore, higher wind speeds show a decreased REL. Consequently, although the RMSE increases further for higher wind speeds, the results suggest that the REL approaches a threshold level. The different background wind cases



thereby also represent varying turbulence intensity conditions, which have been shown to be of importance for Doppler lidar retrieval quality before (Banakh et al., 1995; Wang et al., 2016).

The underlying gray symbols indicate the respective quantities before application of quality filtering. Results show that for the 0 m s$^{-1}$ background wind case all 9600 theoretically available wind profile points are removed by $R^2$ filtering (gray vs.

black diamond). For the 5 m s$^{-1}$ background wind case only 22 wind profile points are filtered for the ideal system (gray vs. red diamond). For the 10 m s$^{-1}$ and 15 m s$^{-1}$ background wind cases no wind profile points are eliminated by quality filtering for the standard system setup and retrieval strategy as they exhibit a magnitude of turbulence that does not result in $R^2$ filtering. The adequately explored volumes make CN filtering not applicable for any wind speed case for the standard system setup and retrieval strategy. Because no wind profile points pass quality filtering for the 0 m s$^{-1}$ background wind case, no RMSE,

REL and bias are specified. However, the unfiltered values show that the RMSE and bias are much larger than for the higher background wind cases. An RMSE above 1 m s$^{-1}$, a REL above 0.1 and a bias above 1 m s$^{-1}$ are treated as unreliable wind profile retrievals in the following, therefore values above this error level are not included in the analysis. A reliable wind profile retrieval under lowest wind speed conditions is impossible to achieve using the standard system setup and retrieval setting as the radial velocity contribution of the vertical wind is dominant in this case. Consequently, an erroneous mapping of vertical

wind into horizontal wind results in a biased retrieval (further discussed in Sec. 4.4).

The noisy system results show that the assumed system inaccuracy contributes to the retrieval error, but turbulence still remains the main driver (fig. 6 b). A noticeable increase in RMSE and REL is observed for the 5 m s$^{-1}$ background wind case, while at higher wind speeds the increase is less. The retrieval remains bias-free. Further, a reduced number of retrieved wind profile points is evident under noisy conditions as a larger number of wind profile points are excluded by the $R^2$ filtering

criteria. This quality filtering, however, only leads to a marginal reduction of error levels.

**Flight direction (DIR)** also has an influence on wind profiling quality for two reasons: first, due to the aircraft flying relative to air (see Sec. 2.2), upwind flights contain more samples per volume than downwind flights. Second, due to the alignment of turbulence with mean flow, different flight directions can show a more or less severe violation of the homogeneity assumption. Both effects can be observed in the ADLS results (fig. 7). The results discussed here have to be interpreted with care, because

the sample size is reduced for the directional analysis (a maximum of 600 wind profile points is retrieved for each setup). In all other analysis the individual flight directions are combined to generate a reliable number of samples. Overall, and especially for the higher background wind cases, upwind profiling yields the lowest RMSE (DIR WSW, colored asterisks) as more points fall into the ground-based volume used for the retrieval. Down-wind profiling on the other hand shows slightly degraded results in comparison (DIR ESE, colored asterisks). The influence of alignment of turbulence with flow is also noticeable. In particular,

for the 15 m s$^{-1}$ background wind case, the flight direction W does not yield the best retrieval quality (lowest RMSE, blue asterisk), as would be expected based on pure sampling grounds. At the W direction, a stronger horizontal shear of the vertical velocity is encountered by the Doppler lidar, possibly due to along-flow alignment of rolls, leading to a higher error of the retrieved wind profiles. Consequently, the directions next to the straight upwind flight exhibit the best retrieval quality. Some directions show a small tendency for retrieval bias, which could be caused by the varying turbulence structure and alignment.

However, the reliability and magnitude of this bias cannot be determined with good confidence due to small sample sizes used



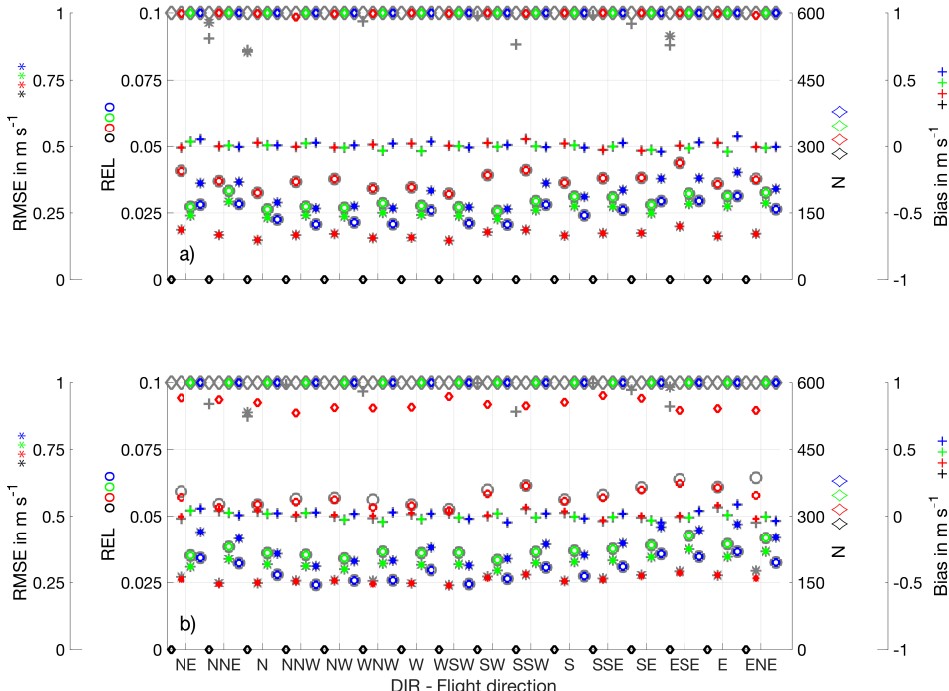

**Figure 7.** RMSE, REL, N and bias for 16 different flight directions using STP, RET. a) Ideal system. b) Noisy system. Color-coded points as in fig 6.

in this analysis. As discussed before, the addition of noise decreases the retrieval quality especially for the $5 \text{ m s}^{-1}$ background wind case. Nevertheless, the above described influence of flight direction on wind profiling error remains valid.

In the following, the results for the different flight directions are combined and then averaged to create a larger statistical dataset, as they show small systematic differences and do sample different air mass volumes.

5     The **aircraft flight altitude (AAL)** is a system parameter which can easily be varied in real measurements. At a lower flight altitude the across-track distance covered by the lidar beam is smaller. Thereby, the across-track averaging distance can be decreased, resulting in a smaller measurement footprint. However, a lower flight altitude also results in a smaller number of available wind profile points above the boundary layer, due to the decreased vertical coverage. The impact of aircraft flight altitude on retrieval quality is shown in fig. 8 for three flight altitudes which allow for full vertical boundary layer coverage.

10 For the conditions investigated here, lower flight altitudes result in a slightly increased RMSE for the $5, 10$ and $15 \text{ m s}^{-1}$ background wind cases (colored asterisks). The flight altitude can influence error levels as the ratio between the lidar scan volume and turbulent eddy size changes. Most likely, for the situation investigated here, at the lowest flight altitude the scan volume inside the boundary is closer to the turbulence dominating eddy size. Thereby, the average is performed over a smaller number of eddies, resulting in a less fulfilled homogeneity assumption due to the short averaging distance. The increase in



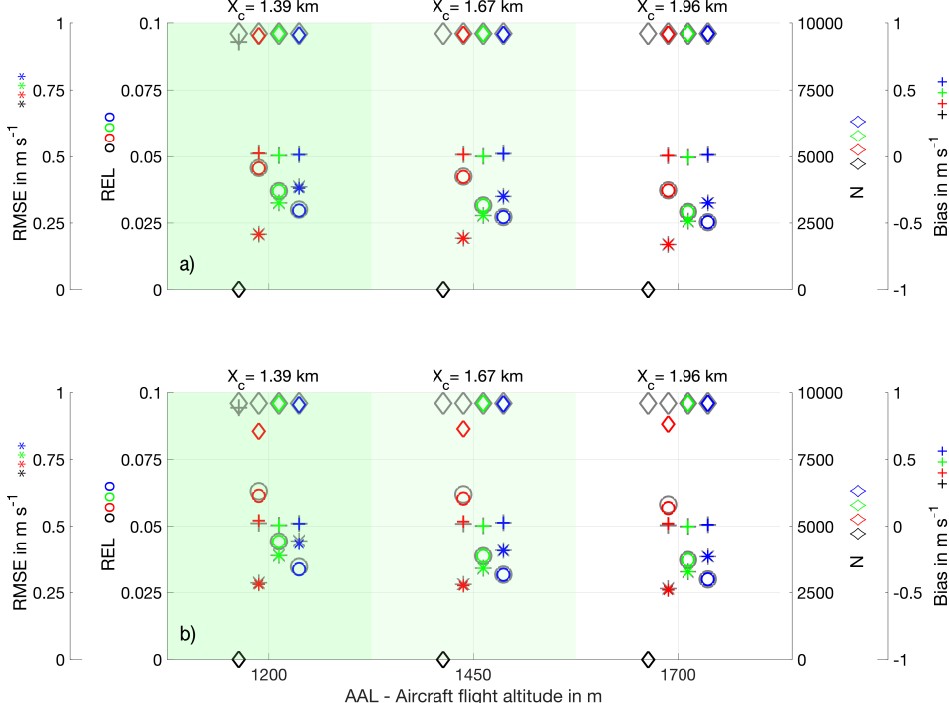

**Figure 8.** RMSE, REL, N and bias for three aircraft flight altitudes using STP, RET otherwise. a) Ideal system. b) Noisy system. Color-coded points as in fig 6. In the colored areas the across-track averaging distance $X_c$ is changed to the values specified above in order to adapt to the distance covered by the lidar beam.

retrieval error for lower flight altitudes is not severe and has the benefit of having a decreased measurement footprint. The retrieval also remains unbiased for all flight altitudes. As before, the introduction of system noise increases the wind profiling error. This increase is of similar magnitude for all settings.

The **scan elevation angle (ELE)** has a strong influence on wind profiling quality but also measurement footprint. To demon-
5  strate the influence of scan elevation on wind profiling quality, the scan elevation angles are varied between $30°$ (shallow) and $80°$ (steep), with $90°$ being nadir (fig. 9). For more shallow scan elevation angles the lidar beam covers a larger across-track distance. Consequently, the across-track averaging distance has to be adjusted, resulting in a larger measurement footprint. For steeper scan elevation angles the lidar beam covers a smaller across-track distance, resulting in a more confined measurement footprint. The along-track averaging distance is kept constant at the standard value of $1300$ m for all setups.
10  At steep scan elevation angles the vertical wind can exhibit greater influence. Consequently, at elevations steeper than $60°$, the wind profiling quality is degraded (higher RMSE, colored asterisks) and can only be partially improved through quality filtering. The degradation is not just due to a larger retrieval scatter, in addition the retrieval also becomes biased (colored plus signs) due to erroneous mapping of vertical wind into horizontal wind, enabled by its large radial velocity contribution. Quality





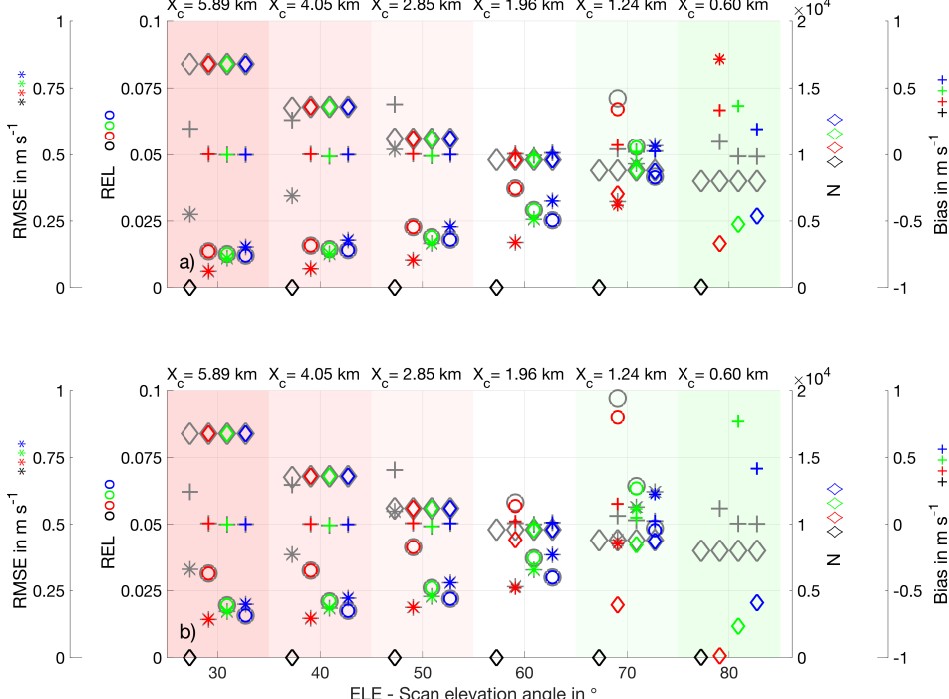

**Figure 9.** RMSE, REL, N and bias for six scan elevation angles using STP, RET otherwise. a) Ideal system. b) Noisy system. Color-coded points as in fig 6. In the colored areas the across-track averaging distance $X_c$ is changed to the values specified above in order to adapt to the distance covered by the lidar beam.

filtering can improve the wind profiling quality for steep scan elevations (gray vs. colored asterisks). However, it also worsens a preexisting wind profiling bias for the $5\,\mathrm{m\,s^{-1}}$ background wind case (gray plus sign vs. red plus sign). Further, quality filtering introduces a previously non-existent profiling bias for the $10\,\mathrm{m\,s^{-1}}$ and $15\,\mathrm{m\,s^{-1}}$ background wind cases (gray plus signs vs. green, blue plus sign). Additionally, for steeper elevation angles the number of usable wind profile points decreases

5 (colored diamonds). The reason is twofold: First, due to the increased projection of the range gate length onto the vertical, the maximum number of available wind profile points is lower. Second, quality filtering also decreases the number of usable wind profile points for scan elevation angles greater than $70°$. The degradation is more severe for the lower background wind cases, as fewer points fulfill the quality filtering criteria (in this case the filtering occurs due to $R^2$). In summary, for the standard retrieval strategy, steep scans with more than $70°$ elevation do not allow for wind profile retrieval with reasonable error margins

10 under turbulent conditions for all wind speeds, despite having smaller measurement footprints.

When using scans with elevation angles more shallow than the standard $60°$ elevation, an improved retrieval quality can be achieved at the cost of having a larger across-track averaging distance. On the one side, ADLS results show decreased error levels (colored asterisks) and an increased number of wind profile points being available (colored diamonds). The decrease





in error level is due to a smaller influence of the vertical wind on retrieval error. The increased number of retrieved wind profile points is due to the more shallow scan elevation, which leads to a smaller vertical projection of the range gate length. The additional points do not necessarily contain additional, independent information due to the vertical autocorrelation of turbulence and thereby wind profile error (discussed in Sec. 4.3), limiting the increase in vertical resolution. On the other side,

for more shallow scans, the footprint of the measurement starts to increase as a larger ground distance is covered by the lidar beam and the across-track averaging distance has to be increased. ADLS results have to interpreted with care for elevation angles smaller than $40°$ due to the limited along- and across-track extent of the LES domain. Once the across-track distance covered by the lidar beam approaches the LES volume, the beam exits the LES volume frequently for some flight directions and estimation of wind profiling quality becomes unreliable. Nevertheless, for scan elevations angles more shallow than $50°$, the

unfiltered $0 \text{ m s}^{-1}$ background wind case retrieval appears within the reliable retrieval limits (gray asterisks). Here, refraining from $R^2$ quality filtering would allow for retrieval of the lowest wind speeds, although with a biased retrieval.

The discussed wind profiling error characteristics are also present when introducing the system noise. The increase of profiling error and bias for steep elevation angles is more pronounced, and the number of measurements eliminated by quality filtering increases further. As before, the profiling error decreases for scans more shallow than $60°$ elevation. Further, the

slightly reduced number of retrieved wind profile points for the $5 \text{ m s}^{-1}$ background wind case is not present for scans with $< 50°$ elevation.

**Scan speed (ROT)** also has an impact on wind profiling quality (fig. 10). At slower scan speeds (longer rotation duration) the number of independent measurement points in the volume used to create the wind speed estimate is reduced due to autocorrelation among the measurements (the number of measurements is the same as measurement frequency is unchanged). Faster scan

speeds allow for greater azimuth diversity and measurements which are less correlated, thereby enabling an improved wind profiling quality, despite having an overall equal number of measurement points available in each volume. All scan speeds show bias-free results and achieve the maximum number of retrievable wind profile points. For fast scan speeds with rotation duration of $16 \text{ s}$ and below, the raw RMSE of the $0 \text{ m s}^{-1}$ background wind case becomes smaller than $1 \text{ m s}^{-1}$. Similarly, the bias is also reduced, but the applied quality filter still eliminates all retrievals. One important aspect which is not captured by

the ADLS is the influence of increasing scan speed on return signal quality and thereby Doppler speed measurement quality. A rapid scan movement leads to a strongly changing IAS contribution during the scan and sampling of more turbulent elements. If these rapid scans cause problems in Doppler velocity estimation due to spectral broadening or other effects, slower scan speeds become preferable again.

When system noise is considered, the wind profiling error reduction for fast scan speeds is lowered, as autocorrelation among

the individual measurements is reduced due to the noise. Thereby, increasing the scan speed does not yield the same magnitude of wind profiling quality improvement as it is the case without system noise. However, for the $5 \text{ m s}^{-1}$ background wind case, the slightly reduced number of retrieved wind profile points is corrected for by faster scan speeds. Further, raw retrievals for the $0 \text{ m s}^{-1}$ background wind case again show a decreasing RMSE and bias.

ADLS results show that an increased **lidar measurement frequency (FME)** does not improve wind profiling quality (fig.

11). This insensitivity exists because higher measurement frequencies produce measurements in between the lower frequency



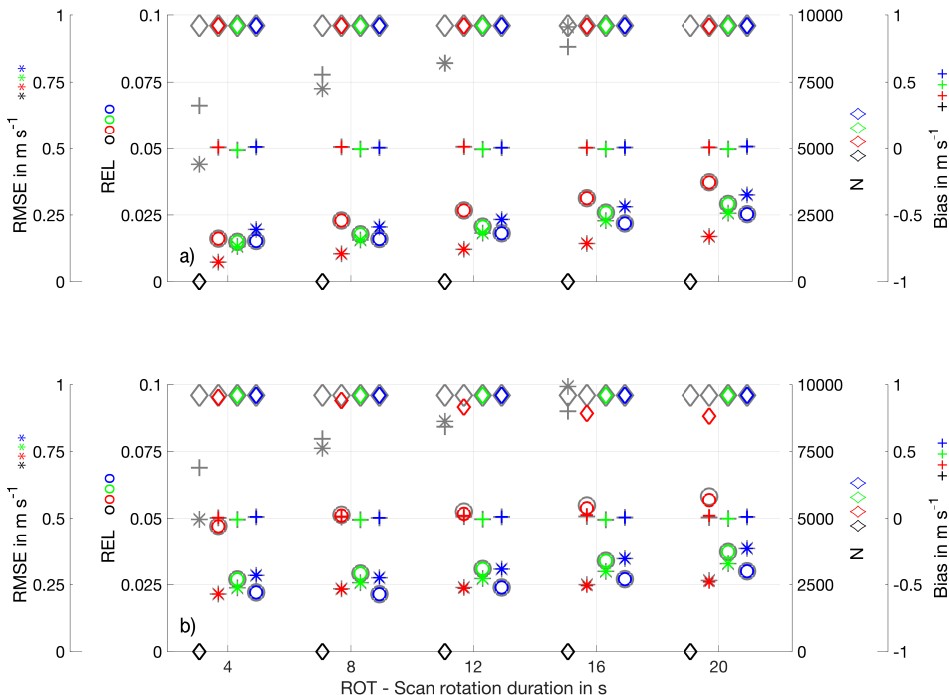

**Figure 10.** RMSE, REL, N and bias for five scan speeds using STP, RET otherwise. a) Ideal system. b) Noisy system. Color-coded points as in fig 6.

measurements which are highly autocorrelated. Due to the autocorrelation (resolving the turbulent structures in greater detail but not fulfilling the homogeneity assumption better) the additional measurements do not provide new information that can be used to improve the LSQ-fit. In fact, the $R^2$ is lowered for some wind profile points from the $5 \text{ m s}^{-1}$ background wind case, as turbulence is resolved in more detail. This $R^2$ reduction results in a filtering of some wind profile points, thereby the number of

5 retrieved wind profile points is slightly degraded for the ideal system case. The results of the measurement frequency analysis could be altered for a real system if the signal return quality is influenced by the signal accumulation time. On the one hand, if a longer pulse accumulation leads to a more reliably estimated Doppler velocity, lower measurement frequencies become preferable. On the other hand, if the change in radial velocity due to the scanner movement leads to problems in Doppler velocity estimation, higher measurement frequencies become preferable.

10 Adding artificial noise to the measured radial velocities in the ADLS leads to a loss of retrieved wind profile points for the lower background wind cases at high measurement frequencies. This behavior is caused by the severely increased radial velocity noise ($1 \text{ m s}^{-1}$ at $10$ Hz) and the frequency constant beam pointing inaccuracy, both leading to more filtering of wind profile points by the $R^2$ criterion. Due to the non-linear increase in random radial velocity noise (see Sec. 2.2.3) the $10$ Hz measurement frequency case even exhibits slightly elevated profiling error levels. The results discussed for the FME analysis



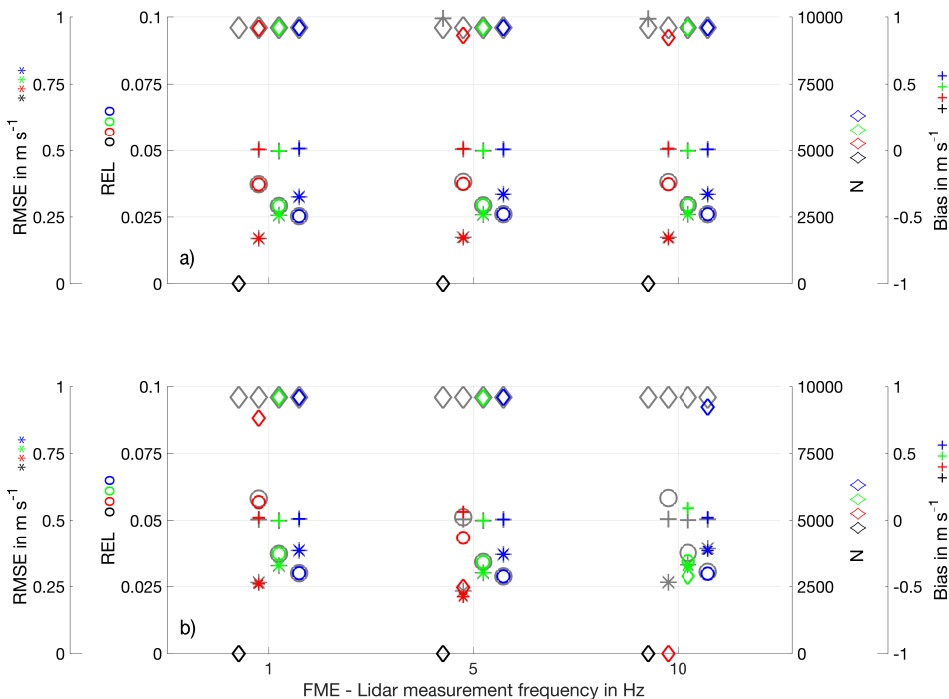

**Figure 11.** RMSE, REL, N and bias for three lidar measurement frequencies using STP, RET otherwise. a) Ideal system. b) Noisy system. Color-coded points as in fig 6.

are only valid for the specific noise setup used. They can be altered for a real system with differing radial velocity and beam pointing inaccuracy. If these are known, a further system optimization can be conducted using the ADLS.

### 4.3 Influence of retrieval settings on wind profiling quality

As a volume velocity processing (VVP) type of algorithm (Waldteufel and Corbin, 1978; Boccippio, 1995) is applied to infer
5 the wind field parameters, a number of options exist to vary the retrieval and inversion settings. In this study, the impact of the number of retrieved parameters (VAR, only horizontal components vs. including vertical component) and the way in which the volume is defined (SEP, volume-based, time-based) is analyzed. The horizontal along-track averaging distance is varied (AVG $650, 1300, 2600$ m) as well as the vertical extent of the volume used for processing (LEV $62, 124, 248$ m). Further, stability of the wind retrieval with sectors being omitted is investigated, to assess effects such as data omissions due to the presence of
10 clouds in real measurements (BLA $0 - 90°, 0 - 180°, 0 - 270°$).

The **number of retrieved parameters (VAR)** is an important setting that can be varied in the inversion process (Boccippio, 1995). This study focuses on the wind profile retrieval under inhomogeneous conditions at highest resolution with a minimum number of data points. Therefore, the two most simple options are analyzed:



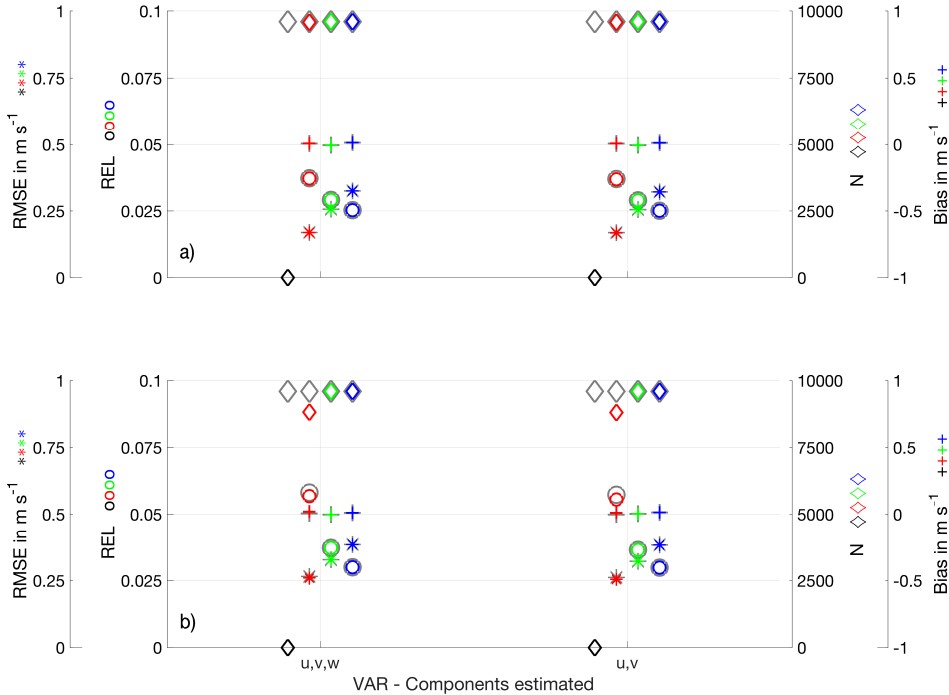

**Figure 12.** RMSE, REL, N and bias for u,v,w and u,v component estimation approaches using STP, RET otherwise. a) Ideal system. b) Noisy system. Color-coded points as in fig 6.

1. Retrieval of the full wind vector, consisting of the u,v,w component (Leon and Vali, 1998; De Wekker et al., 2012).

2. Retrieval of the horizontal wind vector only, consisting of the u,v component (Koch et al., 2014).

If the w component is neglected in the inversion, any persistent vertical wind speed causes a bias in the LSQ-fit. On the other hand, the stability of the solution could be increased for cases in which short scans/few points are available.

5   The number of retrieved components does not exhibit a discernible influence on retrieval quality (fig. 12). Both retrieval methods show equal error levels and an equal number of retrieved wind profile points. This insensitivity goes hand-in-hand with the observation that the average vertical wind speed can be retrieved with reasonable quality (fig. B3), and so retrieving all three wind components is the preferable choice. Inclusion of system noise in the retrieval process does not change the discussed behavior. The results could be altered in more challenging measurement conditions (e.g. due to the presence of

10  sector blanking), as for sector blanking higher CNs are reached.

For the AVAD analysis, all measurements within a defined volume are evaluated for the wind profile retrieval (essentially making it a VVP approach). The vertical and horizontal extent of the volume need to be specified by the user depending on the intended observational task. In the ADLS two options for the **separation of wind profiles (SEP)** used for processing exist:



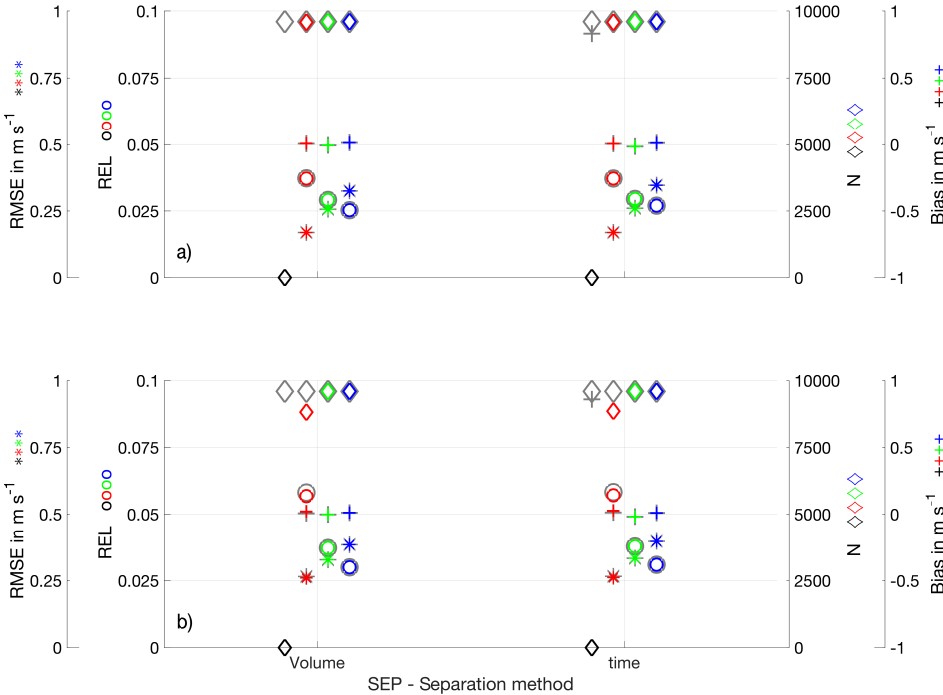

**Figure 13.** RMSE, REL, N and bias for volume and time-based profile separation approaches using STP, RET otherwise. a) Ideal system. b) Noisy system. Color-coded points as in fig 6.

1. Retrieval of the wind profile in fixed, ground-based volumes with a given size that does not depend on the flight direction (Guimond et al., 2014; Tian et al., 2015). In this case, upwind flights contain more samples per given volume than downwind flights.

2. A time-based approach where all measurements within a defined time period (e.g. individual scans) are analyzed independent of their pointing direction and sampling volume (Leon and Vali, 1998; De Wekker et al., 2012). In this case, downwind flights yield slightly degraded and upwind flights slightly improved horizontal resolution. In addition, the scanned volume and thereby measurement footprint can be very large as aircraft motion and scanner movement are added together.

ADLS results show that the separation method does not exhibit a strong influence on the retrieval quality (fig. 13). This insensitivity is expected, as both methods use a similar amount of data from the same setup to retrieve the wind vector. Due to the fixed and known retrieval footprint, the volume-based approach is therefore the preferable choice. The described results remain valid when artificial system noise is considered in the measurement process.

For any of the strategies, the extent of the averaging period of data used for the VVP processing has to be specified. Consequently, the effect of varying the extent of the horizontal averaging domain is analyzed using the ADLS. For the standard



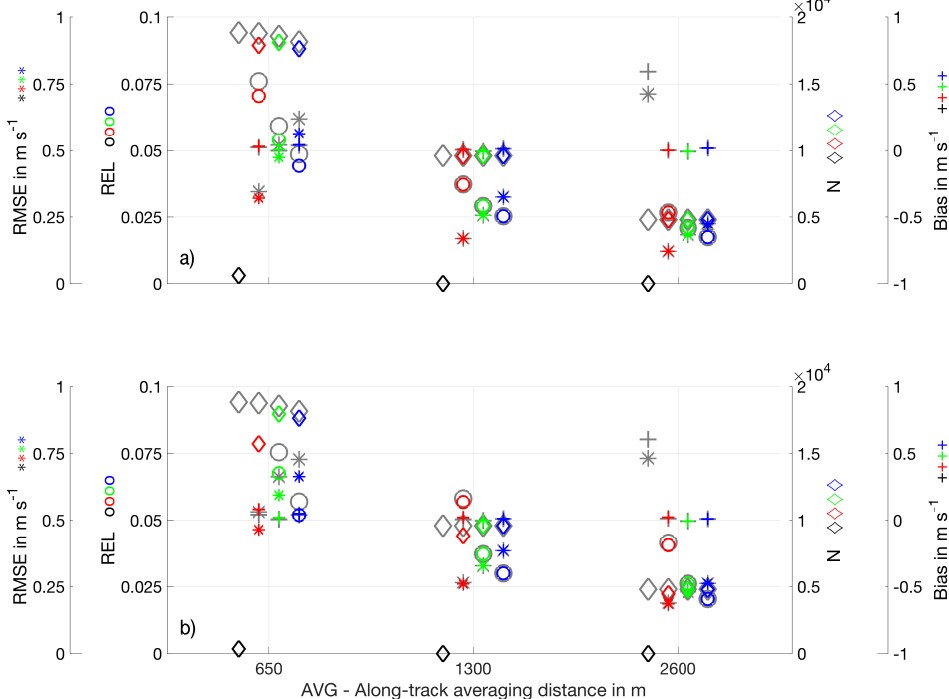

**Figure 14.** RMSE, REL, N and bias for three along-track averaging distances using STP, RET otherwise. a) Ideal system. b) Noisy system. Color-coded points as in fig 6.

system setup and retrieval setting the **along-track averaging distance (AVG)** is varied between $650, 1300$ and $2600$ m (determining the along-track wind profile resolution). The along-track distances correspond to the approximate distance covered by the aircraft during $0.5, 1$ and $2$ scanner rotations. However, the moving aircraft allows for directing the laser beam ahead and back and therefore measures the same spot at different times under varying azimuth angles. The across-track averaging

5    distance is set to $1963$ m for all volumes, corresponding to the across-track ground distance covered by the lidar beam at a flight altitude of $h = 1700$ m and the standard scan elevation of $\phi = 60°$.

    As expected, varying the along-track averaging distance used for profile retrieval has a strong influence on wind profile error (fig. 14). High-resolution wind profiling at $650$ m is associated with a larger RMSE which can be lowered to some extent by quality filtering. For higher background wind cases, the strongest increase is caused by downwind profiling directions which

10   exhibit elevated error levels (not shown). Interestingly, the application of the quality filtering criteria does not completely filter the $0$ m s$^{-1}$ background wind case anymore, leading to a severely biased retrieval. This is further discussed in Sec. 4.4. The error decreases for all wind speeds if a longer averaging distance of $1300$ m is used, and in this case differences between up- and downwind wind profiling become smaller as discussed before. The decrease, however, is not linear and a further extension of the averaging distance to $2600$ m does not yield an improvement of similar magnitude. It does, however, lead to the $0$ m s$^{-1}$





background wind case appearing within the reliable retrieval limits for both RMSE and bias. This analysis can also be inverted, that is, if an acceptable error threshold is given, the required along-track averaging distance can be determined (including usage of the quality filtering analysis, for which the 650 m case is used as a challenging setting). When system noise is included in the analysis, the above described behavior holds at overall higher error levels. As expected, the increase in error due to system

5  noise is more pronounced for the shorter along-track averaging distances. Further, the quality filtering eliminates more points for shorter distances, resulting in slightly decreased error levels.

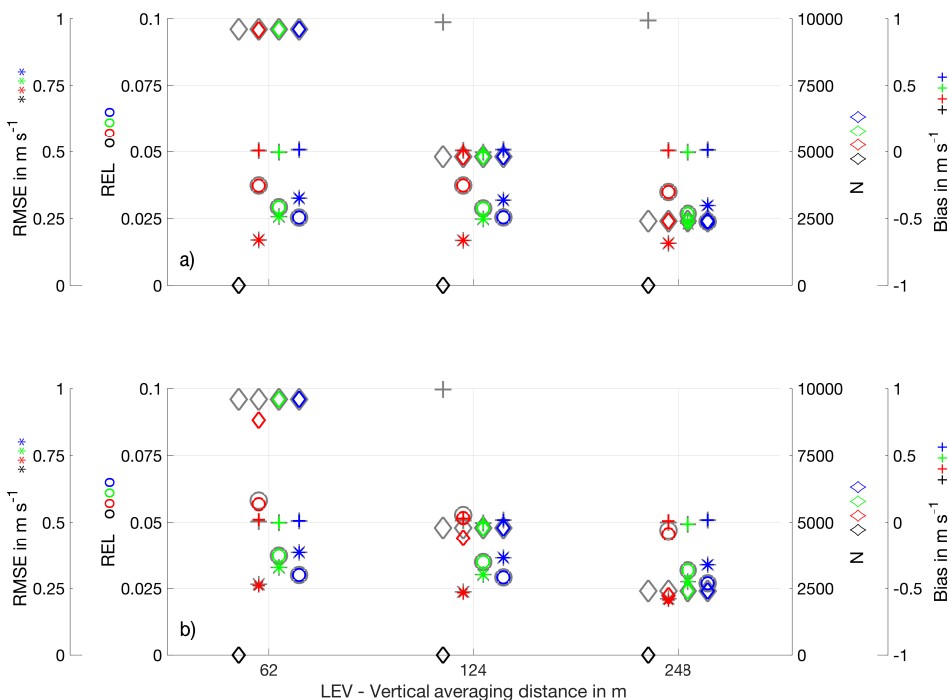

**Figure 15.** RMSE, REL, N and bias for three vertical averaging distances using STP, RET otherwise. a) Ideal system. b) Noisy system. Color-coded points as in fig 6.

Another parameter which can be varied is the **vertical averaging distance (LEV)**, corresponding to the vertical wind profile resolution. Depending on the vertical analysis volume extent, the number of neighboring range gates used to create one wind profile point differs and can influence the LSQ-fit quality. The vertical averaging distance is varied between 62 m, 124 m and

10  248 m, which corresponds to one, two or four range gates falling into the retrieval volume at 60° elevation. The influence of varying the vertical averaging distance is hardly discernible in terms of wind profile error (fig. 15). However, the number of retrieved wind profile points varies strongly, as expected, because the vertical profile resolution is changed. The reason for vertical averaging having no influence is founded in the turbulence structure. Neighboring range gates probe mostly similar turbulent elements and thereby suffer from similar errors. Therefore, including more range gates in the LSQ-fit does not


improve the retrieval quality as the additional data is highly correlated. Only a marginal reduction is observed for the 248 m vertical averaging distance, but at the cost of having far fewer samples available, which can be problematic in regions of strong vertical shear. Another conclusion from the error characteristics is that wind profile error is usually correlated over multiple vertical levels. Consequently, Doppler lidar retrieved wind profiles appear smooth, but can still suffer from non-negligible error

5 throughout the full profile height (resembling a bias, but as this error averages to zero over many profiles it is more accurately described as a random error). This correlation is the reason why the error discussed here is difficult to identify visually in real Doppler lidar wind profiles: No small-scale variations occur, but rather offsets over large sections of the profile. When artificial measurement noise is introduced, vertical averaging over multiple range gates improves the retrieval quality and the number of retrieved wind profile points slightly, with the improvement being stronger for the $5 \, \mathrm{m \, s^{-1}}$ background wind case. With more

10 vertical averaging, more measurements fall into the retrieval volume, and so consequently the radial velocity noise is reduced due to averaging (beam pointing inaccuracy is not, as it is the same for all vertically neighboring range gates).

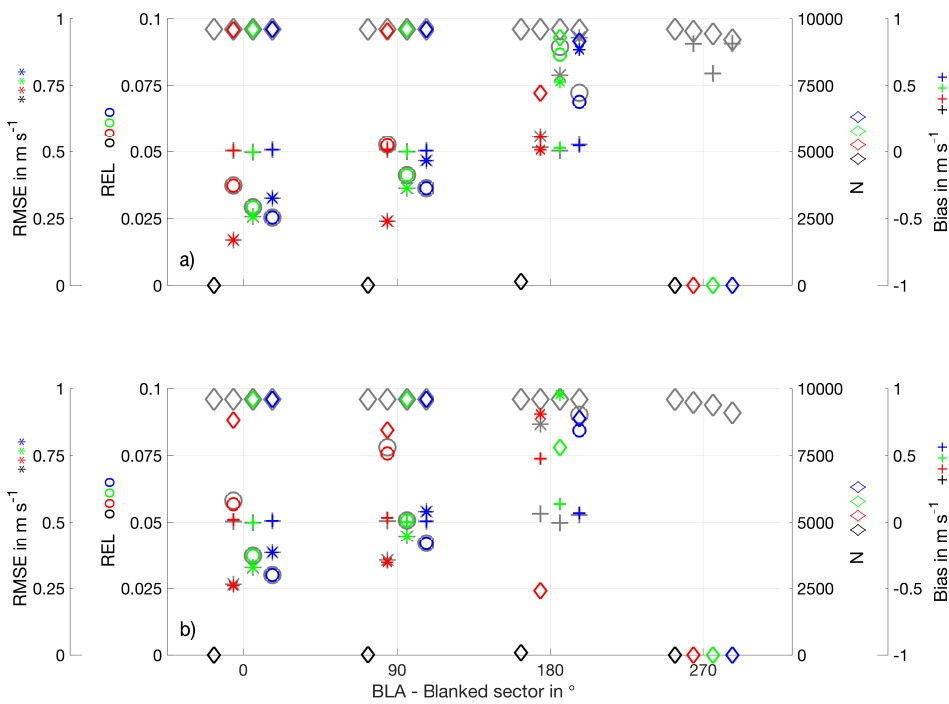

**Figure 16.** RMSE, REL, N and bias for four blanked sectors using STP, RET otherwise. a) Ideal system. b) Noisy system. Color-coded points as in fig 6.

Last, the robustness of the retrieved wind profile to **partial data unavailability (BLA)**, here termed 'blanking', is investigated (fig. 16). In real measurements in the boundary layer, sectors of the scan can become unavailable due to the presence of clouds, rain, obstacles or other disturbances. The effects are simulated by discarding a specific sector of the scan geometry for





wind profile retrieval (in the ground-based reference system ENU, in which the disturbance would be present). For the analysis, three cases are considered: sector blanking from $0-90°, 0-180°$ and $0-270°$. A sector blanking of $90°$ leads to a moderate increase in the average error levels for all background wind cases without significant loss in the number of retrieved wind profile points. With a further increase to $180°$ sector blanking the wind profiling quality is strongly degraded, which is more

prominent for the downwind flight directions (not shown). More of the $5\,\mathrm{m\,s^{-1}}$ background wind speed retrievals are quality filtered, which again introduces a slight bias. Further, the exclusion of the $0\,\mathrm{m\,s^{-1}}$ background wind case is not completely successful anymore, which again results in a severely biased retrieval. With $270°$ sector blanking, the CN exceeds values of 25 and no reliable wind profile retrieval is achievable anymore. Once again, the introduction of noise does not change the discussed behavior significantly but results in elevated error levels especially for the $5\,\mathrm{m\,s^{-1}}$ background wind case. Further, a

reduced number of wind profile points is retrievable and the retrieval becomes biased beyond $90°$ sector blanking.

### 4.4 Quality filtering criteria

In this section, the performance of the most commonly used quality criteria $R^2$ and CN is evaluated, specifically their relation to wind profile quality and their adequacy in detecting violations of the homogeneity assumption and collinearity in the model geometry. A first performance check is conducted for the standard system setup and retrieval setting for all parameters. In a

second performance check, the high resolution retrieval setting (AVG 650 m) is used as in this case the analyzed volumes show varying coverages by the airborne Doppler lidar, making reliable estimation of the wind vector more challenging.

The analysis of the simulator results for the ideal measurement system setup and standard retrieval setting reveals a number of interesting findings (fig. 17). Deviations from the geostrophic background wind towards lower wind speeds, associated with stronger turbulence in the boundary layer, generally decrease the value of the $R^2$ but do not influence the CN, as it is expected

(fig. 17 c, f). Even at high $R^2 > 0.90$, significant wind profile errors up to $1\,\mathrm{m\,s^{-1}}$ do exist. The CN values cluster in a narrow range with values between $3-4$, as the sampling volume is generally well-explored by the lidar.

A noteworthy feature occurs for the $0\,\mathrm{m\,s^{-1}}$ background wind case. Here, a clear dependence of wind speed error on the $R^2$ value is observable (fig. 17, a, b, c). The estimated wind speeds are only slightly biased for coefficients of determination in the range close to 0. However, with increasing $R^2$, the bias of the retrieved wind speed increases linearly to values in the range

$1-2\,\mathrm{m\,s^{-1}}$. This problem exists despite the fact that the individual estimated components themselves are unbiased (fig. B3). Coefficients of determination above 0.9 are non-existent for the $0\,\mathrm{m\,s^{-1}}$ background wind speed, therefore the biased values can be filtered without problems.

The observed behavior is caused by the lidar measurement method: at low wind speeds, the retrieval is strongly influenced by the vertical wind and less by the horizontal wind, especially at steep elevation angles. For high coefficients of determination,

the $w$ variations are more in-phase (with an expected horizontal wind contribution from the simplified horizontal wind field model) and thereby mapped more into horizontal wind, causing a stronger positive bias. The values of the horizontal wind speed in the range of $0-3\,\mathrm{m\,s^{-1}}$ correspond with a mean vertical wind of $0-1.5\,\mathrm{m\,s^{-1}}$ amplitude being mapped into the horizontal wind at $60°$ elevation. At lower coefficients of determination, the noisy (sub-scan-volume) $w$ variations overwhelm the smaller horizontal wind speed signal. However, they do not lead to a mean wind speed being estimated as they cancel out



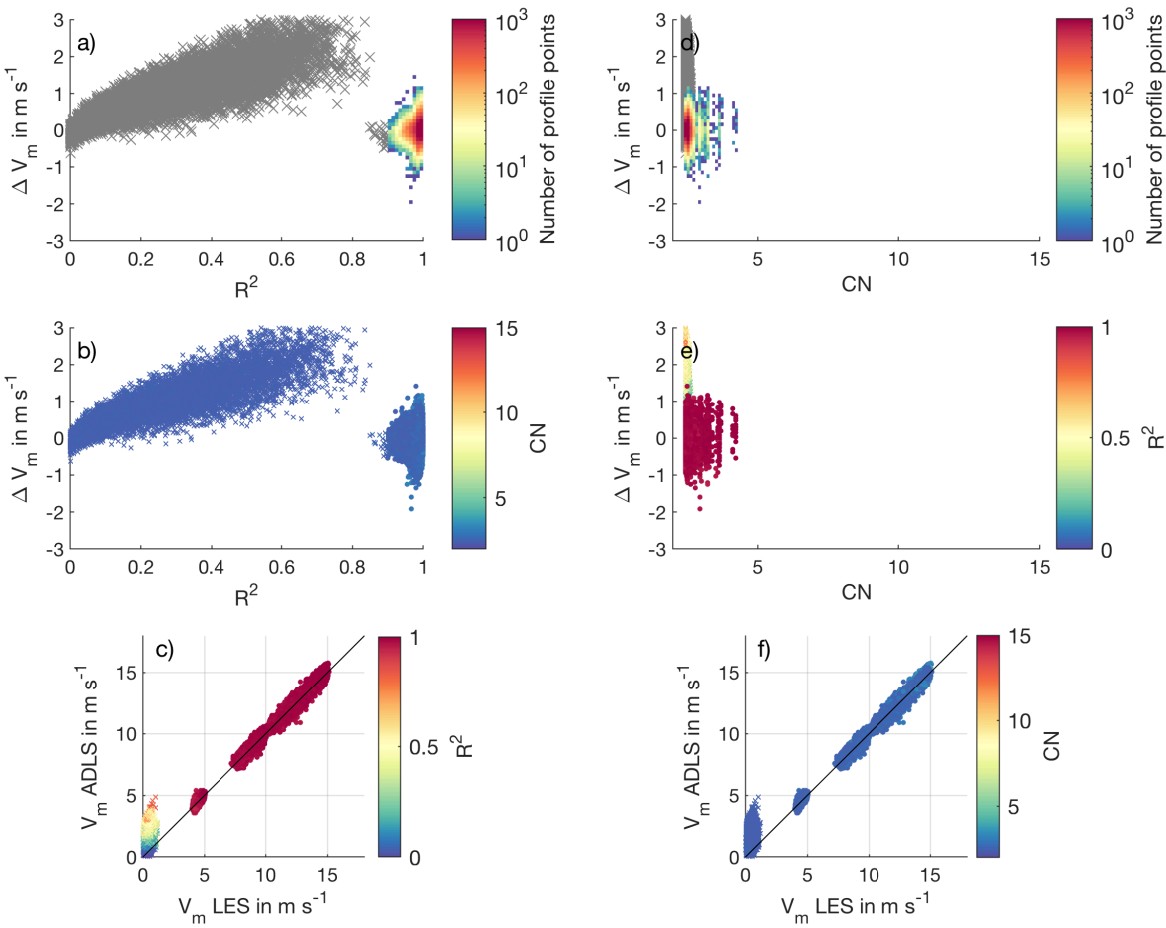

**Figure 17.** Quality filtering criteria for the standard system setup and retrieval strategy (STP, RET). a) Color-coded histogram of number of occurrence for retrieval error and $R^2$. Gray crosses show values eliminated by quality filtering. b) Retrieval error and $R^2$, color-coded is CN. c) LES wind speed and ADLS wind speed, color coded is $R^2$. Profile points which pass quality filtering are displayed as color-coded circles, profile points which are eliminated as color-coded crosses. d) Same as a), but for CN. e) Retrieval error and CN, color coded is $R^2$. f) Same as c), but for CN.

due to averaging, thereby causing no bias. If the scalar averaged LES wind speed is used for lidar comparison instead of the vector averaged LES wind speed discussed here, the described linear trend is equally present, but with an intercept offset of $-1\,\mathrm{m\,s^{-1}}$ along the y-axis.

In summary, for low wind speeds, the application of a higher $R^2$ mainly filters for smoother in-phase variations of the vertical wind (and/or system noise, if present). Consequently, selecting a higher threshold for $R^2$ can cause a higher bias, an important finding that is not expected and, to our knowledge, undocumented so far. For low wind speed cases, the $R^2$ is not an appropriate quality filtering criteria, unless an appropriately high threshold is chosen which then successfully filters the low wind speed





cases. In the investigated case, the $0$ m s$^{-1}$ background wind cases are completely eliminated by setting strict quality filtering criteria. However, if more challenging conditions are used, even the few points that remain can cause a non-negligible retrieval bias due to their large estimation error (fig. B4).

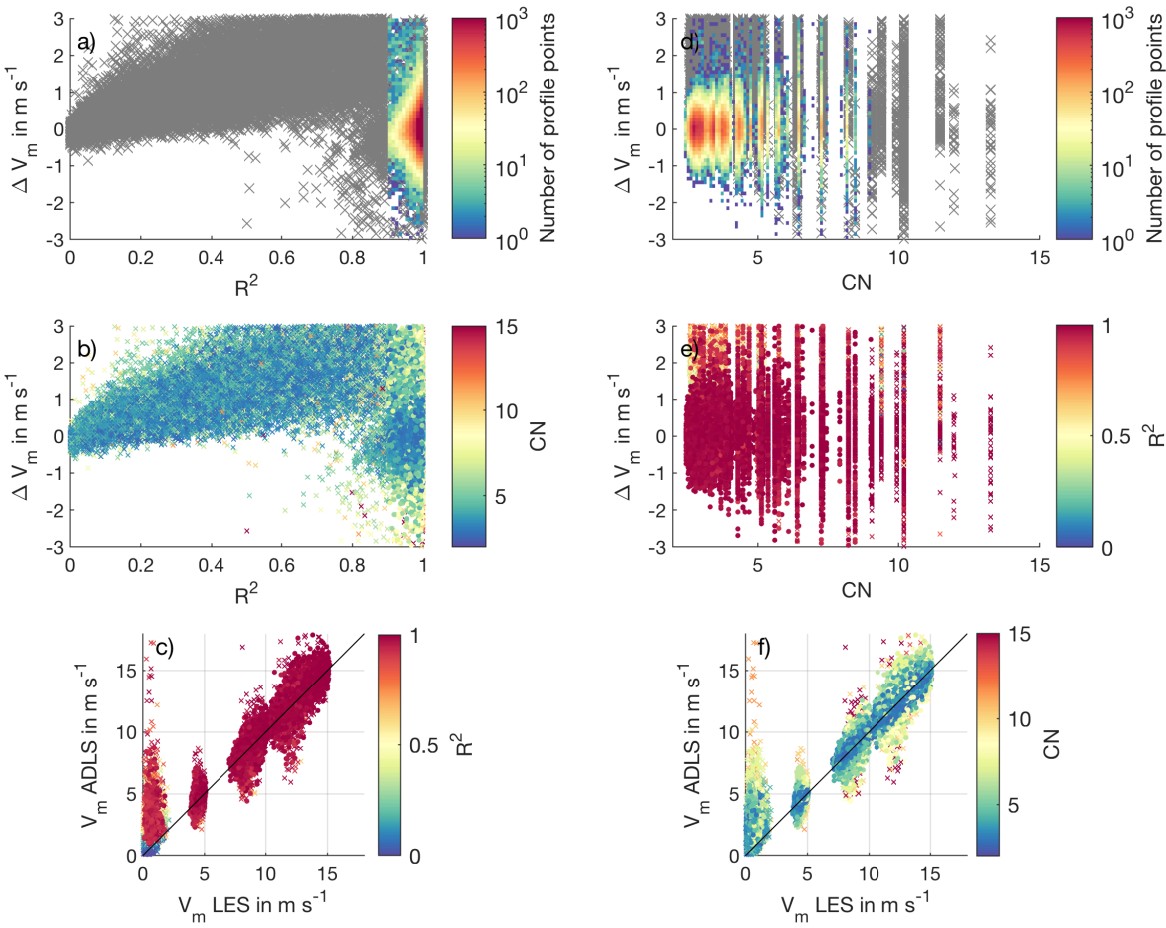

**Figure 18.** Quality filtering criteria for along track averaging distance AVG $650$ m and STP, RET otherwise. Panel labeling same as fig. 17.

In the challenging analysis setup using a shorter averaging distance AVG $650$ m (fig. 18), a higher spread of the retrieved wind profile points compared to the longer averaging distance is apparent. This increased spread arises from the only partial exploration of some retrieval volumes by the lidar. Therefore, a greater range of $R^2$ and CN are explored by the data, and relatively more wind profile points are excluded (see also fig. B4). The previously discussed retrieval bias for low wind speeds with increasing $R^2$ is prominent again. As some of the biased points with a $R^2$ higher than $0.9$ are determined from a good sampling distribution (small CN), they are not completely excluded by the quality filtering criteria. Consequently, the vertical wind mapping causes a strong positive wind speed retrieval bias for low wind speeds.





Nevertheless, in general, outliers are connected to either a high CN or a decreased $R^2$, stressing the importance of both quality filtering criteria. For this retrieval setting, the wind speed retrieval error starts to show outliers beyond 3 m s$^{-1}$ for CN> 9. A reduction of profile error spread is observable for CN< 5 with the loss of relatively few wind profile points. Additionally, relatively few points lie in the region $R^2 = 0.9 - 0.95$, therefore the results justify the usage of stricter quality filtering criteria

in this case to obtain a reduced profiling error. In this way, the ADLS allows for determining a meaningful quality threshold for each setting depending on the accuracy desired.

A possible bias for unknown reasons, varying by day, has been reported by Weissmann et al. (2005) already. They stress the need for further investigation of this phenomena using simulated lidar data. The bias at low wind speeds is also noticeable in studies comparing lidar-measured wind speeds to in-situ measurements, however without discussion or explanation so far.

Some recent examples of the observed overestimation at low wind speeds are, besides others, visible in Holleman (2005) (ground-based; fig. 4, fig. 5), Chouza et al. (2016a) (airborne; fig. 2, fig. 3b under turbulent conditions inside the boundary layer) and Pauscher et al. (2016) (ground-based; fig. 5e).

Therefore, based on the above discussion, we recommend the following procedure in wind profile analysis: In doubtful situations, where the approximate magnitude of the measured wind speed is unknown, one should always analyze a long

spatial average over multiple scan rotations first (approx. 10 times the expected maximum eddy-size). For longer averaging, the mapping of vertical wind does not influence the retrieval significantly as the structure of the vertical wind is not in-phase with the measurement geometry over long distances. Thereby, the approximate magnitude of the wind speed can be analyzed reliably, although at coarse spatial resolution. If the outcome of this analysis is a wind speed magnitude below 5 m s$^{-1}$, a further analysis at higher spatial resolution not advisable due to the possibility of erroneous mapping. If, on the other hand, the

wind speed magnitude is above this threshold, the spatial resolution of the retrieval can be refined further.

When leaving the low wind speed problem aside, even when using strict quality filtering criteria, considerable wind speed retrieval error can still exist due to inhomogeneities. Or, stated in another way, even relatively small deviations from the homogeneity assumption that do not lead to significant degradation of the quality filtering criteria can cause noticeable retrieval error. For these cases, the possibility of an in-situ uncertainty forecast based on the measured radial velocities appears worthwhile to

investigate.

## 5   Conclusions

In this study, an LES-based airborne Doppler lidar simulator, the ADLS, and its capabilities are presented to provide a valuable link between Doppler lidar theory and measurements in order to optimize instrument deployment and to quantify wind profiling error.

The ADLS utilizes an LES wind field at 10-m grid spacing to simulate the turbulent convective boundary layer with geostrophic background wind speeds of $0, 5, 10$ and $15$ m s$^{-1}$. The emulated measurement system consists of an aircraft, scanner and lidar, for which varying setups are available, considering many geometric transformations applicable in real-world measurements. After system emulation, a simulated measurement is conducted based on the LES wind fields. Here, both an



ideal system and a noisy system with an added Gaussian distributed random radial velocity fluctuation and beam pointing inaccuracy are simulated. Finally, the wind profile retrieval is performed on the simulated measurement data using a variety of retrieval strategies. The wind profiles are then evaluated against the original wind field data supplied as input, thereby revealing errors due to the lidar measurement process. The ADLS does neither include a simulation of the signal return (aerosol

scattering) process nor of the lidar instrument physics. These processes have been investigated adequately elsewhere and problems associated with them are expected to be of manageable magnitude in practice. Instead, a direct measurement approach as chosen by other recent studies is further extended here (Stawiarski et al., 2013; Guimond et al., 2014). Nevertheless, the effect of system noise in the measurement process can be emulated through superimposed random radial velocity fluctuations in the simulated measurements and are considered in this study.

As an example application of the ADLS, the accuracy of high-resolution airborne wind profiling under inhomogeneous flow conditions is investigated. The ADLS demonstrates its usefulness as representation problems present in studies comparing lidar to in-situ measurements can be avoided. Thereby we are able to provide an analysis of wind profile retrieval quality for various system setups and retrieval settings.

The results show that wind profiling retrieval errors due to inhomogeneous flow conditions can be of considerable magnitude

and even render the retrieved wind profiles useless in the case of low wind speeds or inadequate system setup and retrieval settings. Under the conditions investigated, errors due to flow inhomogeneities are larger than those due to the worst-case random radial velocity fluctuations and beam pointing inaccuracy. For low wind speeds the system accuracy can become important if the assumed worst-case system noise levels are exceeded. Nevertheless, based on the ADLS, wind profiling error margins can be constrained to an acceptable level, by choosing appropriate system setup and retrieval settings depending on

the accuracy desired.

ADLS results allow for determination of preferential wind profiling strategies to achieve a small measurement footprint. In the comparison of flight altitude decrease versus scan elevation increase, this study shows that a decrease in flight altitude is the preferable option. Although at the cost of decreased vertical coverage, for a decreased flight altitude the retrieval quality remains almost unaffected. A possible remedy for the decreased vertical coverage could be a system which can scan down-

as well as upward iteratively, thereby providing wind information inside the boundary layer as well as above. When using a fast-switching 10-Hz scanner and lidar system, wind profiling could be achieved without loss of along-track resolution due to the autocorrelation present in high-frequency measurements.

Scan elevation angles steeper than 70° are problematic under turbulent conditions in the boundary layer due to the strong influence of vertical wind on retrieval quality. With the specific retrieval setting used in this study, for the scan elevation angle

an acceptable error vs. measurement footprint trade-off is achieved at 60° elevation. For elevations greater than 70°, the wind profiling error increases rapidly, the retrieval can become biased and a smaller number of wind profile points is retrieved. A faster scan speed can have a positive effect on retrieval quality if random system noise remains low, as in this case autocorrelation between adjacent measurements is decreased. The same principle is valid for increasing measurement frequencies: for low noise systems, the benefits of higher measurement frequencies are marginal as the additional measurements provide little more




information due to strong autocorrelation. For higher noise systems the autocorrelation between measurements is decreased and higher measurement frequencies can provide better quality.

Retrieval settings can also impact wind profiling quality. For the standard system setup, our results show only marginal differences between retrieving only the horizontal wind components or including the vertical wind. Comparing volume and time-based approaches, both approaches yield comparable error levels but the volume-based approach has the advantage of having a defined measurement footprint. As expected, the along-track averaging distance has a strong influence on wind profiling quality and longer horizontal averaging increases retrieval quality. In comparison, the vertical averaging distance does not exhibit a strong impact on retrieval quality. Vertical averaging improves retrieval quality only slightly due to the similar structure of turbulence over multiple range gates. Under the standard conditions investigated here, the wind profile retrieval is robust to up to $90°$ of the azimuth scan sector being unavailable (due to clouds or other blocking), at slightly increased error levels.

Our results show that the careful usage of $R^2$ and CN as quality filtering criteria is necessary and adequate, as they can improve wind profiling quality by flagging unreliable measurements (e.g. $0\,\mathrm{m\,s^{-1}}$ background wind case). However, one has to bear in mind that quality filtering by $R^2$ can also introduce a retrieval bias for problematic system setups or retrieval strategies. Quality filtering with CN is always useful and does not introduce any systematic errors. An important new finding of this study is that Doppler lidar wind profiling at low wind speeds under turbulent conditions can be unreliable and even biased despite or even because of applying quality filtering criteria. For low wind speeds erroneous mapping and filtering of the vertical wind into horizontal wind can occur, thereby biasing the retrieved wind speed. The problematic situations cannot be reliably detected by the usage of quality filtering criteria such as the $R^2$ and CN in the least-squares estimation. On the contrary, inadequate usage of the $R^2$ can lead to a higher bias of lidar wind speed retrievals in situations with low background wind speeds. System setups which reduce the magnitude of this problem are those using shallow scan elevation angles or fast scan speeds. However, these come at the cost of having a wider measurement footprint or possibly degraded signal quality. As a remedy, if these system setups are unavailable or undesirable, we propose to always begin the analysis of airborne Doppler lidar wind profiling at coarse resolution, by averaging over long distances. In this way, the wind speed regime can be reliably identified. If, on the one hand, this analysis yields a wind speed magnitude below $5\,\mathrm{m\,s^{-1}}$, a further analysis at higher spatial resolution is not advisable. If, on the other hand, the wind speed magnitude is above this threshold, the spatial resolution of the retrieval can be refined further.

Overall, this study highlights the benefit of an LES-based airborne Doppler lidar simulator. It offers a promising opportunity to investigate the lidar measurement process more closely not only with respect to wind profiling, but for example also with respect to estimating turbulent quantities from airborne Doppler lidar measurements.

*Code and data availability.* The underlying MatLab code and data are available from the author upon request.



*Author contributions.* PG developed the simulator, performed the simulations, conducted their evaluation and prepared the manuscript which was improved by all co-authors who acquired the project funding and also contributed in supervising the work.

*Competing interests.* The authors declare that they have no conflict of interest.

*Acknowledgements.* Christoph Knigge and Siegfried Raasch from the University of Hanover are acknowledged for granting the right to use

5   the LES data. The first author is thankful for PhD supervision by Christoph Kottmeier.





## Appendix A: Geometric calculations for the ADLS

### A1 Triangle of velocities

Given for this example are the in-air speed $IAS$, the aircraft ground track direction $TR$, the wind speed $WS$ and the wind direction $WD$. Directions are to be given in degree from north and speeds in the same units. Needed are the aircraft heading

$HDG$ and the ground speed $GS$. We can calculate them according to the following formulas:

$$HDG = TR + \arcsin\left(\frac{WS \cdot \sin(TR - WD)}{IAS}\right), \tag{A1}$$

$$GS = IAS \cdot \cos(HDG - TR) + WS \cdot \cos(TR - WD). \tag{A2}$$

The calculations can be performed for other combinations of given and needed variables as well.

### A2 Coordinate transforms

As outlined in Lenschow (1972), Lee et al. (1994) and Leon and Vali (1998) transformations between the LES earthbound (E) coordinate system, oriented east-north-up (ENU), and the aircraft (A) coordinate system, oriented along aircraft-right wing-down (ARD), are achieved by using the standard heading-pitch-roll procedure using the transformation matrix $\mathbb{T}$. The coordinate transform matrix from the aircraft ARD reference frame to ground ENU reference frame is $T^{AE}$ (A to E). Here, the transformation is exemplary conducted for the beam direction vector $\boldsymbol{b} = [b_x, b_y, b_z]$. It can be transferred between the two

systems with

$$\boldsymbol{b}^E = \mathbb{T}^{AE}\boldsymbol{b}^A \tag{A3}$$

and

$$\boldsymbol{b}^A = \mathbb{T}'^{AE}\boldsymbol{b}^E. \tag{A4}$$

The transformation matrix can be split into separate rotations around the individual aircraft axes.

$$\mathbb{T}^{AE} = \mathbb{H}\mathbb{P}\mathbb{R}. \tag{A5}$$

Hereby, $\mathbb{H}$ denotes a heading rotation at angle $\psi$ (yaw around the z-axis), $\mathbb{P}$ a pitch rotation at angle $\theta$ (pitch around the y-axis) and $\mathbb{R}$ a roll rotation at angle $\phi$ (roll around the x-axis). Individually, they are given as:

$$\mathbb{H} = \begin{pmatrix} \sin(\psi) & \cos(\psi) & 0 \\ \cos(\psi) & -\sin(\psi) & 0 \\ 0 & 0 & -1 \end{pmatrix}, \tag{A6}$$

$$\mathbb{P} = \begin{pmatrix} \cos(\theta) & 0 & \sin(\theta) \\ 0 & 1 & 0 \\ -\sin(\theta) & 0 & \cos(\theta) \end{pmatrix}, \tag{A7}$$





$$\mathbb{R} = \begin{pmatrix} 1 & 0 & 0 \\ 0 & \cos(\phi) & -\sin(\phi) \\ 0 & \sin(\phi) & \cos(\phi). \end{pmatrix}. \tag{A8}$$

Combined, this results in

$$\mathbb{T}^{AE} = \begin{pmatrix} \sin(\psi)\cos(\theta) & \cos(\psi)\cos(\phi) + \sin(\psi)\sin(\theta)\sin(\phi) & -\cos(\psi)\sin(\phi) + \sin(\psi)\sin(\theta)\cos(\phi) \\ \cos(\psi)\cos(\theta) & -\sin(\psi)\cos(\phi) + \cos(\psi)\sin(\theta)\sin(\phi) & \sin(\psi)\cos(\phi) + \cos(\psi)\sin(\theta)\cos(\phi) \\ \sin(\theta) & -\cos(\theta)\sin(\phi) & -\cos(\theta)\cos(\phi) \end{pmatrix}. \tag{A9}$$

**A3  Lidar beam position, motion and averaging of the LES wind field**

In order to determine the LES wind velocity $v_p^E$ which is projected onto the beam, the range gate position has to be calculated. As the position is needed for the LES it must be calculated in the ground reference system. Therefore, after transferring the beam direction into the ground reference system (App. A2), the range gate center positions are calculated by adding the range gate center distance $x_{RGC}$ in beam orientation to the aircraft position,

$$\boldsymbol{p}_{RGC}^E = \boldsymbol{p}_{AC}^E + x_{RGC}\boldsymbol{b}^E. \tag{A10}$$

This calculation is repeated for the range gate begin and end positions by subtracting or adding half the range gate length to $x_{RGC}$:

$$x_{RGB} = x_{RGC} - l_{RG}/2, \tag{A11}$$
$$x_{RGE} = x_{RGC} + l_{RG}/2. \tag{A12}$$

Thereby, the range gate position at the average measurement time is fully characterized, however, the range gate motion due to aircraft motion and scanner movement still need to be accounted for.

The motion of the range gate during one measurement is accounted for by defining a volume between the range gate position at the beginning and the end of the measurement process. The two positions are calculated using a range gate motion vector. The range gate motion vector is constructed by using the difference of the range gate center position compared to the end range gate position of the previous measurement (this assumption is valid as only continuous aircraft and scanner movements are investigated). Using these positions, difference vectors are constructed for the range gate begin, center and end positions. The first and last range gate positions during one measurement are then obtained by subtracting and adding the range gate motion vector from the range gate center position.

The real lidar beam only has a beam diameter of approx. 10 cm. This diameter is not enough to ensure an adequate sampling
of LES data (grid spacing is $\Delta x = 5$ m). Therefore, the beam volume is artificially enlarged in the direction orthogonal to the motion. The factor is set in relation to the grid spacing, a minimum distance of half the grid point distance is employed to





ensure points inside the volume,

$$d_{\text{inflate}} = \Delta x/2. \tag{A13}$$

After all points that fall inside the volume covered by the lidar beam are determined, they are weighted according to the range gate weighting function (Sec. 2.2) based on their orthogonal distance from the beam center. Last, a linear averaging is

5 applied to obtain the average velocity of all points in the volume. This averaged velocity $v_p^E$ is then projected onto the beam direction according to Eq. 3.



## A4 Overview of terminology and error concepts

**Table A1.** Overview of used acronyms and their meaning.

| General | | | |
|---|---|---|---|
| ADLS | Airborne Doppler lidar Simulator | | |
| AVAD | Airborne Velocity Azimuth Display | | |
| ADL | Airborne Doppler lidar | | |
| CN | Condition Number | | |
| LES | Large-Eddy Simulation | | |
| LSQ | Least-square | | |
| $R^2$ | Coefficient of determination | | |
| RET | Retrieval strategy standard | | |
| RMSE | Root-mean-square error | | |
| REL | Relative root-mean-square error | | |
| STP | System setup standard | | |
| VAD | Velocity-azimuth display | | |
| VVP | Volume-velocity processing | | |
| **Simulator options** | | | |
| Name | Description | Standard | Options |
| AAL | Aircraft flight altitude | 1700 m | $1700, 1450, 1200$ m |
| AVG | Along-track averaging distance | 1300 m | $650, 1300, 2600$ m |
| BLA | Sector blanking | $0°$ | $0°, 90°, 180°, 270°$ |
| DIR | Flight direction | 16 directions | 16 directions |
| ELE | Scan elevation angle | $60°$ elev. | $30° - 80°$ elev. |
| FME | Lidar measurement frequency | 1 Hz | $1, 5, 10$ Hz |
| LEV | Vertical averaging distance | 62 m | $62, 124, 248$ m |
| ROT | Scan rotation duration | 20 s rot. | $4 - 20$ s rot. |
| SEP | Profile separation method | Volume | Volume, time |
| SYS | Measurement system type | Ideal, noisy | Ideal, noisy |
| TIME | LES time steps | 25 time steps | 25 time steps |
| VAR | Retrieval variant | u,v,w | u,v,w; u,v |
| WND | Geostrophic background wind case | $0, 5, 10, 15$ m s$^{-1}$ | $0, 5, 10, 15$ m s$^{-1}$ |
| $\sigma v_D$ | Radial velocity noise | $0, 0.25$ m s$^{-1}$ | $0 - 1$ m s$^{-1}$ |
| $\sigma \boldsymbol{b}$ | Beam pointing noise | $0, 0.25°$ | $0° - 1°$ |





**Table A2.** Definition of general terminology as used in this study.

| Term | General definition | ADLS implementation |
|---|---|---|
| | Processes | |
| Measurement | Obtaining a radial velocity estimate from the atmospheric wind field. | Averaging and weighting LES-wind speeds in lidar beam volume to obtain radial velocity estimate. |
| Motion | Change in position. | Calculated using a fixed IAS and the triangle of velocities for the aircraft, the range gate motion vector for the lidar beam. |
| Motion correction | Obtaining a corrected radial velocity from the measured radial velocity by accounting for aircraft motion and aircraft movement moment arm. | Removes the exact aircraft contribution from the measured radial velocity, no error is introduced. |
| (Wind profile) Retrieval | Inferring an estimate of the underlying wind field from a set of radial velocity observations based on an underlying wind field assumption. | The beam matrix inversion process utilizing singular value decomposition. |
| Range gate motion vector | Change in range gate position from one measurement to the next due to aircraft motion as well as aircraft and scanner movement | Calculated as the difference between range gate begin and end position |
| Movement | Change in orientation. Pitch, roll, yaw for the aircraft; elevation, azimuth for the scanner. | Emulated by artificially prescribed values, not connected to aircraft motion in the ADLS. |
| Ideal measurement system | No measurement error in the radial velocities and model geometry is perfectly known. | No artificial noise is added to the obtained radial velocities and no error introduced into the beam pointing direction. |
| Noisy measurement system | Random fluctuations present in the radial velocity estimates (due to lidar instrument noise, turbulence in the range gate volume etc.) and random fluctuations present in the beam pointing direction (due to imperfect aircraft navigation system, scanner pointing etc.). | The noise is emulated by adding random numbers from a Gaussian distribution with specified standard deviation to the measured radial velocities and measured beam azimuth and elevation. |





**Table A3.** Definition of error terminology as used in this study.

| Term | General definition | ADLS implementation |
|---|---|---|
| Errors | | |
| Retrieval error | The difference between the atmospheric input wind vector in the sampling volume and the retrieved wind vector after the measurement and inversion process. | Directly obtained by subtracting the (vector) average of the wind field components used to create the radial velocity measurements from the wind vector retrieved using the measurements through inversion. |
| Total random error, Baker et al. (1995); Frehlich (2001) | All random differences present between multiple horizontal wind profile retrievals. Divided into random error and representation (or sampling) error. | Consist only of random profile error as representation error can be avoided and no other measurement system is simulated. |
| Random (profile) error, Baker et al. (1995); Frehlich (2001) | Random fluctuations in the wind profile retrieval due to instrument noise (especially at low-signal quality), turbulence in the measurement volume and retrieval algorithm stability. | This error contribution is the focus of this study. Adjacent profile points for an individual profile appear smooth despite random error being present as the turbulence between them is similar. |
| Representation or sampling error, Baker et al. (1995); Frehlich (2001) | Observed differences due to sampling of different parts of the atmosphere and/or only partial coverage of the atmosphere by different measurement systems. E.g. between Doppler lidar wind profile retrieval and radiosonde wind profile. | Can be avoided in the ADLS as the atmospheric input data is known exactly and the same volumes are used, therefore not present in this study. |
| Random radial velocity fluctuation, Frehlich et al. (1994) | Random fluctuations in the radial velocity measurement due to instrument noise (especially at low-signal quality), turbulence in the range gate volume and radial velocity estimation stability (spectral peak finding). | Not modeled directly, but emulated by adding random Gaussian distributed values to the measured radial velocities for the noisy system simulation. |
| Beam pointing inaccuracy | Random fluctuations in the beam pointing orientation due to aircraft and scanner vibration or insufficient knowledge of mirror orientation. | Not modeled directly, but emulated by adding random Gaussian distributed values to the scanner azimuth and elevation angles. |



## Appendix B: Additional figures

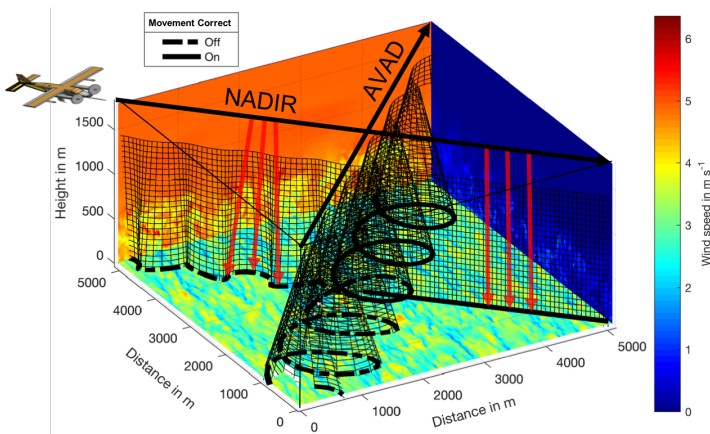

**Figure B1.** Illustration of aircraft trajectory and scanner position simulation for two transects through the LES domain. The LES wind speed is color-coded. The black vectors represent aircraft trajectories. The black curtains show the range gate positions used to conduct the measurement which are calculated from lidar and scanner setting. For the first half of each transect the scanner movement correction is disabled, whereas for the second half it is enabled. The nadir transect is used to retrieve the vertical wind, whereas the AVAD pattern is used to retrieve the horizontal wind.

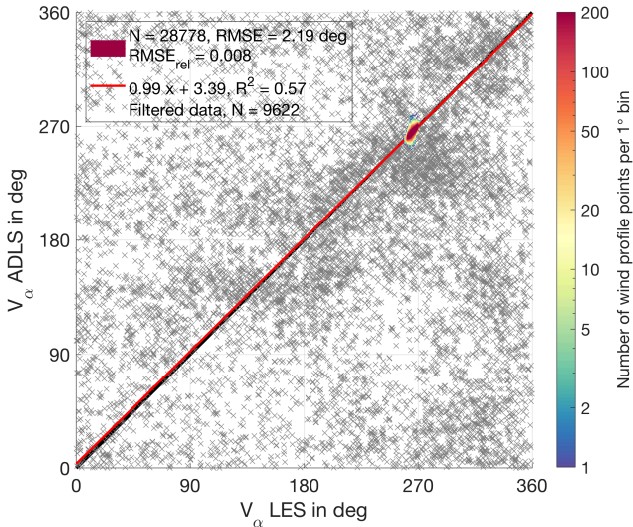

**Figure B2.** LES truth and ADLS retrieved wind direction for an ideal measurement system with STP, RET. Color-coded are all measurements which pass quality filtering, grayed out crosses are the ones which do not. Values which pass quality filtering cluster around $270°$ wind direction. The $0$ m s$^{-1}$ background wind case is filtered completely by applying quality filtering criteria explained in Sec. 4.4.





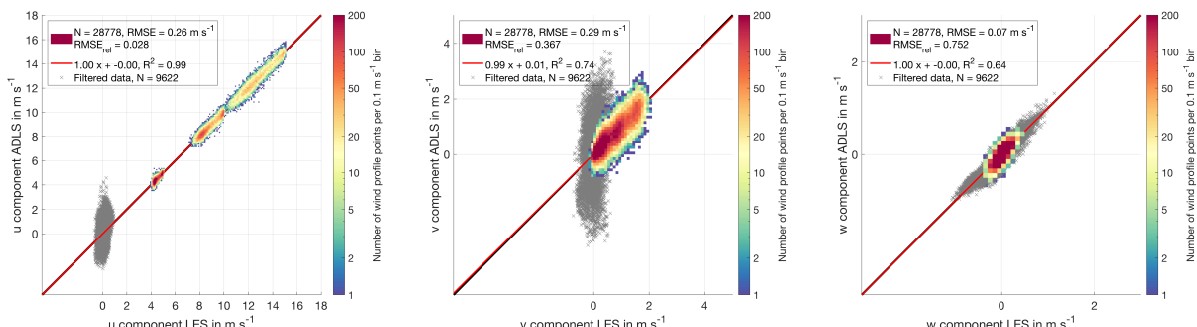

**Figure B3.** LES truth and ADLS retrieved wind speed components for an ideal measurement system with STP, RET. Color-coded are all measurements which pass quality filtering, grayed out crosses are the ones which do not. Quality filtering criteria explained in Sec. 4.4 are applied.

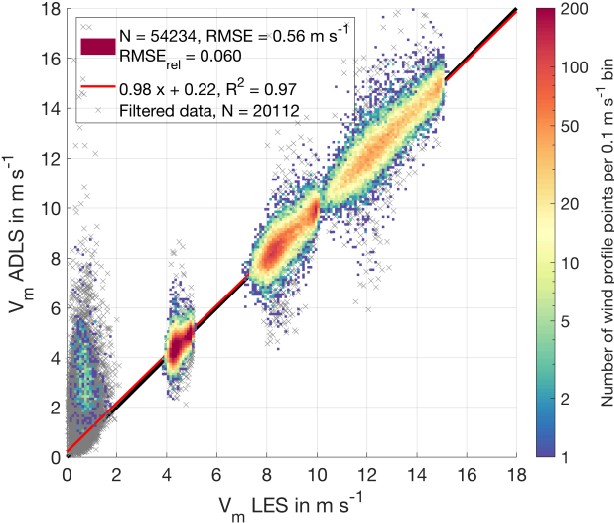

**Figure B4.** Histogram of LES truth and ADLS retrieved wind speed for an ideal measurement system with AVG 650 m and STP, RET otherwise. Color-coded are all measurements which pass quality filtering, grayed out crosses are the ones which do not. Quality filtering criteria explained in Sec. 4.4 are applied.



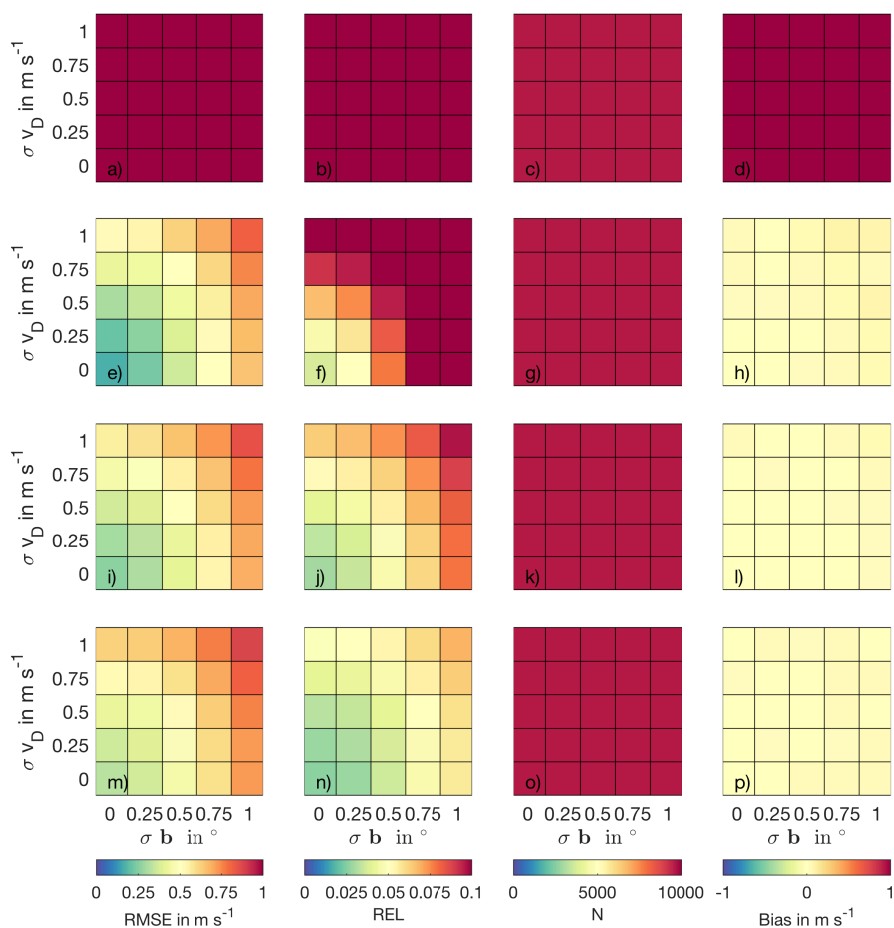

**Figure B5.** RMSE, REL, N and bias for four background wind cases at varying random radial velocity fluctuations and beam pointing inaccuracies without application of quality filtering. a), b), c), d) 0 m s$^{-1}$ background wind case. e), f), g), h) 5 m s$^{-1}$ background wind case. i), j), k), l) 10 m s$^{-1}$ background wind case. m), n), o), p) 15 m s$^{-1}$ background wind case.





**Figure B6.** Profile of wind speed retrieval parameters as a function of height for an ideal system using STP, RET. a) 0 m s$^{-1}$ background wind case. b) 5 m s$^{-1}$ background wind case. c) 10 m s$^{-1}$ background wind case. d) 15 m s$^{-1}$ background wind case.



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
