# Peer review of "An LES-based airborne Doppler lidar simulator and its application to wind profiling in inhomogeneous flow conditions"

_Atmospheric Measurement Techniques, 2019_

## Referee Comment (RC1) · Anonymous Referee #2 · 13 Jul 2019

Review of "An LES-based airborne Doppler lidar simulator for investigation of wind profiling in inhomogeneous flow conditions"

By Gasch et al.

Summary: This is an interesting and useful paper that utilizes output from a small-domain, large-eddy simulation to examine wind retrieval errors from an airborne Doppler lidar. The premise of the study is good, but there are a significant number of revisions needed. While the writing is not terrible, it is far from publication worthy at this point. I provide many fixes listed below, but have not gone through the whole document as that should be the job of the authors. The paper is much too long and I didn't have the time to get to everything. I had many major and minor comments on the methodology section that I only got briefly into the results section. I will review the results section more clearly on the second round of revisions.

Recommendation: Major revisions

Major comments:

(1) The writing style, English and grammar needs work and many comments are listed below. I stopped after several pages because it was taking too much time. Significant re-writing and organizational changes are needed in the manuscript. I also think the paper is much too long. It appears that this journal does not have a page limit, but it would help readers to shorten the discussion in several places and remove sections that are not needed (some examples given below). The paper reads a bit like a dissertation with too much background and drawn out detail. A published paper should be more concise without sacrificing understanding of the problem. Please shorten the results section, it looks like too much information is presented and it might not be worthy to publish all of it.

(2) Throughout the manuscript the word "homogeneous" and "inhomogeneities" are used and this represents a critical aspect of the study and results. For example, "…mismatch between assumed homogeneous wind field models and the wind field inhomogeneities during the measurement process". These are ambiguous terms and I don't understand how they are being used in this context. Since they represent critical points of the paper, it is hard for me to assess the method and results. The authors need to lay out in detail what they are referring to here and clarify this throughout the manuscript. Are you talking about wind variability below the scale of the instrument footprint, grid spacing of the wind retrievals, something else?

(3) The LES domain size of 5 km X 5 km X 1.8 km is extremely small and I have doubts that this domain will represent a realistic environment to test the lidar wind sampling. The authors state that a single flight through the LES does not yield sufficient statistics. However, making 25 different aircraft trajectories through a very small box, probably does not generate any real independent statistics since the retrieved winds are sampling

almost the same flow (the decorrelation spatial scale is probably larger than the box itself). It appears the grid spacing of the wind retrievals might be 1.3 – 1.9 km for along/across track. Given this spacing, I don't think the authors can generate independent flight tracks and statistics through a 5 km X 5 km domain. The authors should try their simulations with a flow in a larger domain (with coarser resolution) in order to study a more realistic environment and allow for independent statistics.

Specific comments (some major, some minor):

Line 2: should say "…additional insights **relative** to ground-based systems…"

Line 2: what does "spatially resolved" mean here?  This term is too ambiguous.

Line 3: "…prepares the ground…", is poor English and needs a change.

Line 4: spell out LES for the first time used; what is meant by "first"?  Note that other studies have used large eddy simulations to study remote sensing instruments.

Line 6: I believe it should be "…wind profiles **in** inhomogeneous flow…".

Line 7: Need to clarify with numbers what is meant by "acceptable error margins". Acceptable is ambiguous and could mean very different things to different people.

Line 7/8: sentence that starts with "Results allow for determination…" should be removed. This is an obvious outcome of the simulations.

Line 8: What is meant by "flow inhomogeneities"? Seems like this is key since much of the manuscript mentions this, but again, this term seems ambiguous to me.

Line 16: What is meant by "short horizontal averaging distances"?

Line 3: Need a comma after "benefits".

Line 11/12: "considering both wind profiling and nadir measurements of the wind field"; I don't understand the difference between these two things as it is written.

Line 13: What is meant by "mean" horizontal flow?  Average over space/time and what scales?

Line 18: I don't understand how winds are retrieved through "inversion of the beam matrix", this sounds like an incorrect statement or writing error. Winds are retrieved by inverting the least squares fit between the model and observations.

Line 25: "…assume homogeneous conditions throughout the sample volume.", what is meant by this statement? One can't measure things that are sub-grid-scale, but I don't understand what this is referring to.

Line 28: Statement about how high elevation angles are used to constrain the footprint. Tilt angles closer to nadir will provide a shorter slant path and thus smaller footprint, but there are other reasons for choosing this steep tilt. Some things could be hardware limitations, range limitations and attenuation.

Line 29: I think it should be "measured radial velocity" and not "retrieved radial velocity".

Line 34: I am still confused on what is meant by "homogeneity assumption".

Lines 15 – 17: Sentence is too long, need to break up for clarity.

Line 24: Need a comma after "capabilities".

Line 25: Need a comma after "systems".

Line 29: "challenged" should be "challenges".

First paragraph: I don't understand what "assume homogeneous wind field and inhomogeneities during the measurement process" means in this context. Need to make significant changes to clarify this and possibly make a diagram to illustrate what this is referring to.

Line 16: Need a period after "follows" instead of a colon.

General question: what is this a large eddy simulation of, homogeneous, isotropic turbulence? The domain size of 5 km X 5 km X 1.8 km is extremely small and I have doubts that this domain will represent a realistic environment to test the lidar wind sampling. The authors state that a single flight through the LES does not yield sufficient statistics. However, making 25 different aircraft trajectories through a very small box, probably does not generate any real independent statistics since the retrieved winds are sampling almost the same flow (the decorrelation spatial

scale is probably larger than the box itself). The authors should try to find a simulation with a larger domain (with coarser resolution) to test the sampling and/or mention that the results of this study are limited to very idealized flow conditions.

Line 19: 65 m/s seems like a low speed to me; what type of aircraft is this instrument targeted for?

Lines 20 – 25: regarding the difference between aircraft heading and ground track…I assume you are talking about drift here. Note that Guimond et al. (2018) found an error in the Lee et al. (1994) mapping equations for Earth-relative coordinates, which don't contain a correction for drift. If you are incorporating drift into your mapping coordinates, this correction should be applied.

Guimond, S.R., J.A. Zhang, J. Sapp and S.J. Frasier, 2018: Coherent turbulence in the boundary layer of Hurricane Rita (2005) during an eyewall replacement cycle. *J. Atmos. Sci.*, **75,** 3071-3093.

Line 25: I don't understand this sentence and the bold claim that this is the "first presentation of a correct airborne sampling simulation", please explain more clearly.

Top half of page: I am confused with this section and Appendix A1. The aircraft position (lat,lon,height) are provided by the GPS on any aircraft and the mapping equations provide locations of the pulse volume centers relative to these positions. Can't the authors just generate a realistic aircraft position vector (possibly from real data) and sample the model winds with that? This extra stuff seems irrelevant. If there is a large aircraft head wind, then the plane might only go forward very slowly and the wind retrievals would only cover a small region. In reality head winds are usually very small relative to the aircraft speed so I don't understand the motivation to get into all this detail. Just use a realistic aircraft position vector because ultimately this will be applied to real situations.

Line 26: Is the 20-second full circle scan time (3 revolutions-per-minute) the lidar scan rate used in the remainder of the study? So with the aircraft speed of 65 m/s, the along-track spacing is 1.12 km? Make these parameters more clearly stated in the paper. Also, what is the grid spacing of the wind retrievals?

Section 2.4 Retrieval – nadir as an example application: I suggest removing this section. The paper is getting much too long and this method has little practical use. Also, the authors say that "wind profiling" is the focus of this study rather than the nadir method.

Line 15/16: "The model is given by the beam geometry…". As stated, this is incorrect, the model is the radial velocity equation, which includes the beam geometry.

Line 22: "…the wind field is usually assumed to be homogeneous…". Again, I don't understand what you mean by homogeneous here. See major comments.

Equations (8) and (9): The matrices U, S and W are not defined so I have no idea how they are used. It is hard for me to evaluate this paper without proper identification of variables.

---

## Author Comment (AC1) · 17 Jul 2019

Dear Referee #2,

We thank you for the initial review of our manuscript and your detailed remarks. Unfortunately, it seems that there was a problem which prevented a full review of our study, particularly the results section. Therefore, we would like to address your major concerns here in an immediate reply. Hopefully our response can enable a further of review of the manuscript.

You find our remarks on your main concerns below. We will address the rest of your questions adequately in the full review. Please do not hesitate to voice any further findings.

Many thanks for your work so far and kind regards,

Philipp Gasch and Co-authors

Major comments:

(1) The writing style, English and grammar needs work and many comments are listed below. I stopped after several pages because it was taking too much time. Significant re-writing and organizational changes are needed in the manuscript. I also think the paper is much too long. It appears that this journal does not have a page limit, but it would help readers to shorten the discussion in several places and remove sections that are not needed (some examples given below). The paper reads a bit like a dissertation with too much background and drawn out detail. A published paper should be more concise without sacrificing understanding of the problem. Please shorten the results section, it looks like too much information is presented and it might not be worthy to publish all of it.

We are sorry to hear that the style, English and grammar did not suit your tastes. We are surprised about the comments about the English, as one of our co-authors is a native speaker and carefully proofread the manuscript. We are therefore very sorry for the inconsistencies that have slipped our attention.

Regarding the shortening: The manuscript was designed for publication and not as a dissertation. Due to the many aspects considered we felt that some background information might be helpful for some readers. We will sincerely consider a further shortening in the review process and have noted your remark on the 'nadir' section.

(2) Throughout the manuscript the word "homogeneous" and "inhomogeneities" are used and this represents a critical aspect of the study and results. For example, "...mismatch between assumed homogeneous wind field models and the wind field inhomogeneities during the measurement process". These are ambiguous terms and I don't understand how they are being used in this context. Since they represent critical points of the paper, it is hard for me to assess the method and results. The authors need to lay out in detail what they are referring to here and clarify this throughout the manuscript. Are you talking about wind variability below the scale of the instrument footprint, grid spacing of the wind retrievals, something else?

In this study we are concerned about errors in wind profiling that arise from the fact that the analysis method assumes that the flow is homogeneous (i.e. constant mean wind profiles), whereas an actual boundary layer experiences flow inhomogeneities driven by turbulence, such as discussed in Shapiro and Fedorovich (2007), Kiemle et al. (2011) and Lundquist et al. (2015).

By the term "wind flow inhomogeneities" we refer to deviations of the wind speed from the mean state due to boundary layer turbulence. In order to consider only turbulent conditions, we limit the extent of our analysis domain to 0-800 m vertically (inside the boundary layer) in Sec. 4. The structure of the turbulence present in the LES is further detailed in our answer to remark (3). The range of the wind variability is presented by the spectra of the turbulence (fig. 5.8 in Stawiarski, 2014). The dominant range of the turbulence spectra is larger than the instrument footprint (10 cm beam diameter, 300 ns Gaussian pulse width, 72 m range gate length), enabling an accurate representation of the turbulence in the simulated measurements. The dominant range of the turbulence spectra is smaller than the horizontal grid spacing of the retrieval (1.3x1.9 km) but larger than the vertical spacing (60 m). This spatial mismatch clearly influences our results in Sec. 4 and is discussed there.

The term homogeneous flow (a synonym is homogeneous wind field conditions) means a uniform flow and the absence of any deviations from the mean flow state, both spatially and temporally (Stull, 2000). Homogeneous flow is assumed in the most simple form of the Velocity Azimuth Display (VAD) and Volume Velocity Processing (VVP) retrievals (Koscielny et al. 1984, Boccippio, 1994, Banakh et. al, 1995, Leon and Vali, 1996). Homogeneous flow is rarely present in the real atmosphere and certainly not inside the boundary layer. The violation of the homogeneity assumption in the VAD/VVP retrieval due to boundary layer turbulence causes an error in the retrieved wind profile. This error is the focus of our study.

We will certainly try to make our focus and the meaning of the term 'inhomogeneity/inhomogeneous flow' as well as 'homogeneous flow' more clear in the revised version of the manuscript.

(3) The LES domain size of 5 km X 5 km X 1.8 km is extremely small and I have doubts that this domain will represent a realistic environment to test the lidar wind sampling. The authors state that a single flight through the LES does not yield sufficient statistics. However, making 25 different aircraft trajectories through a very small box, probably does not generate any real independent statistics since the retrieved winds are sampling almost the same flow (the decorrelation spatial scale is probably larger than the box itself). It appears the grid spacing of the wind retrievals might be 1.3 – 1.9 km for along/across track. Given this spacing, I don't think the authors can generate independent flight tracks and statistics through a 5 km X 5 km domain. The authors should try their simulations with a flow in a larger domain (with coarser resolution) in order to study a more realistic environment and allow for independent statistics.

While we appreciate the reviewer's request for a larger domain, applying coarser resolution would actually undermine the attempt to represent fully developed turbulence. We are confident of the decorrelation spatial scale, as the integral length

scale (Tennekes and Lumley, 1972, Lenschow and Stankov, 1986) of the turbulence present in the LES is $L_i < 500$ m, as stated in section 2.1. Further documentation of the decorrelation scale of the turbulence is provided in the attached fig. 6.3 and fig. 6.6 from Stawiarski (2014) below.

As we constrain the analysis to heights < 800 m (above which gravity waves are present, increasing the integral length scale), the assumption of statistical independence is valid for our analysis volumes of 1.3x1.9 km, which are much larger than the integral length scale (keeping in mind that turbulence is the driver of profiling error which we are interested in).

At the given spatial and temporal integral length scales, we argue that the 16 flight directions, repeated at 1-minute spacing and with a random initial profiling offset, do sample independent air masses. The spatial distance of the wind profiles is larger than the integral length scale. The temporal spacing is on the order of the integral time scale, but aided by decorrelation through advection between subsequent wind profiles.

Unfortunately, we think a coarser model would be problematic to investigate the errors due to turbulent structures in the boundary layer. In our study the grid resolution is 10 m (corresponding to a resolution > 50 m). Thereby, Doppler lidar measurements at a measurement frequency of 1 Hz, with a range gate length of 72 m and a flight speed of 65 m s$^{-1}$ can be realistically represented, as the resolution corresponds to the sampling rate and spatial extent of the range gates. In a coarser model, the high sampling rate would not produce independent measurements. Further, the structure of the turbulence, causing the wind profiling error we are interested in, would be less accurately represented.

Please note that our approach follows established methods. These LES wind fields have been used for Doppler lidar studies in a similar way before (Stawiarski, 2014, Stawiarski et al., 2015) and that other authors choose similar approaches for ground-based studies (Scipion et al., 2009, Scipion, 2011).

We will make sure to discuss the independence of the acquired data in a more clear way in the revised version of the manuscript.

**References**

Banakh, V. A., Smalikho, I. N., Köpp, F., & Werner, C., Representativeness of wind measurements with a CW Doppler lidar in the atmospheric boundary layer. Applied optics, 34(12), 2055-2067, 1995.

Boccippio, D. J., A diagnostic analysis of the VVP single-Doppler retrieval technique. Journal of Atmospheric and Oceanic technology, 12(2), 230-248, 1995.

Kiemle, C., Wirth, M., Fix, A., Rahm, S., Corsmeier, U., & Di Girolamo, P., Latent heat flux measurements over complex terrain by airborne water vapour and wind lidars. Quarterly Journal of the Royal Meteorological Society, 137(S1), 190-203, 2011.

Koscielny, A. J., Doviak, R. J., & Zrnic, D. S., An evaluation of the accuracy of some radar wind profiling techniques. Journal of Atmospheric and Oceanic Technology, 1(4), 309-320, 1984.

Lenschow, D. H., & Stankov, B. B.: Length scales in the convective boundary layer. Journal of the Atmospheric Sciences, 43(12), 1198-1209, 1986.

Lundquist, J. K., Churchfield, M. J., Lee, S., & Clifton, A., Quantifying error of lidar and sodar Doppler beam swinging measurements of wind turbine wakes using computational fluid dynamics. Atmospheric Measurement Techniques, 8, 2015.

Scipion, D.: Characterization of the convective boundary layer through a combination of large-eddy simulations and a radar simulator, Phd, University of Oklahoma, 2011.

Scipión, D., Palmer, R., Chilson, P., Fedorovich, E., and Botnick, A.: Retrieval of convective boundary layer wind field statistics from radar profiler measurements in conjunction with large eddy simulation, Meteorol. Zeitschrift, 18, 175–187, https://doi.org/10.1127/0941-2948/2009/0371, 2009.

Shapiro, A., & Fedorovich, E., Coriolis effects in homogeneous and inhomogeneous katabatic flows. Quarterly Journal of the Royal Meteorological Society: A journal of the atmospheric sciences, applied meteorology and physical oceanography, 134(631), 353-370, 2008.

Stawiarski, C.: Optimizing Dual-Doppler Lidar Measurements of Surface Layer Coherent Structures with Large-Eddy Simulations, KIT Scientific Publishing, Karlsruhe, 2014.

Stawiarski, C., Traumner, K., Knigge, C., and Calhoun, R.: Scopes and challenges of dual-doppler lidar wind measurements-an error analysis, J. Atmos. Ocean. Technol., 30, 2044–2062, https://doi.org/10.1175/JTECH-D-12-00244.1, 2013.

Stull, R., *Meteorology for scientists and engineers*. Brooks/Cole Thomson Learning, Pacific Grove, USA, 1-528, 2000.

Tennekes, H., Lumley, J. L., & Lumley, J. L., *A first course in turbulence*. MIT press, Cambridge, USA, 1-310, 1972.

**Figures** from Stawiarski (2014)

[Figure]

Figure 6.3.: Development of the spatial autocorrelation in the LES $u$ wind fields with the background wind, $u_G = \{0, 5, 10, 15\}$ m/s from left to right.

[Figure]

Figure 6.6.: *Integral length scales*: Scales $L_x$ of the $u$ (left) and $v$ (right) wind fields in mean wind direction for the time series, computed with the full mean and variance of the data set (light green) and the mean and variance of the respective series (dark green). The black bars show the comparative LES results. The bars cover the range between the 25th and 75th percentile, the circles mark the median.

---

## Referee Comment (RC2) · Anonymous Referee #3 · 22 Aug 2019

The article details about the use of LES data to create an airborne Doppler Lidar simulator. The article is aiming to account of the accuracy level of the retrieved wind and investigate the effect of flow in-homogeneity on the VAD scans. It is a well-known fact, that the flow in-homogeneity will cause significant errors in a simple LSQ or SVD type algorithms. There has been sufficient literature and studies on this aspect, using real simulated data. A lot of the sentences have been repeated in the article, one of the reasons for making it such a long article. I should be honest and say that I could not review the entire paper, as I had noted several issues with the article. I would request the authors to re-evaluate, revise and edit the paper significantly before sending the revised version for another round of reviews. But overall I feel its an interesting pa-

per/topic, and has the potential to be published. Some major and minor comments for the authors to address are provided below.

Major comments:

1. The LES domain is too short for this type of study. In the 5 km x 5 km x 1.8 km domain, the authors only use 2 points within to do the analysis for a given trajectory. And approximately 20 trajectories are used in their analysis. The authors also mention that they do not have enough statistics, which is a big concern in such a study. The authors base their methods based on Stawiarski 2014  2015, which is for a ground-based Doppler Lidar and who were looking at detection of coherent structures and the ability of Lidars to resolve those structures. But here, the authors are trying to assess the accuracy of the in-homogeneity, and more simulations and larger domains are a necessity. I can understand that the authors want to have high-resolution LES model data to validate their simulator, but statistically these results cannot be taken seriously. Again, it is very well known that flow in-homogeneity causes errors in Doppler Lidar retrievals, but quantifying the amount for airborne Doppler Lidar measurements is key. And the reviewer feels, that the quantification part is missing here, which is key.

2. The optical effects of the laser beam due to motion has not been considered in the results. It is just demonstrated that the simulator can do motion correction. It would complicate the analysis and make it more realistic and interesting to study such realistic effects (Kavaya 2014  Hill 2008) rather than only flow in-homogeneity (again its a well known issue). Maybe the authors can take motion data from a real aircraft and simulate the effects of wind retrieval accuracy.

3. The article has several repetitive statements all over the document and is extremely long. I feel a large portion of this article can be moved to the supplement section or deleted. The sentence formations are very abrupt, which sometimes I feel are unnecessary and not adding any additional information. I have noted a couple below, but I would urge the authors to take a deeper look at the entire article.

[Figure]

4. The Figures are not very helpful in understanding the results. They all are dumped into one figure with results from several simulations and different type of statistics. Figures 6-16 are very confusing and do not make it obvious without reading the winded text. I can understand the authors sentiment in showing a lot of information, but they need to use smarter approaches such as Taylor's diagram or similar.

Specific comments:

Page 5 Line 10 – The flight temporal sampling is not clear. 1-minute temporal sampling, does that mean the aircraft takes 1 minute to travel the 5 km domain? Isn't that a function of the scan trajectory? Its not very clear. Is the wind assumed to be stationary over that 1 minute?

Page 8 Line 7: Instead of saying "their" weighting function . . . please mention the type of weighting function. Was is a Gaussian or rectangular or? Similarly, other things in the paper.

Page 8 Line 8 – What is deltaX? Please use lidar technical terms or variables to be consistent with the literature.

Page 8: I feel a short table of these characteristics would make it much more legible and easier for the readers.

Page 8 line 19: "can be added to the simulated wind field"? Please refine, as it has been added in the flow field, is my understanding.

Page 8 line 9 – What is FME? The acronym comes doesn't make sense with the sentence before.

Section 2.3 – A motion correction algorithm is introduced, but the author mentions that they have not used any of that in their paper. I feel this section is not useful for this paper, if not used. Please cite a reference, such as Kavaya et al., 2014 or Hill et al., 2009, which has similar motion correction algorithms implemented to real lidar data. If its different that these algorithms, then please state only the difference. Please move

this  the next section to supplement.

Page 3, Line 26... Remove "These include" and just list the references.

Page 5 Line 6... Gravity waves being present above what? – Sentence structure is wrong! The sentence structures sometimes are too abrupt and unnecessary. So maybe the authors have to give it another read and remove some of the abrupt sentence structures. Such as Page 4, Line 26, it is not necessary to have the first statement "The simulator is tested with a set of underlying wind fields". There are other instances such as this and needs to be addressed. I understand that it's a personal choice of the reviewer.

Page 27 line 11: why bold number of retrieved parameters (VAR)? And similar other locations. Please change.

Since there were too many issues with the paper, I feel like the authors need to revise the paper carefully and provide a better and shorter version for the reviewers.

References: Kavaya et al., 2014 - DOI: 10.1175/JTECH-D-12-00274.1 Hill et al., 2008 - https://doi.org/10.1175/2007JTECHA972.1

---

## Author Comment (AC2) · 26 Aug 2019

Dear Referee #3,

We thank you for the initial review of our manuscript and your remarks. Unfortunately, it seems that there was a problem which prevented a full review of our study, particularly the results section. Further, we had difficulties in understanding the exact meaning and background of some of your remarks. Therefore, we would like to address your major concerns here in an immediate reply. Hopefully our response can enable a further review of the manuscript.

[Figure]

You find our remarks on your main concerns below. We will address the rest of your questions adequately in the full review. Please do not hesitate to voice any further findings.

Many thanks for your work so far and kind regards, Philipp Gasch and Co-authors

Please also note the supplement to this comment:
https://www.atmos-meas-tech-discuss.net/amt-2019-118/amt-2019-118-AC2-supplement.pdf

**Supplement:**

Dear Referee #3,

We thank you for the initial review of our manuscript and your remarks. Unfortunately, it seems that there was a problem which prevented a full review of our study, particularly the results section. Further, we had difficulties in understanding the exact meaning and background of some of your remarks. Therefore, we would like to address your major concerns here in an immediate reply. Hopefully our response can enable a further review of the manuscript.

You find our remarks on your main concerns below. We will address the rest of your questions adequately in the full review. Please do not hesitate to voice any further findings.

Many thanks for your work so far and kind regards, Philipp Gasch and Co-authors

Major comments:

1. The LES domain is too short for this type of study. In the 5 km x 5 km x 1.8 km domain, the authors only use 2 points within to do the analysis for a given trajectory. And approximately 20 trajectories are used in their analysis. The authors also mention that they do not have enough statistics, which is a big concern in such a study. The authors base their methods based on Stawiarski 2014 2015, which is for a ground-based Doppler Lidar and who were looking at detection of coherent structures and the ability of Lidars to resolve those structures. But here, the authors are trying to assess the accuracy of the in-homogeneity, and more simulations and larger domains are a necessity. I can understand that the authors want to have high-resolution LES model data to validate their simulator, but statistically these results cannot be taken seriously. Again, it is very well known that flow in-homogeneity causes errors in Doppler Lidar retrievals, but quantifying the amount for airborne Doppler Lidar measurements is key. And the reviewer feels, that the quantification part is missing here, which is key.

It seems that a misunderstanding occurred. Because the error due to flow inhomogeneity is a well-known issue, we do use the simulator to quantitatively assess this error. (We are not trying to validate the simulator and/or assess the accuracy of the inhomogeneity, further we don't state that our statistics are insufficient). To our knowledge, the LES-based analysis is new, which is why we present the first LES-based airborne Doppler lidar simulator.

In the study we are using 16 different flight directions (called trajectories by you) through the LES domain. Each flight direction yields 2 retrieved wind profiles. The 16 flight directions are repeated 25 times at 1-minute temporal spacing (the wind field is not assumed to be frozen during each passage but evolving instead). Thereby, this procedure yields 800 independent wind profiles for each of the four background wind cases (3200 wind profiles overall). Additionally, each of the wind profiles consists of 12 wind speed retrievals at different vertical levels (these 12 wind profile *points* of each wind profile are indeed correlated, as they are in reality), giving a total number of 9600 wind profile points for each background wind case (38400 wind profile points overall). If you are concerned about the spatial and temporal independence of the wind profiles we would like to refer you to our short reply to referee #1.

The 3200 independent wind profiles (38400 retrieved wind profile points) are much more than what is typically used in real world measurement comparisons (e.g. 1612 wind profiles in Weissmann 2005, approx. 10 wind profiles in DeWekker 2012, a single profile in Kavaya 2014, 2056 wind profile points in Bucci 2018).

We would thereby like to know what is insufficient about our statistical database and what you would consider a sufficient statistical database for a quantitative error analysis.

2. The optical effects of the laser beam due to motion has not been considered in the results. It is just demonstrated that the simulator can do motion correction. It would complicate the analysis and make it more realistic and interesting to study such realistic effects (Kavaya 2014 Hill 2008) rather than only flow in-homogeneity (again its a well known issue). Maybe the authors can take motion data from a real aircraft and simulate the effects of wind retrieval accuracy.

As you noted, we do consider the geometric transformations applicable to real world airborne Doppler lidar measurements. Further, we also investigate the effect of a measurement system imperfection. We do this through an assumed scanner pointing inaccuracy (called beam pointing inaccuracy by us), which is likely to occur in real world measurements. We are thereby able to analyze the effect of an imperfect scanner and/or aircraft inertial navigation system, which is done in Sec. 4.1. Further, we also analyze the effect of an imperfect laser system by adding a random radial velocity fluctuation to the measured radial velocities.

We do not see an advantage in using real world aircraft motion data, as the actual motion of the aircraft does not influence the VVP (not VAD) wind speed retrieval quality (only the accuracy with which the motion is known, which we take into account through the beam pointing inaccuracy).

We would thereby like to know what optical effect of the laser beam due to motion you are referring to.

3. The article has several repetitive statements all over the document and is extremely long. I feel a large portion of this article can be moved to the supplement section or deleted. The sentence formations are very abrupt, which sometimes I feel are unnecessary and not adding any additional information. I have noted a couple below, but I would urge the authors to take a deeper look at the entire article.

We will make sure to have another critical look at the wording of the manuscript. Further, we will also shorten its length.

4. The Figures are not very helpful in understanding the results. They all are dumped into one figure with results from several simulations and different type of statistics. Figures 6-16 are very confusing and do not make it obvious without reading the winded text. I can understand the authors sentiment in showing a lot of information, but they need to use smarter approaches such as Taylor's diagram or similar.

We are sorry for the inconvenience. It is very difficult to convey the amount of information we present without a detailed description in the text. We will make sure to

have a look at alternative forms of displaying the information as well as a more compact wording.

References

Bucci, L. R., O'Handley, C., Emmitt, G. D., Zhang, J. A., Ryan, K., & Atlas, R. (2018). Validation of an Airborne Doppler Wind Lidar in Tropical Cyclones. *Sensors*, *18*(12), 4288.

De Wekker, S. F. J., Godwin, K. S., Emmitt, G. D., & Greco, S. (2012). Airborne Doppler lidar measurements of valley flows in complex coastal terrain. *Journal of Applied Meteorology and Climatology*, *51*(8), 1558-1574.

Kavaya, M. J., Beyon, J. Y., Koch, G. J., Petros, M., Petzar, P. J., Singh, U. N., ... & Yu, J. (2014). The Doppler aerosol wind (DAWN) airborne, wind-profiling coherent-detection Lidar system: overview and preliminary flight results. *Journal of Atmospheric and Oceanic Technology*, *31*(4), 826-842.

Weissmann, M., Busen, R., Dörnbrack, A., Rahm, S., & Reitebuch, O. (2005). Targeted observations with an airborne wind lidar. *Journal of Atmospheric and Oceanic Technology*, *22*(11), 1706-1719.

---

## Referee Comment (RC3) · Alan Brewer (Referee) · 22 Oct 2019

Alan Brewer (Referee)

alan.brewer@noaa.gov

An LES-based airborne Doppler lidar simulator for investigation of wind profiling in in-homogeneous flow conditions This is an excellent tool for evaluating airborne Doppler lidar measurement design and will be an invaluable part of experimental design. The authors have done an excellent job of addressing the key considerations of airborne Doppler lidar sampling strategy. General comments: It was not clear to me if the paper was designed to illustrate the capabilities of the simulator using a single set of LES and lidar configurations or if the authors are attempting to explore the operational trade space of an airborne system with sufficient detail so that the community could

use this paper as a design tool rather than the simulator itself. If the goal of the paper is to illustrate the application of the tool, then I think the authors need not explore as much of phase space in every degree of freedom. A single variable plot accompanied by discussion of the underlying mechanism for the relationship for each of the major impacts would suffice. Concentrate on those plots/relationships that show significant structure and focus there. For example Fig 10 would benefit from another approach that highlights the differences between the two plots (highlighting the effect/impact of noise) and changing structure within a plot (slow increase in RMSE/REL). The actual values of the static variables in the plots are only meaningful for one particular set of conditions (both LES and lidar characteristics) and may not be of interest to a wide range of folks. How certain variables change wrt the chosen independent variable is of interest as it will generally do so for all configurations. These relationships should be highlighted and the underlying mechanism explained if possible. If the goal of the paper is to probe the trade space of all systems/conditions, then I think some effort should go into finding fewer "normalized" variables that are independent of specific choices for system values or LES characteristics. For example – my understanding is that the underlying bias sensitivity of the horizontal wind fit to coherent structures in the vertical wind field comes down to how many "pairs" of these features you average/accumulate over prior to doing a fit. As you increase the number of pairs of up/down motions in the volume – the residual of the fit may have a higher RMS, but the bias will decrease. This condition will depend on the dominant spatial scale of the turbulence, the size of the sample volume, the density of the sample points... A potential "normalized" variable might be the number of independent turbulent scale lengths per sample volume. For a given turbulence profile in the LES and a given scan pattern and beam PRF, one could imagine a profile of this "normalized" variable and corresponding uncertainty in the fit... If the authors are able to combine variables and break the analysis into the underlying mechanisms, it may serve to widen the impact of the paper and reduce the number of variables that have to be studied. I'm happy to discuss directly with the authors if need be – I'll ask the editor to share my contact information. For our application, we have a
fixed scan geometry (wedge scanner) so the only system variables we have to adjust are scan rate, beam PRF & pulse width. The adjustable fit parameters are height resolution and number of sweeps to integrate over. The hope would be to use a vertical transect to characterize the strength and dominant spatial scale of the turbulence as a function of height then combine that with the system/scan parameters to come up with a normalized variable "number of turbulent scale lengths per sweep" as a function of height. The hope would be that we could use that profile and your results to determine the uncertainty in the horizontal wind fit as a function of number of sweeps integrated. . . Another set of variables that could be combined – SNR, beam PRF, and LOS vel uncertainty (using the CRLB discussed below). Once the results are expressed in these terms, they are no longer only applicable to the system defined in the study. At 50 pages, this is a long manuscript and the length may limit its impact and applicability. Much of the wind profiling theory section, evaluation of errors section (definitions) could be moved to appendices. Some concerns/questions on the approach taken in the study (some of these may be redundant with first section) 1. Vertically averaging/combining results when you have a height dependent footprint combined with a height dependent turbulence profile / integral scale. Are each of the 9600 profile points from different distances below the AC? If so – how does the dominant spatial scale of the turbulence compare to the spatial scale of the sample volume/arc? (ie "number of independent samples" within a scan arc).

2. Sensitivity to static errors in pointing/orientation – does this just fall within the RMS of the assumed uncertainty in pointing or does a static offset impact the fit differently?

3. Dependence on one set of operational parameters (one lidar design) coupled with non-physical simplifications (constant uncertainty in vel as a function of height) is problematic. Need to find a set of independent variables that the user can calculate for their system / scan design.

4. Break analysis into single dimensions where possible and describe the underlying process if you have been able to glean that from the analysis so the user can project

the result into their operational space. 5. Describe mechanism behind bias mentioned in paper in more detail.

Specific Comments Pg 1 Line 10: "laser system noise" - detection uncertainty (see Cramer Rao Lower bound- CRLB) "beam pointing inaccuracy due to system vibrations" – We have found dominant "pointing" errors come from Inertial Navigation Unit (INU) orientation uncertainties. Most concerning are bias errors rather than RMS from vibration. Pg 1 Line 11: "system setups" – define first use Pg 1 Line 16: "short horizontal averaging distances" Along track vs cross track? Pg 2 Line 9: "the vertical wind through nadir " - the vertical wind with nadir Pg 2 Line 20: "while neglecting the non-standard beam geometry" poorly worded / not clear Pg 2 Line 22: anelastic -> an elastic Pg 4 Line 13: "collinearity in model geometry" you mention this on Pg 2 in terms of the matrix inversion, but given that this is a primary research question, you should describe more fully. This will allow the reader to better interpret the phrase at the end of this sentence. Pg 4 Line 30: "5 x 5 x 1.8 km" I am concerned whether this will allow for sufficient independence in the multiple sampling configuration described at Pg 6 Line 10 (see comment below). By using one small set of data and varying the sampling heading angle of the plane through the domain, all paths share a common volume in the center and are not completely independent. This will impact each height differently due to sampling footprint as a function of distance below plane. I assume that the dominant turbulent spatial scales at each height are also different – so this further complicates the interpretation.

Pg 5 Line 5: "The boundary layer height is approximately. . ." At first consideration, it would seem that an important quantity would be the relationship between the integral scale and the scan volume/sample arc length - how many "integral scale" lengths are averaged over in one scan/retrieval. Do you vary this relationship and, if so, is it done by changing the input wind field or only the scan configuration? After my initial reading, it seems you are averaging your results in height, so you may be diluting the effect by averaging over multiple conditions in height. As mentioned in the prior comment, you

have both height dependent sample volumes and turbulent length scales.

Pg 6 Line 11: "As the different flight tracks lead to different air mass volumes being sampled, the different flight directions are independent and can be used to increase the sample size of the statistical analysis" This is the phrase I was basing the Pg4 line 30 comment about independence.

Pg 6 Line 27: "For the frozen-in-time wind field, the LES coordinate" Are you considering atmospheric features that might be correlated to the ground - ie land usage or topographic effects? Making measurements in complex terrain might lead to making measurements in where coherent features in the vertical wind field might be present .

Pg 7 Line 29: "direction noise" Several issues to consider here: 1) While it may end up being a similar effect in the end our experience is that the estimation of orientation angles p,r,y and their angular rates are more prone to error then reading an encoder on the scanner. 2) The uncertainty in pointing angle also feeds into the LOS platform motion correction algorithm which then feeds back into the wind profile. 3) The static/low frequency component of the pointing offset is more problematic than a zero mean Gaussian noise source on the pointing. Depending on the inertial navigation unit (INU), errors in the drift correction of the sensor can lead to low frequency/static errors. 4) Latency in the communication with INU sensor and fast scanning/beam rates can lead to static offsets in orientation/angular rates estimation.

While you can use the ground to "calibrate" the static angle offsets – time varying, low frequency effects can still be present in the data. Upward looking scans, cloud cover, operation over water could all lead to periods where ground strikes are not available. Being able to quantify the sensitivity of these errors propagated into the wind profiles would be great.

Pg 8 Line 15: "detector noise, speckle effects and turbulence within the measurement volume" In areas where there are adequate aerosol, the dominant mechanism for uncertainty in the Doppler measurement comes from uncertainty in estimating the

spectral peak (take a look at Frehlich, Coherent Doppler lidar measurements of winds in the weak signal regime). CRLB depends on SNR, pulse width, and averaging time – (beam rate, vertical resolution, distance from AC) these are quantities that are part of the trade space when designing a scan/sample strategy.

Within a beam you will get a range of SNR and hence uncertainty in the LOS Vel measurement (as a function of range). Setting up your runs with a single velocity error for all ranges is not representative of a true measurement. This effect is only exacerbated when you add a variable aerosol field as a function height as well.

Pg 8 Line 27 "System components" (and used elsewhere) this is an ambiguous term. In this case it would seem that you are referring to pointing vectors – but elsewhere you seem to have different meaning.

Pg 9 Line 6 "aircraft center of gravity…" The moment arm should be between the location of the INU and the center of the final turning mirror in the scanner.

Pg 10 Line 2 "restore" do you mean isolate?

Pg 10 Line 4 "system noise" No matter how well you apply the alignment calibration, there will always be a non-zero static offset. You should consider a sensitivity analysis of the wind profiles to static errors in the orientation. (rather than always assuming the errors are zero mean.)

Pg 10 Line 34 "horizontal wind profile" It can remove the effect of a static profile, but not the natural variability. If you are trying to measure $w'^2$ – the variability in the horizontal wind will still manifest in the $w'^2$ profile.

─────────────────────

---

## Author Response (AR1)

Dear Referee #2,

We thank you for the review of our manuscript and your detailed remarks. We have revised the manuscript, considering your remarks and those from the other referees. The revision led to a substantially changed and shortened version of the manuscript.

In order to allow a focus on the most important results we now reduced the content to the introduction of the LES-based airborne Doppler lidar simulator. Following this, illustrating the benefits of this new tool, the error in wind profiling due to violation of AVAD retrieval assumptions by boundary layer turbulence is investigated for a standard system setup and retrieval strategy. The result sections on system setup and retrieval strategy characteristics are not part of the manuscript anymore due to their lengthy nature as well as comments by other referees.

Please find our answers to your major and specific comments below.

Many thanks for your work so far and kind regards,

Philipp Gasch and Co-authors

Initial, short replies are marked in this color

The final response addition is marked in this color

Major comments:

(1) The writing style, English and grammar needs work and many comments are listed below. I stopped after several pages because it was taking too much time. Significant re-writing and organizational changes are needed in the manuscript. I also think the paper is much too long. It appears that this journal does not have a page limit, but it would help readers to shorten the discussion in several places and remove sections that are not needed (some examples given below). The paper reads a bit like a dissertation with too much background and drawn out detail. A published paper should be more concise without sacrificing understanding of the problem. Please shorten the results section, it looks like too much information is presented and it might not be worthy to publish all of it.

We are sorry to hear that the style, English and grammar did not suit your tastes. We are surprised about the comments about the English, as one of our co-authors is a native speaker and carefully proofread the manuscript. We are therefore very sorry for the inconsistencies that have slipped our attention.

Regarding the shortening: The manuscript was designed for publication and not as a dissertation. Due to the many aspects considered we felt that some background information might be helpful for some readers.

We have considerably shortened the manuscript now in order to be more concise. Our native speaker co-author has carefully proofread the revised manuscript version again.

 (2) Throughout the manuscript the word "homogeneous" and "inhomogeneities" are used and this represents a critical aspect of the study and results. For example, "...mismatch between assumed homogeneous wind field models and the wind field inhomogeneities during the measurement process". These are ambiguous terms and I don't understand how they are being used in this context. Since they represent critical points of the paper, it is hard for me to assess the method and results. The authors need to lay out in detail what they are referring to here and clarify this throughout the manuscript. Are you talking about wind variability below the scale of the instrument footprint, grid spacing of the wind retrievals, something else?

In this study we are concerned about errors in wind profiling that arise from the fact that the analysis method assumes that the flow is homogeneous (i.e. constant mean wind profiles), whereas an actual boundary layer experiences flow inhomogeneities driven by turbulence, such as discussed in Shapiro and Fedorovich (2007), Kiemle et al. (2011) and Lundquist et al. (2015).

By the term "wind flow inhomogeneities" we refer to deviations of the wind speed from the mean state due to boundary layer turbulence. In order to consider only turbulent conditions, we limit the extent of our analysis domain to 0-800 m vertically (inside the boundary layer) in Sec. 4. The structure of the turbulence present in the LES is further detailed in our answer to remark (3). The range of the wind variability is presented by the spectra of the turbulence (fig. 5.8 in Stawiarski, 2014). The dominant range of the turbulence spectra is larger than the instrument footprint (10 cm beam diameter, 300 ns Gaussian pulse width, 72 m range gate length), enabling an accurate representation of the turbulence in the simulated measurements. The dominant range of the turbulence spectra is smaller than the horizontal grid spacing of the retrieval (1.3x1.9 km) but larger than the vertical spacing (60 m). This spatial mismatch clearly influences our results in Sec. 4 and is discussed there.

The term homogeneous flow (a synonym is homogeneous wind field conditions) means a uniform flow and the absence of any deviations from the mean flow state, both spatially and temporally (Stull, 2000). Homogeneous flow is assumed in the most simple form of the Velocity Azimuth Display (VAD) and Volume Velocity Processing (VVP) retrievals (Koscielny et al. 1984, Boccippio, 1994, Banakh et. al, 1995, Leon and Vali, 1996). Homogeneous flow is rarely present in the real atmosphere and certainly not inside the boundary layer. The violation of the homogeneity assumption in the VAD/VVP retrieval due to boundary layer turbulence causes an error in the retrieved wind profile. This error is the focus of our study.

We have clarified the meaning of the term 'inhomogeneity/inhomogeneous flow' as well as 'homogeneous flow' in the revised version of the manuscript. This includes rewording the manuscript in many places in order to clarify and stress that we are concerned about wind profiling errors due to boundary layer turbulence violating the AVAD assumptions. We have included an illustrative

figure, in which the presence of flow inhomogeneities is clear (fig. 4), and another one, in which the resulting error is visible (fig. 5)

(3) The LES domain size of 5 km X 5 km X 1.8 km is extremely small and I have doubts that this domain will represent a realistic environment to test the lidar wind sampling. The authors state that a single flight through the LES does not yield sufficient statistics. However, making 25 different aircraft trajectories through a very small box, probably does not generate any real independent statistics since the retrieved winds are sampling almost the same flow (the decorrelation spatial scale is probably larger than the box itself). It appears the grid spacing of the wind retrievals might be 1.3 – 1.9 km for along/across track. Given this spacing, I don't think the authors can generate independent flight tracks and statistics through a 5 km X 5 km domain. The authors should try their simulations with a flow in a larger domain (with coarser resolution) in order to study a more realistic environment and allow for independent statistics.

While we appreciate the reviewer's request for a larger domain, applying coarser resolution would actually undermine the attempt to represent fully developed turbulence. We are confident of the decorrelation spatial scale, as the integral length scale (Tennekes and Lumley, 1972, Lenschow and Stankov, 1986) of the turbulence present in the LES is $L_i < 500$ m, as stated in section 2.1. Further documentation of the decorrelation scale of the turbulence is provided in the attached fig. 6.3 and fig. 6.6 from Stawiarski (2014) below.

As we constrain the analysis to heights < 800 m (above which gravity waves are present, increasing the integral length scale), the assumption of statistical independence is valid for our analysis volumes of 1.3x1.9 km, which are much larger than the integral length scale (keeping in mind that turbulence is the driver of profiling error which we are interested in).

At the given spatial and temporal integral length scales, we argue that the 16 flight directions, repeated at 1-minute spacing and with a random initial profiling offset, do sample independent air masses. The spatial distance of the wind profiles is larger than the integral length scale. The temporal spacing is on the order of the integral time scale, but aided by decorrelation through advection between subsequent wind profiles.

Unfortunately, we think a coarser model would be problematic to investigate the errors due to turbulent structures in the boundary layer. In our study the grid resolution is 10 m (corresponding to a resolution > 50 m). Thereby, Doppler lidar measurements at a measurement frequency of 1 Hz, with a range gate length of 72 m and a flight speed of 65 m s$^{-1}$ can be realistically represented, as the resolution corresponds to the sampling rate and spatial extent of the range gates. In a coarser model, the high sampling rate would not produce independent measurements. Further, the structure of the turbulence, causing the wind profiling error we are interested in, would be less accurately represented.

Please note that our approach follows established methods. These LES wind fields have been used for Doppler lidar studies in a similar way before

(Stawiarski, 2014, Stawiarski et al., 2015) and that other authors choose similar approaches for ground-based studies (Scipion et al., 2009, Scipion, 2011).

We will make sure to discuss the independence of the acquired data in a more clear way in the revised version of the manuscript.

Sufficient independence of the acquired data is an important point. Therefore, we are addressing the concerns about independence of the sample data in-depth in the revised manuscript version in-depth (Sec. 3.2). To this end, we have changed the sampling strategy to only include profiles which are sampled with spatial or temporal independence. This is done by using a checkerboard approach, which is discussed and illustrated in the manuscript now. Further, the independence of the sampled wind profiles is now checked using statistical methods including autocorrelation.

Specific comments (some major, some minor):

Line 2: should say "...additional insights **relative** to ground-based systems..." Included.

Line 2: what does "spatially resolved" mean here? This term is too ambiguous. Changed to spatially distributed.

Line 3: "...prepares the ground...", is poor English and needs a change. Deleted.

Line 4: spell out LES for the first time used; what is meant by "first"? Note that other studies have used large eddy simulations to study remote sensing instruments. LES is now spelled out. We mean the first LES-based airborne Doppler lidar simulator, which to our knowledge doesn't exist so far.

Line 6: I believe it should be "...wind profiles **in** inhomogeneous flow...". Changed.

Line 7: Need to clarify with numbers what is meant by "acceptable error margins". Acceptable is ambiguous and could mean very different things to different people. Changed.

Line 7/8: sentence that starts with "Results allow for determination..." should be removed. This is an obvious outcome of the simulations. Changed.

Line 8: What is meant by "flow inhomogeneities"? Seems like this is key since much of the manuscript mentions this, but again, this term seems ambiguous to me. As stated above, we tried to clarify what we mean by flow inhomogeneities.

Line 16: What is meant by "short horizontal averaging distances"? Changed.

Line 3: Need a comma after "benefits". Included.

Line 11/12: "considering both wind profiling and nadir measurements of the wind field"; I don't understand the difference between these two things as it is written. Changed.

Line 13: What is meant by "mean" horizontal flow? Average over space/time and what scales? Changed.

Line 18: I don't understand how winds are retrieved through "inversion of the beam matrix", this sounds like an incorrect statement or writing error. Winds are retrieved by inverting the least squares fit between the model and observations. It is crucial for reader to understands how the inversion process works as this is the basis of the AVAD retrieval. Therefore, this statement is now moved to the section explaining the retrieval process in order to provide more context. We follow the notation of Leon and Vali (1998). What is inverted (in our case using a SVD decomposition) is the beam matrix G, yielding a least-square solution to the problem. The LSQ-fit is the result of the inversion process (to our knowledge, the LSQ-fit cannot be inverted). The LSQ-fit is obtained by multiplying the beam matrix with the estimated wind vector, obtained by the inversion. We have included fig. 4, illustrating the input truth wind field as well as the retrieval procedure and LSQ-fit. We quote Leon and Vali (1998, p. 865):

"The matrix of beamvectors (for the selected form of the velocity field) is then inverted using a singular value decomposition (Bevington 1969; Menke 1989), such that

$$\mathbf{V_{est}} = \mathbf{G^{-g}} \, \mathbf{V_{Dopp}} \, . \qquad (19)$$

The estimated velocity field parameters can then be used, together with the matrix of beamvectors, to produce an array of predicted Doppler velocities:

$$\mathbf{V}_{\mathrm{Dopppre}} = \mathbf{G} \, \mathbf{V}_{\mathrm{est}}. \qquad (20)$$

Comparison between the predicted and actual data values can reveal how well the wind fields have been fit and the appropriateness of the velocity field form."

Line 25: "...assume homogeneous conditions throughout the sample volume.", what is meant by this statement? One can't measure things that are sub-grid-scale, but I don't understand what this is referring to. Please see our answer to your major comment 2 for an in-depth answer on this. We assume that you are referring to the retrieval volume when using the term sub-grid-scale. Sub-retrieval volume flow inhomogeneities can introduce error into the retrieved wind vector when using AVAD (as AVAD assumes homogeneous flow throughout the retrieval volume). As you state, the inhomogeneities cannot be retrieved by the mean wind vector obtained as a solution. Therefore, we are not trying to measure sub-retrieval volume processes. However, deviations from the assumed mean flow state on scales smaller than the retrieval volume will introduce error into the AVAD retrieval. This is the error which we are concerned about. The advantage of the ADLS is that the sub-retrieval volume inhomogeneity is exactly known, because the LES is used as a known input wind field, which is impossible in reality. Therefore, we can quantify the error due to sub-retrieval volume flow inhomogeneities.

We have tried to give more context by referring to boundary layer turbulence directly, as well as rewording.

Line 28: Statement about how high elevation angles are used to constrain the footprint. Tilt angles closer to nadir will provide a shorter slant path and thus smaller footprint, but there are other reasons for choosing this steep tilt. Some things could be hardware limitations, range limitations and attenuation. We agree and have included your suggestion.

Line 29: I think it should be "measured radial velocity" and not "retrieved radial velocity". The process of obtaining a radial velocity also involves a process termed retrieval by some researches (e.g. spectral peak estimation), which is why we chose this wording. We now reworded according to your suggestion in order to avoid confusion.

Line 34: I am still confused on what is meant by "homogeneity assumption". As stated above we have tried to clarify and explain.

Lines 15 – 17: Sentence is too long, need to break up for clarity. Done.

Line 24: Need a comma after "capabilities". Done.

Line 25: Need a comma after "systems". Done.

Line 29: "challenged" should be "challenges". Done.

First paragraph: I don't understand what "assume homogeneous wind field and inhomogeneities during the measurement process" means in this context. Need to make significant changes to clarify this and possibly make a diagram to illustrate what this is referring to. We reworded in order to clarify in combination with the above changes. In addition, as suggested by you, we have included fig. 4. Fig. 4 illustrates the input truth wind field as well as the retrieval procedure. It also includes an illustration of the flow inhomogeneities due to turbulence, which we are concerned about.

Line 16: Need a period after "follows" instead of a colon. Done.

General question: what is this a large eddy simulation of, homogeneous, isotropic turbulence? The domain size of 5 km X 5 km X 1.8 km is extremely small and I have doubts that this domain will represent a realistic environment to test the lidar wind sampling. The authors state that a single flight through the LES does not yield sufficient statistics. However, making 25 different aircraft trajectories through a very small box, probably does not generate any real independent statistics since the retrieved winds are sampling almost the same flow (the decorrelation spatial scale is probably larger than the box itself). The authors should try to find a simulation with a larger domain (with coarser resolution) to test the sampling and/or mention that the results of this study are limited to very idealized flow conditions. Please see our answer to your major comment number 3.

Line 19: 65 m/s seems like a low speed to me; what type of aircraft is this instrument targeted for? This is a low speed indeed, the full sentence reads 'The aircraft speed relative to air (IAS) is set to 65 m s$^{-1}$, representative for a medium-range turboprop aircraft at measurement speed'. The system is intended for use aboard a medium-range turboprop aircraft, the Dornier 128-6 (D-IBUF), which measures at such a speed (Corsmeier et al., 2001).

Lines 20 – 25: regarding the difference between aircraft heading and ground track...I assume you are talking about drift here. Note that Guimond et al. (2018) found an error in the Lee et al. (1994) mapping equations for Earth-relative coordinates, which don't contain a correction for drift. If you are incorporating drift into your mapping coordinates, this correction should be applied. Guimond, S.R., J.A. Zhang, J. Sapp and S.J. Frasier, 2018: Coherent turbulence in the boundary layer of Hurricane Rita (2005) during an eyewall replacement cycle. *J. Atmos. Sci.*, **75,** 3071- 3093. We are talking about drift indeed. The error reportedly found by Guimond et al. (2018) does not affect our simulations, as the unbiased wind direction retrieval results show. We base our equations on the approach by Leon and Vali (1998). We have removed an erroneous reference to Lee et al. (1994) and replaced it with the correct one to Lenschow (1972). We have also adjusted our notation by switching the signs in eq. (1-3) in order to be fully consistent with them.

Regardless of the above, in this paragraph, we are concerned about the change in sampling distance in the LES due to drift. As the change in sampling distance due to drift is important, we have reworded and explained with an example in order to make this clear.

Line 25: I don't understand this sentence and the bold claim that this is the "first presentation of a correct airborne sampling simulation", please explain more clearly. Due to the general shortening, we have removed this statement. Nevertheless, we are not aware of any discussion of the effect of drift on the sampling spacing when simulating airborne measurement systems (if you are, please let us know). This is an important effect, and we believe this is the first description on how to take it into account in simulations.

Top half of page: I am confused with this section and Appendix A1. The aircraft position (lat,lon,height) are provided by the GPS on any aircraft and the mapping equations provide locations of the pulse volume centers relative to these positions. Can't the authors just generate a realistic aircraft position vector (possibly from real data) and sample the model winds with that? This extra stuff seems irrelevant. If there is a large aircraft head wind, then the plane might only go forward very slowly and the wind retrievals would only cover a small region. In reality head winds are usually very small relative to the aircraft speed so I don't understand the motivation to get into all this detail. Just use a realistic aircraft position vector because ultimately this will be applied to real situations. We have significantly shortened this paragraph in order to make it more clear. Unfortunately, due to the effect of wind on aircraft track development, real GPS aircraft data should not be used in the ADLS for the time varying wind field (unless it would stem from a flight with the exact same wind field conditions and flight direction).

Further, due to the slow measurement speed of the aircraft, the effect of wind speed on track development is not small. We now illustrate the magnitude of the effect with a small calculation example in the revised manuscript.

Line 26: Is the 20-second full circle scan time (3 revolutions-per-minute) the lidar scan rate used in the remainder of the study? So with the aircraft speed of 65 m/s, the along-track spacing is 1.12 km? Make these parameters more clearly stated in the paper. Also, what is the grid spacing of the wind retrievals? In the revised, simplified study we are now only using the 20-second full circle scan time in order to avoid confusion. This gives an along-track spacing of 1300 m (20 s*65 m s$^{-1}$ = 1300 m). We are now illustrating the system setup and retrieval procedure, as well as the new checkerboard approach in fig. 6.

Section 2.4 Retrieval – nadir as an example application: I suggest removing this section. The paper is getting much too long and this method has little practical use. Also, the authors say that "wind profiling" is the focus of this study rather than the nadir method. We have significantly shortened the manuscript. In our opinion, this section provides an illustrative example of the ADLS sampling, as well as the turbulence present in the LES. Due to the significant shortening in other parts of the manuscript we would therefore like to keep the nadir section as part of the manuscript as it is.

Line 15/16: "The model is given by the beam geometry...". As stated, this is incorrect, the model is the radial velocity equation, which includes the beam geometry. We have reworded this, please also see our answer regarding the LSQ-fit procedure and the terminology followed above.

Line 22: "...the wind field is usually assumed to be homogeneous...". Again, I don't understand what you mean by homogeneous here. See major comments. Please see our answer to your major comment 2, as well as our answer to your previous comment on this above.

Equations (8) and (9): The matrices U, S and W are not defined so I have no idea how they are used. It is hard for me to evaluate this paper without proper identification of variables. We are sorry that introduction of these variables slipped our attention, we are now identifying these variables (they are based on the standard SVD procedure).

[Figure]

Figure 6.3.: Development of the spatial autocorrelation in the LES $u$ wind fields with the background wind, $u_G = \{0, 5, 10, 15\}$ m/s from left to right.

[Figure]

Figure 6.6.: *Integral length scales*: Scales $L_x$ of the $u$ (left) and $v$ (right) wind fields in mean wind direction for the time series, computed with the full mean and variance of the data set (light green) and the mean and variance of the respective series (dark green). The black bars show the comparative LES results. The bars cover the range between the 25th and 75th percentile, the circles mark the median.

Dear Referee #3,

We thank you for the review of our manuscript and your detailed remarks. We have revised the manuscript, considering your remarks and those from the other referees. The revision led to a substantially changed and shortened version of the manuscript.

In order to allow a focus on the most important results we now reduced the content to the introduction of the LES-based airborne Doppler lidar simulator. Following this, illustrating the benefits of this new tool, the error in wind profiling due to violation of AVAD retrieval assumptions by boundary layer turbulence is investigated for a standard system setup and retrieval strategy. The result sections on system setup and retrieval strategy characteristics are not part of the manuscript anymore due to their lengthy nature as well as comments by other referees.

Please find our answers to your major and specific comments below. Many thanks for your work so far and kind regards,

Philipp Gasch and Co-authors

Initial, short replies are marked in this color

The final response addition is marked in this color

Major comments:

1. The LES domain is too short for this type of study. In the 5 km x 5 km x 1.8 km domain, the authors only use 2 points within to do the analysis for a given trajectory. And approximately 20 trajectories are used in their analysis. The authors also mention that they do not have enough statistics, which is a big concern in such a study. The authors base their methods based on Stawiarski 2014 2015, which is for a ground-based Doppler Lidar and who were looking at detection of coherent structures and the ability of Lidars to resolve those structures. But here, the authors are trying to assess the accuracy of the in-homogeneity, and more simulations and larger domains are a necessity. I can understand that the authors want to have high-resolution LES model data to validate their simulator, but statistically these results cannot be taken seriously. Again, it is very well known that flow in-homogeneity causes errors in Doppler Lidar retrievals, but quantifying the amount for airborne Doppler Lidar measurements is key. And the reviewer feels, that the quantification part is missing here, which is key.

It seems that a misunderstanding occurred. Because the error due to flow inhomogeneity is a well-known issue, we do use the simulator to quantitatively assess this error. (We are not trying to validate the simulator and/or assess the accuracy of the inhomogeneity, further we don't state that our statistics are insufficient). To our knowledge, the LES-based analysis is new, which is why we present the first LES-based airborne Doppler lidar simulator.

In the study we are using 16 different flight directions (called trajectories by you) through the LES domain. Each flight direction yields 2 retrieved wind profiles. The 16 flight directions are repeated 25 times at 1-minute temporal spacing (the wind field is

not assumed to be frozen during each passage but evolving instead). Thereby, this procedure yields 800 independent wind profiles for each of the four background wind cases (3200 wind profiles overall). Additionally, each of the wind profiles consists of 12 wind speed retrievals at different vertical levels (these 12 wind profile *points* of each wind profile are indeed correlated, as they are in reality), giving a total number of 9600 wind profile points for each background wind case (38400 wind profile points overall). If you are concerned about the spatial and temporal independence of the wind profiles we would like to refer you to our short reply to referee #1.

The 3200 independent wind profiles (38400 retrieved wind profile points) are much more than what is typically used in real world measurement comparisons (e.g. 1612 wind profiles in Weissmann 2005, approx. 10 wind profiles in DeWekker 2012, a single profile in Kavaya 2014, 2056 wind profile points in Bucci 2018).

We would thereby like to know what is insufficient about our statistical database and what you would consider a sufficient statistical database for a quantitative error analysis.

We have significantly altered the sampling procedure used to obtain the wind profiles as well as reworded its description. A newly developed checkerboard approach is used and illustrated (fig. 6). Using this approach a sufficient number of profiles are sampled, while maintaining statistical independence through clear temporal and spatial separation. We are also investigating the independence of the sample wind profiles in-depth in the revised manuscript version (Sec. 3.2). Even with the reduced checkerboard sampling approach, our statistical basis is larger than that of other studies reported in literature (who were using real world measurements which are inherently more expensive and complicated to conduct). We are emphasizing the comparably large statistical sample size more clearly in the revised version of the manuscript (we have to correct a small error in our initial answer: Weissmann et al. (2005) used 33 wind profiles for comparison, not 1612).

2. The optical effects of the laser beam due to motion has not been considered in the results. It is just demonstrated that the simulator can do motion correction. It would complicate the analysis and make it more realistic and interesting to study such realistic effects (Kavaya 2014 Hill 2008) rather than only flow in-homogeneity (again its a well known issue). Maybe the authors can take motion data from a real aircraft and simulate the effects of wind retrieval accuracy.

As you noted, we do consider the geometric transformations applicable to real world airborne Doppler lidar measurements. Further, we also investigate the effect of a measurement system imperfection. We do this through an assumed scanner pointing inaccuracy (called beam pointing inaccuracy by us), which is likely to occur in real world measurements. We are thereby able to analyze the effect of an imperfect scanner and/or aircraft inertial navigation system, which is done in Sec. 4.1. Further, we also analyze the effect of an imperfect laser system by adding a random radial velocity fluctuation to the measured radial velocities.

We do not see an advantage in using real world aircraft motion data, as the actual motion of the aircraft does not influence the VVP (not VAD) wind speed retrieval quality (only the accuracy with which the motion is known, which we take into account through the beam pointing inaccuracy).

We would thereby like to know what optical effect of the laser beam due to motion you are referring to.

Following referee comments by Mr. Brewer, we have removed the noisy system simulation from the manuscript. We now focus purely on an assumed ideal measurement system in order to highlight the wind profiling error introduced due to violation of the AVAD assumptions in the turbulent boundary layer.

3. The article has several repetitive statements all over the document and is extremely long. I feel a large portion of this article can be moved to the supplement section or deleted. The sentence formations are very abrupt, which sometimes I feel are unnecessary and not adding any additional information. I have noted a couple below, but I would urge the authors to take a deeper look at the entire article.

We will make sure to have another critical look at the wording of the manuscript. Further, we will also shorten its length.

As stated above, we have significantly revised and shortened the manuscript.

4. The Figures are not very helpful in understanding the results. They all are dumped into one figure with results from several simulations and different type of statistics. Figures 6-16 are very confusing and do not make it obvious without reading the winded text. I can understand the authors sentiment in showing a lot of information, but they need to use smarter approaches such as Taylor's diagram or similar.

We are sorry for the inconvenience. It is very difficult to convey the amount of information we present without a detailed description in the text. We will make sure to have a look at alternative forms of displaying the information as well as a more compact wording.

We have tried to reduce the amount of information we try to convey. The problematic figures mentioned by you are not part of the manuscript anymore.

Specific comments:

Page 5 Line 10 – The flight temporal sampling is not clear. 1-minute temporal sampling, does that mean the aircraft takes 1 minute to travel the 5 km domain? Isn't that a function of the scan trajectory? Its not very clear. Is the wind assumed to be stationary over that 1 minute? We have changed the sampling procedure to a checkerboard approach. We are now illustrating the used checkerboard sampling procedure using figure 6 in order to make the sampling procedure clear. The time it takes to transect the LES domain depends on the distance, the aircraft speed and the wind speed (as drift is taken into account). The wind field is changing while the aircraft is flying through the box, as we are using the time-varying wind field approach.

Page 8 Line 7: Instead of saying "their" weighting function . . . please mention the type of weighting function. Was is a Gaussian or rectangular or? Similarly, other things in the paper. It is a Gaussian weighting function and we now mention this.

Page 8 Line 8 – What is deltaX? Please use lidar technical terms or variables to be consistent with the literature. We have changed the notation to be consistent with Stawiarski et al. (2013) for lidar technical terms.

Page 8: I feel a short table of these characteristics would make it much more legible and easier for the readers. We hope that adapting the notation and rewording is sufficient.

Page 8 line 19: "can be added to the simulated wind field"? Please refine, as it has been added in the flow field, is my understanding. The sentence read: „Therefore, in the ADLS, a Gaussian noise with standard deviation $\sigma_{vD}$ can be added to the simulated measured radial velocities." As stated, the noise was added to the simulated radial velocities. This statement has now disappeared due to the focus on an ideal measurement system.

Page 8 line 9 – What is FME? The acronym comes doesn't make sense with the sentence before. We are not using the acronyms anymore following the shortening of the manuscript. FME was measurement frequency.

Section 2.3 – A motion correction algorithm is introduced, but the author mentions that they have not used any of that in their paper. I feel this section is not useful for this paper, if not used. Please cite a reference, such as Kavaya et al., 2014 or Hill et al., 2009, which has similar motion correction algorithms implemented to real lidar data. If its different that these algorithms, then please state only the difference. Please move this the next section to supplement. We are using the motion correction algorithm in section 2.4 (nadir retrieval), as well as figure 1. Therefore, we would like to keep the statement, but as before we are not presenting the algorithm itself as it is simple.

Page 3, Line 26. . . Remove "These include" and just list the references. Done.

Page 5 Line 6... Gravity waves being present above what? – Sentence structure is wrong! The sentence structures sometimes are too abrupt and unnecessary. So maybe the authors have to give it another read and remove some of the abrupt sentence structures. Such as Page 4, Line 26, it is not necessary to have the first statement "The simulator is tested with a set of underlying wind fields". There are other instances such as this and needs to be addressed. I understand that it's a personal choice of the reviewer. We have significantly rewritten the manuscript and hope to provide a smoother reading now. As you say, this is a matter of personal choice as well.

Page 27 line 11: why bold number of retrieved parameters (VAR)? And similar other locations. Please change. We are not using the acronyms anymore following the shortening of the manuscript.

Since there were too many issues with the paper, I feel like the authors need to revise the paper carefully and provide a better and shorter version for the reviewers. This was done.

Dear Alan Brewer,

We thank you for the careful review of our manuscript and your detailed remarks. We have revised the manuscript, considering your remarks and those from the other referees. The revision led to a substantially changed and shortened version of the manuscript.

As stated previously, we reduced the content of this study to the introduction of the simulator. As you suggested, we highlight the ability of the LES-based simulator tool to investigate wind profiling error due to flow inhomogeneity with a basic example for a standard system setup and retrieval strategy. The simulator is then available for use by the community and specific setups, as suggested by you. We will explore the trade space of an airborne system and the effect of measurement system inaccuracies in detail in a future study.

Please find our answers to your major and specific comments below. Many thanks for your work so far and kind regards,

Philipp Gasch and Co-authors

Initial, short replies are marked in this color

The final response addition is marked in this color

An LES-based airborne Doppler lidar simulator for investigation of wind profiling in in-homogeneous flow conditions This is an excellent tool for evaluating airborne Doppler lidar measurement design and will be an invaluable part of experimental design. The authors have done an excellent job of addressing the key considerations of airborne Doppler lidar sampling strategy.

General comments: It was not clear to me if the paper was designed to illustrate the capabilities of the simulator using a single set of LES and lidar configurations or if the authors are attempting to explore the operational trade space of an airborne system with sufficient detail so that the community could use this paper as a design tool rather than the simulator itself. If the goal of the paper is to illustrate the application of the tool, then I think the authors need not explore as much of phase space in every degree of freedom. A single variable plot accompanied by discussion of the underlying mechanism for the relationship for each of the major impacts would suffice. Concentrate on those plots/relationships that show significant structure and focus there.

It is true that we tried to achieve both, introducing the simulator as well as exploring the trade space of an airborne system, at the same time. We now realize that this is too complex for one study. Therefore, we will reduce the content of this study to the introduction of the simulator. As you suggested, we will highlight the ability of the LES-based simulator tool to investigate wind profiling error due to flow inhomogeneity with a basic example. The simulator is then available for use by the community and specific setups, as suggested by you. We will explore the trade space of an airborne system and the effect of measurement system inaccuracies in detail in a future study.

We have proceeded as outlined in our initial answer. We now focus on the ideal system simulation only, as the noisy system simulation was lacking some aspects which can be important in reality (see answers below). Further, we are not exploring the trade space of an airborne Doppler lidar system anymore, as suggested by you. This is subject to a further study (including an improved noise simulation). Specific questions on system setups and retrieval strategies can be addressed directly by using the simulator after publication.

For example Fig 10 would benefit from another approach that highlights the differences between the two plots (highlighting the effect/impact of noise) and changing structure within a plot (slow increase in RMSE/REL). The actual values of the static variables in the plots are only meaningful for one particular set of conditions (both LES and lidar characteristics) and may not be of interest to a wide range of folks. How certain variables change wrt the chosen independent variable is of interest as it will generally do so for all configurations. These relationships should be highlighted and the underlying mechanism explained if possible. If the goal of the paper is to probe the trade space of all systems/conditions, then I think some effort should go into finding fewer "normalized" variables that are independent of specific choices for system values or LES characteristics. For example – my understanding is that the underlying bias sensitivity of the horizontal wind fit to coherent structures in the vertical wind field comes down to how many "pairs" of these features you average/accumulate over prior to doing a fit. As you increase the number of pairs of up/down motions in the volume – the residual of the fit may have a higher RMS, but the bias will decrease. This condition will depend on the dominant spatial scale of the turbulence, the size of the sample volume, the density of the sample points. . . A potential "normalized" variable might be the number of independent turbulent scale lengths per sample volume. For a given turbulence profile in the LES and a given scan pattern and beam PRF, one could imagine a profile of this "normalized" variable and corresponding uncertainty in the fit. . . If the authors are able to combine variables and break the analysis into the underlying mechanisms, it may serve to widen the impact of the paper and reduce the number of variables that have to be studied. I'm happy to discuss directly with the authors if need be – I'll ask the editor to share my contact information. For our application, we have a fixed scan geometry (wedge scanner) so the only system variables we have to adjust are scan rate, beam PRF & pulse width. The adjustable fit parameters are height resolution and number of sweeps to integrate over. The hope would be to use a vertical transect to characterize the strength and dominant spatial scale of the turbulence as a function of height then combine that with the system/scan parameters to come up with a normalized variable "number of turbulent scale lengths per sweep" as a function of height. The hope would be that we could use that profile and your results to determine the uncertainty in the horizontal wind fit as a function of number of sweeps integrated. . . Another set of variables that could be combined – SNR, beam PRF, and LOS vel uncertainty (using the CRLB discussed below). Once the results are expressed in these terms, they are no longer only applicable to the system defined in the study. At 50 pages, this is a long manuscript and the length may limit its impact and applicability. Much of the wind profiling theory section, evaluation of errors section (definitions) could be moved to appendices.

With the limitation of this study to the basic simulator and method description we cannot include more general findings, using normalized variables as suggested, in this study. Nevertheless, your remarks make total sense and we are happy to discuss these aspects directly with you in the near future. As you proposed, based on the simulator,

we have already developed a method to estimate the uncertainty of the retrieved wind profiles in-situ without the need for additional data except for some basic boundary layer parameters. Unfortunately, including this method in the study would be beyond the scope of this manuscript as it is too long already, but we are happy to share it with you immediately.

We are now focusing on a basic example as stated above. Exploration of the system trade space as well as generalization and uncertainty estimation will be investigated in a further study. According to your suggestion, we have moved many sections to the appendices.

Some concerns/questions on the approach taken in the study (some of these may be redundant with first section)

1. Vertically averaging/combining results when you have a height dependent footprint combined with a height dependent turbulence profile / integral scale. Are each of the 9600 profile points from different distances below the AC? If so – how does the dominant spatial scale of the turbulence compare to the spatial scale of the sample volume/arc? (ie "number of independent samples" within a scan arc).

The 9600 profile points are from different altitudes below the aircraft. As you say, they were determined using height dependent footprints as well as turbulence profiles in order to achieve a spread among these important input variables.

With our proposed new uncertainty estimation method the average ratio of the dominant turbulence scale compared to the scale of the sample volume is taken into account by calculating an effective sample size. Therefrom, the uncertainty is determined. Unfortunately, this is too complicated to include in this study, but as stated before we are happy to share it with you immediately.

We have now simplified the retrieval approach to the checkerboard sampling procedure. We hope that this make the sampling and retrieval more understandable. The uncertainty forecast method is prepared and will be published at a later point.

2. Sensitivity to static errors in pointing/orientation – does this just fall within the RMS of the assumed uncertainty in pointing or does a static offset impact the fit differently?

As you state below, our error emulation does not capture all aspects of errors encountered in real world systems. The effect of a static error in pointing/orientation depends on its origin, whether it is a static offset in the scanner pointing or the INS orientation. In general, it should lead to increased error levels and potentially a bias depending on the flight and wind direction. Due to the complexity of the situation and our incomplete emulation we will remove this part from the manuscript and investigate more in-depth in the discussed further study.

As suggested in our preliminary answer, we have removed the noisy system simulation from the manuscript due to the over-simplification as well as complexity. Possible system errors require further study, as there is a multitude of possible error sources and the associated influences are complex.

3. Dependence on one set of operational parameters (one lidar design) coupled with non-physical simplifications (constant uncertainty in vel as a function of height) is problematic. Need to find a set of independent variables that the user can calculate for their system / scan design.

As stated above, this will be the subject of a future study. Nevertheless, we are willing to share the method developed with you immediately.

As stated above, we have removed the noisy system simulation from the manuscript. Any interested user can use the simulator directly to answer specific questions related to their system / scan design.

4. Break analysis into single dimensions where possible and describe the underlying process if you have been able to glean that from the analysis so the user can project the result into their operational space.

The system trade space analysis will be removed from the manuscript and subjected to further study.

As the exploration of system trade space was removed from the manuscript we are happy to investigate this for specific user setups if needed.

5. Describe mechanism behind bias mentioned in paper in more detail.

The mentioned possible bias at low wind speeds will remain in the revised manuscript, as this is a new and important finding. We will explain in more depth.

The mentioned potential bias at low wind speeds is an important new finding of this study. We are now illustrating an AVAD LSQ-fit for the no-background wind speed case (fig. 4). This figure serves to illustrate the mechanism responsible for erroneous mapping of vertical wind into horizontal wind.

We will respond to the specific comments below in our final response together with the revised manuscript. Thank you once again for all the effort invested.

Specific Comments Pg 1 Line 10: "laser system noise" - detection uncertainty (see Cramer Rao Lower bound- CRLB) "beam pointing inaccuracy due to system vibra-tions" – We have found dominant "pointing" errors come from Inertial Navigation Unit (INU) orientation uncertainties. Most concerning are bias errors rather than RMS from vibration. As important error aspects were missing for the noisy system simulation we have removed this part of the study in order to extend research into these topics and publish them at a later point.

Pg 1 Line 11: "system setups" – define first use. This was reworded.

Pg 1 Line 16: "short horizontal averaging distances" Along track vs cross track? This was removed, we are now using equal along- vs. across-track averaging distances.

Pg 2 Line 9: "the vertical wind through nadir " - the vertical wind with nadir. This was removed due to the shortening of the manuscript.

Pg 2 Line 20: "while neglecting the non-standard beam geometry" poorly worded / not clear. This was moved to Sec. 3.1 and 'non-standard' replaced with 'altered'.

Pg 2 Line 22: anelastic -> an elastic. They actually use the anelastic mass continuity equation.

Pg 4 Line 13: "collinearity in model geometry" you mention this on Pg 2 in terms of the matrix inversion, but given that this is a primary research question, you should describe more fully. This will allow the reader to better interpret the phrase at the end of this sentence. We are not focusing on collinearity filtering using the condition number CN anymore (pseudo difficult conditions through sector blanking), but rather on filtering by the correlation coefficient $R^2$ in low wind speed situations. We have reworded accordingly.

Pg 4 Line 30: "5 x 5 x 1.8 km" I am concerned whether this will allow for sufficient independence in the multiple sampling configuration described at Pg 6 Line 10 (see comment below). By using one small set of data and varying the sampling heading angle of the plane through the domain, all paths share a common volume in the center and are not completely independent. This will impact each height differently due to sampling footprint as a function of distance below plane. I assume that the dominant turbulent spatial scales at each height are also different – so this further complicates the interpretation. We have substantially changed the sampling procedure in order to ensure spatial and temporal disjunct sampling. Further, we are now investigating the independence of the acquired wind profiles (Sec. 3.2).

Pg 5 Line 5: "The boundary layer height is approximately. . ." At first consideration, it would seem that an important quantity would be the relationship between the integral scale and the scan volume/sample arc length - how many "integral scale" lengths are averaged over in one scan/retrieval. Do you vary this relationship and, if so, is it done by changing the input wind field or only the scan configuration? After my initial reading, it seems you are averaging your results in height, so you may be diluting the effect by averaging over multiple conditions in height. As mentioned in the prior comment, you have both height dependent sample volumes and turbulent length scales. Indeed, the relationship between the integral scale and the scan volume/sample arc length is important. In our study it is inherently varied over a wide range by using retrievals from different heights below the aircraft, as well as the changing boundary layer structure and the different wind cases (fig. B4). With the newly developed uncertainty forecast method this parameter is accounted for and estimated.

Pg 6 Line 11: "As the different flight tracks lead to different air mass volumes being sampled, the different flight directions are independent and can be used to increase the sample size of the statistical analysis" This is the phrase I was basing the Pg4 line 30 comment about independence. We refer to our answer there.

Pg 6 Line 27: "For the frozen-in-time wind field, the LES coordinate" Are you considering atmospheric features that might be correlated to the ground - ie land usage or topographic effects? Making measurements in complex terrain might lead to making measurements in where coherent features in the vertical wind field might be present.

We are not considering features correlated to the ground at the moment, as the LES is using a homogeneous, constant surface heat flux. We agree, this effect is very important to consider when making measurements in complex terrain and/or different land usage scenarios. It definitely requires further study but unfortunately is beyond the scope of this study.

Pg 7 Line 29: "direction noise" Several issues to consider here: 1) While it may end up being a similar effect in the end our experience is that the estimation of orientation angles p,r,y and their angular rates are more prone to error then reading an encoder on the scanner. 2) The uncertainty in pointing angle also feeds into the LOS plat- form motion correction algorithm which then feeds back into the wind profile. 3) The static/low frequency component of the pointing offset is more problematic than a zero mean Gaussian noise source on the pointing. Depending on the inertial navigation unit (INU), errors in the drift correction of the sensor can lead to low frequency/static errors. 4) Latency in the communication with INU sensor and fast scanning/beam rates can lead to static offsets in orientation/angular rates estimation. Because of this comment, as written above, we have removed the noisy system simulation from the manuscript due to the over-simplification as well as complexity. We agree, that all other important effects mentioned by you can occur and need to be investigated. Due to the multitude of possible system errors this requires further study, as the associated influences are complex.

While you can use the ground to "calibrate" the static angle offsets – time varying, low frequency effects can still be present in the data. Upward looking scans, cloud cover, operation over water could all lead to periods where ground strikes are not available. Being able to quantify the sensitivity of these errors propagated into the wind profiles would be great. We will investigate the effects of system noise in a further study.

Pg 8 Line 15: "detector noise, speckle effects and turbulence within the measure- ment volume" In areas where there are adequate aerosol, the dominant mechanism for uncertainty in the Doppler measurement comes from uncertainty in estimating the spectral peak (take a look at Frehlich, Coherent Doppler lidar measurements of winds in the weak signal regime). CRLB depends on SNR, pulse width, and averaging time – (beam rate, vertical resolution, distance from AC) these are quantities that are part of the trade space when designing a scan/sample strategy.

Within a beam you will get a range of SNR and hence uncertainty in the LOS Vel measurement (as a function of range). Setting up your runs with a single velocity error for all ranges is not representative of a true measurement. This effect is only exacerbated when you add a variable aerosol field as a function height as well.

Here, the same statement as above is valid. Due to the over-simplification in our existing approach we have removed this section from the manuscript.

Pg 8 Line 27 "System components" (and used elsewhere) this is an ambiguous term. In this case it would seem that you are referring to pointing vectors – but elsewhere you seem to have different meaning. System components refer to aircraft, scanner and lidar. We have reworded this section to clarify and be consistent throughout.

Pg 9 Line 6 "aircraft center of gravity..." The moment arm should be between the location of the INU and the center of the final turning mirror in the scanner. Correct, we changed it accordingly.

Pg 10 Line 2 "restore" do you mean isolate? Isolate is a better wording, we changed it accordingly.

Pg 10 Line 4 "system noise" No matter how well you apply the alignment calibration, there will always be a non-zero static offset. You should consider a sensitivity analysis of the wind profiles to static errors in the orientation. (rather than always assuming the errors are zero mean.) Here, the same statement as above is valid. Due to the over-simplification in our existing approach we have removed the noisy system simulation from the manuscript.

Pg 10 Line 34 "horizontal wind profile" It can remove the effect of a static profile, but not the natural variability. If you are trying to measure $w'^2$ – the variability in the horizontal wind will still manifest in the $w'^2$ profile. Correct, we changed the wording accordingly.

[revised manuscript text omitted]

---

## Author Response (AR2)

Dear Referee #3,

We thank you once again for the review of our manuscript and your remarks. We have revised the manuscript considering your remarks. We thereby hope to sufficiently address your comments and enable publication in AMT. Please find our answers to your minor comments below. Many thanks for your work and kind regards,

Philipp Gasch and Co-authors

The response is marked in this color

Minor comments:

The condensed version of the article reads well, and the significant effort put in by the authors is noticeable. The motivation is very clear now and makes helps the reader with the objective of the paper. Overall the paper looks publishable, except for a few minor clarifications mentioned below.

Page 6 Line 9: Replace "starting at a distance" with "a blind zone". Done.

Page 13/14 Line 6/20: I guess there is still not a clear conclusion on why winds deviation is above 1 m/s after strict data filtering for higher wind speeds? Is it not purely due to heterogeneities from your data and filtering? Are there further data filtering thresholds that remove those data points or maybe using better regression statistics on radial velocity rejection like using COOKS distance or DFITTS parameters (as in Bocciopio 1995)? Maybe an uncertainty analysis is important here. The observed errors are purely due to the violation of the homogeneity assumption used for retrieval. There is no reason why the retrieval error should not exceed 1 m s$^{-1}$. For example, the LES vertical wind speed for our highest background wind case of 15 m s$^{-1}$ can exceed 2 m s$^{-1}$. Worst case, full in-phase mapping (perfect horizontally sheared vertical wind field) of w = 2 m s$^{-1}$ vertical wind will results in a radial velocity amplitude change of $\Delta v_D$ = 1.73 m s$^{-1}$ compared to the original radial velocity in an undisturbed, homogeneous wind field (for the investigated system setup, scanning at $\varphi$ = 60 degree elevation, $\Delta v_D$ = w*sin($\varphi$), the radial velocity change due to the vertical wind as the lidar sees it). For perfect in-phase mapping, this change of $\Delta v_D$ = 1.73 m s$^{-1}$ corresponds to a horizontal wind speed (= erroneous mapping retrieval error, the lidar assumes this contribution is caused by a different horizontal wind than actually present) of $\Delta v_m$ = 3.46 m s$^{-1}$ ($\Delta v_m$ = $\Delta v_D$/cos($\varphi$)). This is the difference in horizontal wind which would explain such a change in the observed radial velocity. In this worst case, radial velocity rejection or better fitting techniques will not change the result as for an assumed full in-phase mapping (due to the horizontally sheared vertical wind field) the fit is perfect with no deviations. We agree, for sub-volume inhomogeneities an uncertainty analysis is important and we do so in an upcoming study. The effect of other retrieval parameters on retrieval quality was investigated in the original version of the manuscript but we had to remove this for the sake of clarity.

Page 14 line 25 – Steep is very subjective; can an elevation/zenith angle be provided here for reference instead? And the same for Page 15 line 1 (Shallow).

In line with our above statement we have reworded this section and tried to be more clear and precise.

Page 15 Line 6: Is it because the reduced number of samples? At what higher R2 threshold do you notice higher bias? We have reworded in order to avoid confusion. The bias appears due to the stronger filtering of in-phase vertical wind variations in the case of low background wind speeds. The evolution of retrieval bias with the correlation coefficient for our data is shown in figure B9, (b). From these results, for example setting the $R^2$ threshold to 0.6 will result in introducing biased retrievals with approx. +1.5 m s$^{-1}$ (only for the lowest background wind speed case).

Page 15 Line 19: This feels not a very practical solution. Are you are assuming that the winds are constant with height here? In case of significant shear, if the winds are < 5 m/s at the bottom 200 m, and > 5 m/s above 200 m, this type of analysis will not hold. Is my understanding correct? Maybe you can rephrase this section to better convey your thoughts. We have reworded and added a statement on the problem mentioned by you. Based on the presented results, we do not see another solution except for longer averaging to avoid the problem of erroneous mapping at the moment. Nevertheless, one should optimize the system setup and retrieval strategy in order to minimize the magnitude of this problem. For system optimization, scans closer to horizontal and faster scans will reduce the possible bias, as was presented in our first version of the manuscript. For the retrieval strategy, one can use wind speed (and thereby height dependent) analysis volumes. The along- and across track averaging distance are freely selectable parameters for the retrieval at any altitude, they do not need to be the same for all altitudes. In your example, if the wind speed is low close to the ground a longer averaging volume has to be applied at this altitude, whereas at higher altitudes a shorter spatial averaging is required. We now state so in the manuscript.

Page 15 Line 35: I am not sure I follow, what is an "in-situ uncertainty forecast"? Can you provide some references or some quick explanation? We have reworded this, as in-situ can be misleading in combination with lidar measurements. What we refer to is the possibility of an uncertainty estimation method, based on the measured data alone. This method should be able to capture the expected uncertainty in the retrieved wind speed due to the violation of the homogeneity assumption. Ideally, this method would be applicable to all meteorological situations, system setups and retrieval strategies. We think a promising approach is available through the measured radial velocity variance and we will show the functionality of this method in an upcoming study.

Page 16 Line 27: Remove "or even because of". Done.

Figure b7: The RMSE for 0m/s background wind case goes above 1 m/s? The marker disappears after ~ 100m. Similarly, the relative error. Thank you for noting this display error which slipped our attention before. We have adjusted the axis limits in order to display the RMSE fully. We are not displaying the REL for the 0 m s$^{-1}$ background wind case anymore, as it is not meaningful without background wind speed due to the division by 0 m s$^{-1}$.

[revised manuscript text omitted]